# Leveraging large-scale biobank EHRs to enhance pharmacogenetics of cardiometabolic disease medications

Marie C. Sadler [1,2,3], Alexander Apostolov [3], Caterina Cevallos [4], Chiara Auwerx [1,2,3,4], Diogo M. Ribeiro [3], Russ B. Altman [5] & Zoltán Kutalik [1,2,3] ✉

Electronic health records (EHRs) coupled with large-scale biobanks offer great promises to unravel the genetic underpinnings of treatment efficacy. However, medication-induced biomarker trajectories stemming from such records remain poorly studied. Here, we extract clinical and medication prescription data from EHRs and conduct GWAS and rare variant burden tests in the UK Biobank (discovery) and the All of Us program (replication) on ten cardiometabolic drug response outcomes including lipid response to statins, HbA1c response to metformin and blood pressure response to antihypertensives (N = 932-28,880). Our discovery analyses in participants of European ancestry recover previously reported pharmacogenetic signals at genome-wide significance level (*APOE*, *LPA* and *SLCO1B1*) and a novel rare variant association in *GIMAP5* with HbA1c response to metformin. Importantly, these associations are treatment-specific and not associated with biomarker progression in medication-naive individuals. We also found polygenic risk scores to predict drug response, though they explained less than 2% of the variance. In summary, we present an EHR-based framework to study the genetics of drug response and systematically investigated the common and rare pharmacogenetic contribution to cardiometabolic drug response phenotypes in 41,732 UK Biobank and 14,277 All of Us participants.

Genetic factors can contribute to inter-individual variability in drug response. However, despite the immense progress of genome-wide association studies (GWAS) for complex traits and diseases, the scale of pharmacogenetics (PGx) studies to find genetic predictors of drug efficacy remains limited. PGx GWAS represent less than 10% of all entries in the GWAS Catalog with median sample sizes of 1220 for PGx GWAS published between 2016 and 2020[1]. As a result of low sample size and lack of cohorts suitable for pharmacogenomic studies, relatively few PGx associations determining drug efficacy have been identified in a genome-wide approach[1–3].

Several PGx GWAS consortia have formed over the years to study the genetics of drug efficacy in larger sample sizes by investigating the change in biomarker levels following medication start. For instance, the Genomic Investigation of Statin Therapy (GIST) consortium has identified variants in the *LPA*, *APOE*, *SORT1/CELSR2/PSRC1* and *SLCO1B1* regions as modulators of low-density lipoprotein cholesterol (LDL-C) response to statins by combining randomized controlled trials (RCTs) and observational studies[4]. Using electronic health records (EHRs), the Genetic Epidemiology Research on Adult Health and Aging (GERA) cohort has additionally identified the *APOB* and *SMARCA4/LDLR* loci as

[1]University Center for Primary Care and Public Health, Lausanne, Switzerland. [2]Swiss Institute of Bioinformatics, Lausanne, Switzerland. [3]Department of Computational Biology, University of Lausanne, Lausanne, Switzerland. [4]Center for Integrative Genomics, University of Lausanne, Lausanne, Switzerland. [5]Department of Bioengineering, Stanford University, Stanford, CA, USA. ✉e-mail: zoltan.kutalik@unil.ch

genetic determinants of statin response[5,6]. Similarly, the Metformin Genetics (MetGen) consortium has identified *SLC2A2* as influencing haemoglobin A1c (HbA1c) response to metformin[7], and more recently a meta-GWAS on HbA1c response to GLP-1 receptor agonists found variants in *ARRB1* to influence drug efficacy[8]. Furthermore, the International Consortium for Antihypertensive Pharmacogenomics Studies (ICAPS) has published multiple GWAS investigating blood pressure response to several antihypertensive drug classes (beta blockers, calcium channel blockers (CCBs), thiazide/thiazide-like diuretics and ACE-inhibitors (ACEi)/angiotensin receptor blockers (ARB))[9–11].

Biobanks coupled with EHRs that comprise medication data provide new opportunities to discover PGx associations[1,12,13]. These massive datasets have already contributed to the replication of known PGx interactions as well as the discovery of new putative associations in national biobanks such as the Estonian[14] and UK Biobank (UKBB)[15,16]. More recently, GWAS on longitudinal medication patterns extracted from the Finnish nationwide drug purchase registry in the FinnGen study identified tens of cardiometabolic risk loci specific to medication use and not associated with the underlying indication[17]. Yet, PGx biobank studies so far have either focused on known pharmacogenes and their associations with adverse drug reactions, drug dosage and drug prescribing behaviour or analyzed the genetics of temporal medication use in isolation of disease phenotypes. Except for the GERA cohort which solely utilized EHRs for the study of LDL-C response to statins[5,6], the integration of longitudinal medication and phenotypic data to screen for genetic determinants of drug efficacy at a biobank scale remains largely unexplored.

Here, we extracted clinical and medication prescription data from EHRs and conducted PGx association analyses on the change in biomarker levels following drug therapy to treat cardiometabolic diseases (Fig. 1a). We performed PGx GWAS to assess the contribution of common variants to drug response and compared the results with GWAS conducted on RCT and observational data. Furthermore, we assessed the cumulative impact of rare variants on drug response by conducting rare variant burden tests using sequencing data. Discovery analyses were conducted in the UK Biobank (UKBB)[18] and replication analyses in the All of Us (AoU) research program[19] (Fig. 1b). In follow-up analyses, we compared drug response genetics to the genetics of baseline and longitudinal biomarker changes in medication-naive individuals to dissect medication- and disease-specific components while also highlighting common pitfalls in the analysis of longitudinal (response) phenotypes (Fig. 1c). Finally, we demonstrated that polygenic risk scores (PRS) of the underlying condition can predict drug response. In summary, we provide guidance on how to design drug response studies with longitudinal medication prescriptions and biomarker measures stemming from real-world data, introduce a more reliable model for studying genetic associations with drug response and present a comprehensive resource on the genetic architecture of cardiometabolic drug response. Our study showcases the value as well as the challenges when analyzing EHR-coupled biobanks to study interindividual variability in drug response and identify clinically relevant genetic predictors.

## Results
### Overview of the analysis
In the drug response discovery analyses, we extracted longitudinal prescription and response biomarker data from the UKBB primary care records which we combined with phenotypic data from the assessment visits (currently $\approx$ 230,000 (45%) UKBB participants are linked to their EHRs). We then emulated EHR-derived drug response cohorts for the following medication-biomarker pairs: statin-lipids (LDL-C, high-density lipoprotein cholesterol (HDL-C), total cholesterol (TC)), metformin-HbA1c, antihypertensive-systolic blood pressure (SBP; by antihypertensive class (ACEi, CCB, thiazide diuretics) and all classes combined), beta blocker-SBP and beta blocker-heart

rate (HR). Individuals were only part of a drug response cohort if a phenotype measurement was available before and after treatment initiation in addition to passing several other quality control (QC) steps (Method section: Study design and phenotype definitions, Supplementary Fig. 1, Supplementary Data 3). For each drug response phenotype, we derived an absolute and logarithmic relative biomarker difference as outcome traits as both approaches are commonly employed in drug response studies. Lipid and blood pressure response have been studied on both the absolute[9,10,20–22] and relative scale[4,6,20–23], while HbA1c response has commonly been analyzed on the absolute scale[7,24,25]. Furthermore, we considered two filtering scenarios to define drug response phenotypes, a stringent and a lenient one. More stringent QC should result in a cleaner phenotype definition, with the trade-off of reduced sample size (and thus potentially lower statistical power). Given the sharp drop in sample size with more stringent criteria, the lenient filtering strategy constitutes the default setting throughout this study. In both stringent and lenient scenarios, we tested single and average baseline and post-treatment values over multiple measures, if available, with average values being the default (Fig. 1).

In each drug response cohort, we first conducted GWAS to discover common genetic predictors (minor allele frequency (MAF) $\geq$ 0.05) of drug efficacy. In a second step, we performed genome-wide burden tests using whole exome sequencing (WES) data to assess associations with rare variants (MAF < 0.01). Replication analyses of identified PGx variants in the discovery analyses and across the literature were conducted in ~ 250,000 participants of the AoU research program with available whole genome sequencing data (WGS). In follow-up analyses, we showcase the biases emerging from the popular approach of regression-based baseline adjustment to derive drug response outcomes and assess baseline trait PRS as predictors of drug response.

### Drug response GWAS using EHRs from the UKBB
Following cardiometabolic drug treatment, lipid, HbA1c and blood pressure levels significantly dropped while HDL-C levels moderately increased ($\Delta$HDL-C = 0.012 mmol/L, two-sided paired *t* test *p*-value = 2.41e−23; Fig. 2a; Supplementary Data 4). In comparison, biomarker levels measured during an equivalent time window as baseline and post-treatment levels remained stable for control individuals who did not take any related medications (Fig. 2a; Supplementary Data 7). In the LDL-C response to statin GWAS, *APOB* (rs10199768 T > G, beta = 0.056, *p*-value = 1.77e−08) and *LDLR* (rs118068660 T > C, beta = −0.119, *p*-value = 1.23e−10) were found to influence absolute biomarker change, while the *SLC22A3/LPA* (rs10455872 G > A, beta = −0.146, *p*-value = 4.87e−17) and *APOE* (rs7412 T > C, beta = 0.232, *p*-value = 1.11e−28) loci were found to influence relative (logarithmic) biomarker change (Table 1; Fig. 2b; lenient filtering with average values if available, *N* = 18,753). All four genes encode proteins with a well-established role in serum lipids level regulation, with *APOB*, *APOE*, and *LPA* encoding for apolipoproteins that bind lipids such as cholesterol and triglycerides and organize them into various types of lipoprotein particles (e.g., LDL-C), enabling their transport in the blood and distribution throughout the body[26]. The two former apolipoproteins further act as ligands for the LDL receptor encoded by *LDLR*, allowing the binding and internalization of lipoprotein particles harboring these lipoproteins, ensuring delivery of lipids to the cell and regulation of serum LDL-C levels[26]. Importantly, all four genes have previously been involved in modulating response to statins[4,6,20,27]. More specifically, the CARDS trial showed that the PGx association in the *SLC22A3/LPA* locus results from LDL-C levels also including LDL-C residing in Lp(a) particles[27]. The *LPA* variant rs10455872-G which is associated with a lower response to statins also associates with increased levels of Lp(a) which remain unchanged upon statin treatment. Thus, the relative higher proportion of Lp(a) particles in rs10455872-G carriers gives rise

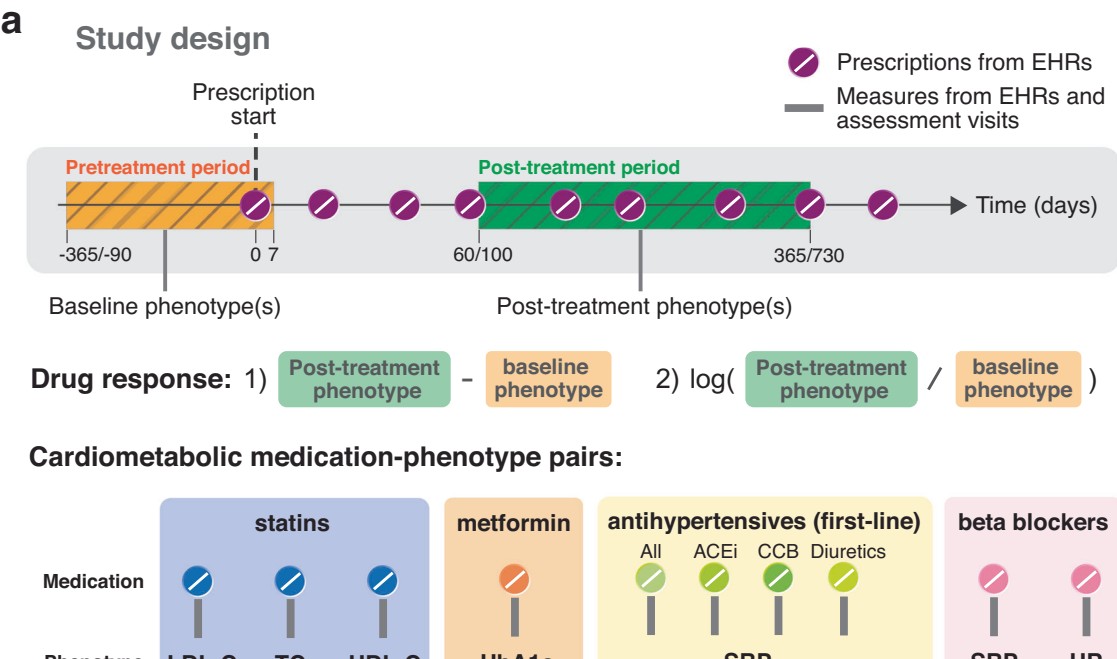

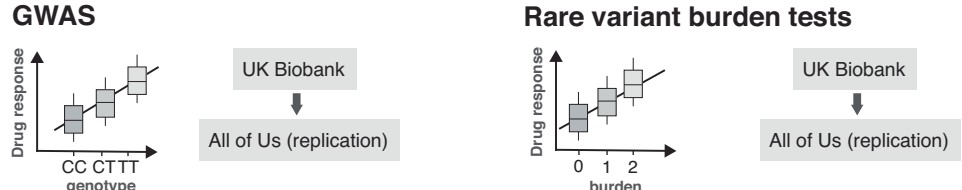

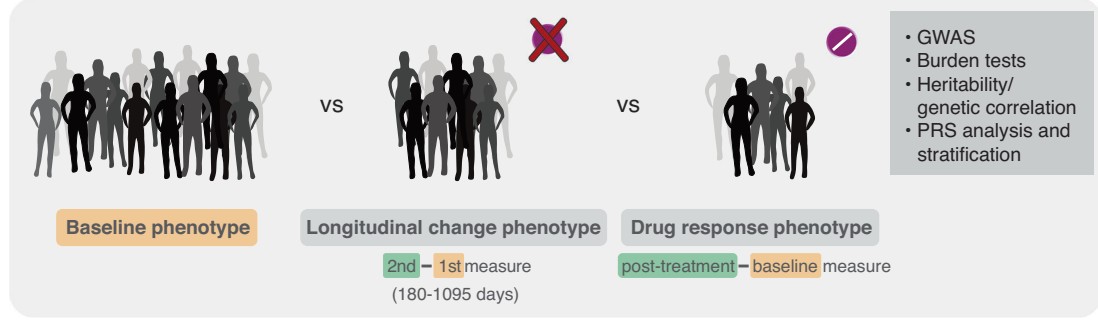

**Fig. 1 | Study design. a** Drug response study design using electronic health records (EHRs) from the UK and All of Us biobanks. Baseline (orange) and post-treatment (green) phenotypes were extracted from EHRs or biobank assessment visits before and after the first recorded prescription, respectively. Different timings relative to the first prescription were tested as well as the use of single and average values over multiple baseline and post-treatment measures if available. Drug response phenotypes defined by the 1) absolute and 2) relative logarithmic difference in post-treatment and baseline biomarker measures were tested for ten cardiometabolic medication-phenotype pairs: LDL cholesterol (LDL-C), total cholesterol (TC) and HDL cholesterol (HDL-C) response to statins (blue); HbA1c response to metformin (orange); systolic blood pressure (SBP) response to antihypertensives (green, ACE-inhibitors (ACEi), calcium channel blockers (CCBs) and diuretics); SBP and heart rate (HR) response to beta blockers (purple). **b** Discovery genetic association analyses were conducted in the UK Biobank and replicated in the All of Us research program on common variants (GWAS analysis) and rare variants through burden tests. **c** Follow-up analyses compared the genetics of baseline, longitudinal change and drug response genetics including polygenic risk score (PRS) analysis.

to this PGx association even if LDL-C residing in statin-responsive LDL particles drop to similar levels as in rs10455872-A carriers.

TC response to statins, for which we had a larger sample size (more TC than LDL-C measures are available in the primary care data, N = 28,880) confirmed the identified loci at *APOB*, *LDLR*, *SLC22A3/LPA* (the latter being identified in both the absolute and relative biomarker change GWAS), and *APOE*, while also identifying the SNP rs4149056 C > T in the *SLCO1B1* locus (beta = −0.063, *p*-value = 1.10e−08). *SLCO1B1* encodes for the OATP1B1 transporter, which mediates the intracellular uptake of a wide range of substrates, including statins[28,29]. Importantly, the variant we identified, also known as Val174Ala or SLCO1B1*5, has previously been associated with LDL-C statin response[4,30] as well as

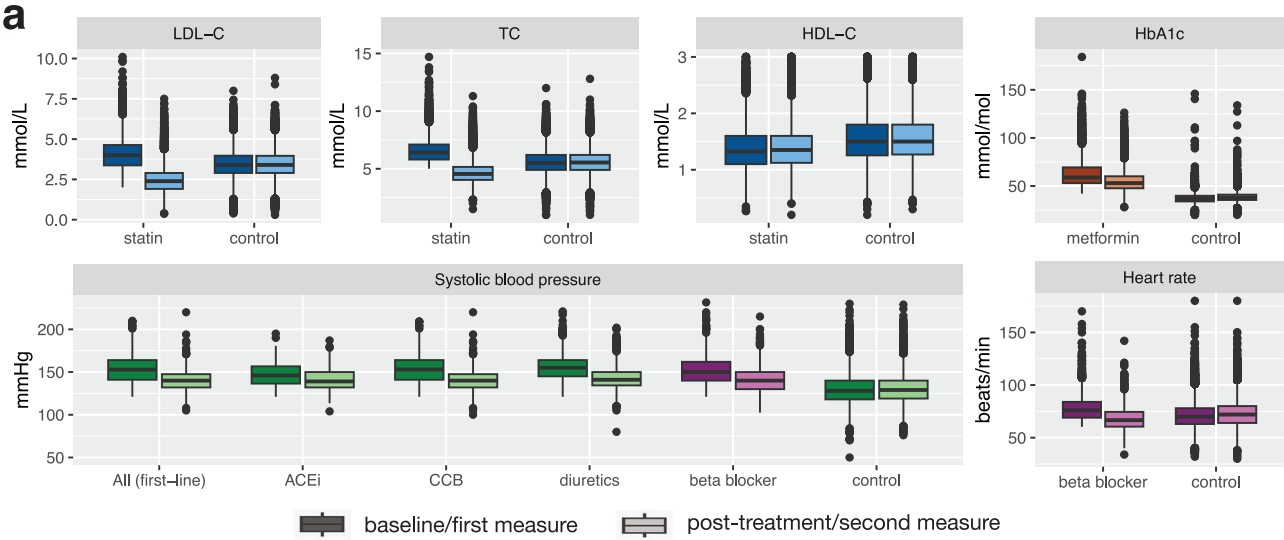

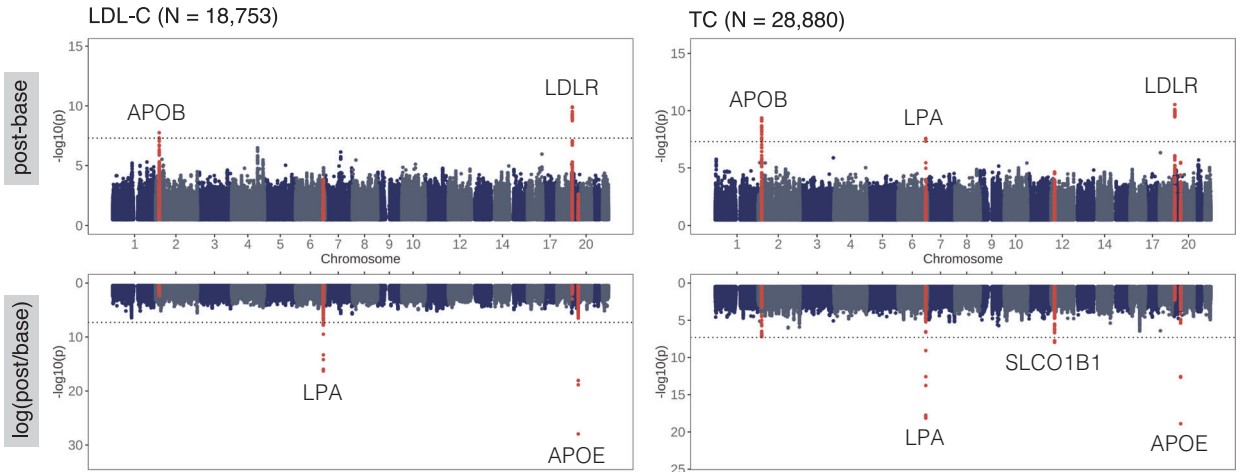

**Fig. 2 | EHR drug response phenotypes and PGx GWAS results derived from the UKBB. a** Baseline and post-treatment biomarker levels of statin (blue), metformin (orange), first-line antihypertensives (green) and beta blocker (purple) medication users as well as first and second measures of controls who do not take any related medications. Boxes bound the 25th, 50th (median, centre), and the 75th quantile of LDL-C post-treatment measures. Whiskers range from minima (Q1 - 1.5*IQR) to maxima (Q3 + 1.5*IQR) with points above or below representing potential outliers. Drug response sample sizes are in Supplementary Data 4 and control sample sizes in Supplementary Data 7. LDL cholesterol (LDL-C); total cholesterol (TC); HDL cholesterol (HDL-C); ACE-inhibitor (ACEi); calcium channel blocker (CCB). **b** Manhattan plots of LDL-C and TC response to statins. GWAS association results of the top and bottom show the absolute and logarithmic relative biomarker differences, respectively. GWAS were performed using a linear additive model, with a two-sided test of association. Loci with genome-wide significant signals (p-value < 5e−8) for either the absolute or relative difference are highlighted in red. All loci are annotated with the closest gene and the horizontal line denotes genome-wide significance (p-value < 5e−8). Results in (**a**) and (**b**) correspond to the lenient filtering setting with average values over multiple measures, if available.

myopathy[31], a rare but well-described side effect that has been attributed to increased statin blood concentrations due to the reduced uptake capacity of the encoded protein[29].

No genome-wide significant hits were found in the HDL-C response to statin GWAS ($N = 25,405$), HbA1c response to metformin GWAS ($N = 4424$), HR ($N = 2157$)/SBP ($N = 1750$) response to beta blockers, and SBP response to antihypertensives ($N = 932–8250$; Supplementary Figs. 6, 7).

### The impact of filtering in the PGx-EHR study design
In EHRs, medication start, baseline and post-treatment measures are not readily available, and we tested multiple strategies to extract drug response phenotypes and assess their impact on PGx associations. We introduced a prescription regularity parameter to proxy drug adherence under the assumption that skipped prescriptions are the result of inconsistent medication intake. This prescription regularity parameter (or prescription completeness) is defined by the presence of a prescription at least every two months for the duration of the post-treatment period and a completeness of 100% is obtained if this is the case. To account for varying time window thresholds, prescription regularities, changes in medication regimens, and single or multiple biomarker measures at baseline and post-treatment, we defined stringent and lenient filtering strategies, assessing the impact of using single versus multiple biomarker measures for each.

A main difference between the lenient and stringent filtering scenario was the extension of the baseline and post-treatment periods. In Supplementary Fig. 4, we show the distribution of the time between the closest baseline measure and prescription start, and in Supplementary Fig. 5, the distribution of the time between prescription start and the closest post-treatment measure. The median of this distribution was between 0 and 41 days for the baseline measure and 111 and 273 days for the post-treatment measure across medication-biomarker

**Table 1 | Genome-wide significant loci in discovery analyses (UK Biobank) across all assessed cardiometabolic drug response traits together with replication results (All of Us)**

| Pharmacogenetics trait | Response definition | Chr | Position (GRCh37) | Lead SNP | Gene | Effect allele | Other allele | UKBB (discovery) | | | | AoU (replication) | | | |
|---|---|---|---|---|---|---|---|---|---|---|---|---|---|---|---|
| | | | | | | | | EAF | N | Beta | p-value | EAF | N | Beta | p-value |
| LDL-C response to statins | post-base | 2 | 21244000 | rs10199768 | APOB | G | T | 0.539 | 18753 | 0.056 | 1.77E-08 | 0.613 | 10335 | -0.018 | 0.18 |
| LDL-C response to statins | post-base | 19 | 11190544 | rs118068660 | LDLR | C | T | 0.911 | 18753 | -0.119 | 1.23E-10 | 0.884 | 10325 | -0.003 | 0.90 |
| LDL-C response to statins | log(post/base) | 6 | 161010118 | rs10455872 | SLC22A3, LPA | A | G | 0.910 | 18753 | -0.146 | 4.87E-17 | 0.950 | 10336 | -0.093 | 1.52E-03 |
| LDL-C response to statins | log(post/base) | 19 | 45412079 | rs7412 | APOE | C | T | 0.941 | 18753 | 0.232 | 1.11E-28 | 0.938 | 10337 | 0.215 | 3.56E-16 |
| TC response to statins | post-base | 2 | 21208211 | rs7557067 | APOB | A | G | 0.782 | 28880 | -0.061 | 4.34E-10 | 0.771 | 6997 | -0.000 | 0.99 |
| TC response to statins | post-base | 6 | 161010118 | rs10455872 | SLC22A3, LPA | A | G | 0.910 | 28880 | -0.079 | 2.62E-08 | 0.947 | 6998 | -0.054 | 0.12 |
| TC response to statins | post-base | 19 | 11190481 | rs77265569 | LDLR | G | T | 0.902 | 28880 | -0.091 | 2.92E-11 | 0.887 | 6846 | -0.013 | 0.61 |
| TC response to statins | log(post/base) | 6 | 161010118 | rs10455872 | SLC22A3, LPA | A | G | 0.910 | 28880 | -0.124 | 7.10E-19 | 0.947 | 6998 | -0.083 | 1.62E-02 |
| TC response to statins | log(post/base) | 12 | 21331549 | rs4149056 | SLCO1B1 | T | C | 0.849 | 28880 | -0.063 | 1.10E-08 | 0.858 | 6999 (10337[a]) | -0.037 (-0.045[a]) | 9.17E-02 (1.31E-02[a]) |
| TC response to statins | log(post/base) | 19 | 45412079 | rs7412 | APOE | C | T | 0.941 | 28880 | 0.153 | 1.28E-19 | 0.941 | 6999 | 0.154 | 1.84E-06 |

GWAS were performed using a linear additive model, with a two-sided test of association.

Chr chromosome, EAF frequency of effect allele, post-base absolute biomarker difference, log(post/base) logarithmic (relative) biomarker difference.

[a]Results for LDL-C.

pairs (Supplementary Data 4). As we anticipate the stringent filtering scenario to more closely reflect the design of an RCT such as the JUPITER trial (LDL-C response to rosuvastatin) where the post-treatment value was taken 1 year following medication start[20], we chose a baseline period starting 100 days before prescription start and a post-treatment period starting 60 (antihypertensives and beta blockers) or 100 days (statins and metformin which have delayed effects on lipids[32] and HbA1c[33]) up to 1 year following prescription start. In the lenient filtering, we extended the baseline period up to 1 year preceding and up to 2 years following prescription start. Note that the post-treatment period in the lenient filtering setting remains more stringent than in observational studies such as those included in the GIST study where the post-treatment period varied widely between cohorts and could last up to 5 years[4]. When testing the impact of the time to first post-treatment measure on the biomarker difference, we observed that increased follow-up times resulted in reduced biomarker differences, even in the stringent filtering scenario where this parameter could not vary as widely (the follow-up time explained up to 1.1% of the variance; Supplementary Data 6). Likewise, a lower prescription completeness value resulted in decreased biomarker differences in the stringent and the lenient filtering settings, where the required completeness were 60% and 30%, respectively (the prescription regularity explained up to 7.6% of the variance; Supplementary Data 6). Despite the differences in the stringent and lenient filtering scenarios, average baseline and post-treatment biomarker levels were almost identical between the resulting drug cohorts (differences of less than 3%; Supplementary Data 4).

Between the two filtering strategies, the sample size increased from 40% (metformin-HbA1c) up to 334% (CCB-SBP) when relaxing the filtering criteria (Supplementary Data 4). For statins, this rise was largely due to the extended baseline and post-treatment period. For metformin and antihypertensives, we excluded individuals taking any related medication in the stringent filtering setting, whereas, in the lenient setting, sample size largely increased by allowing metformin and antihypertensives to act as add-on therapy to sulfonylureas and second-line antihypertensives, respectively, if consistently taken during pre- and post-treatment periods of the studied medication (Supplementary Figs. 2, 3). As a consequence of lower statistical power, only 4 out of the 10 signals found in the lipid-statin GWAS were detected in the stringent filtering scenarios (Supplementary Figs. 8, 9; Supplementary Data 8). Furthermore, we tested the difference between assessing a single baseline and post-treatment measure (the closest to the prescription start) and averaging over all available measures present in the baseline and post-treatment periods. The impact was minimal, and the only difference was observed for the SLCO1B1-associated SNP rs4149056 which did not reach genome-wide significance with a single measure (p-value =1.49e−5).

### Replication analysis in the All of Us research program

We conducted replication analyses in the AoU program (v7; N ≈ 250,000 with available short-read WGS data). As in the UKBB, longitudinal prescription and phenotypic data were extracted from EHRs and used to construct drug response cohorts by following the same methodology as in the UKBB (Methods; Supplementary Data 9). Cohort characteristics were similar to those in the UKBB (Supplementary Data 9; Supplementary Fig. 10). The mean statin starting age was 58 years compared to 61 years in the UKBB and as in the UKBB post-treatment lipid levels were on average measured within a year (average of 290 days) following the first prescription. The main difference was observed in the regularity of statin prescriptions. Whereas in the UKBB, participants had on average a prescription every two months 89% of the time, this number dropped to 44% in the AoU. There were slightly fewer statin users than in the UKBB, but similar to the UKBB, the main reasons for being excluded in the PGx cohort were missing baseline and/or post-treatment measures in the considered

time windows leaving 10,337 and 6999 individuals in the LDL-C and TC response to statins, respectively. Among the 10 signals, 3 signals implicating the *APOE* and *LPA* loci replicated at the Bonferroni-corrected replication threshold of 0.05/10 = 0.005 and 4 at a nominal *p*-value of 0.05 (all directionally concordant; Table 1). A fifth signal, *SLCO1B1*, replicated at nominal significance (*p*-value = 1.31e−02) when considering LDL-C instead of TC. Signals not replicating nominally include the *APOB* and *LDLR* loci.

Participants in the AoU biobank represent a more diverse range of genetic ancestries compared to those in the UKBB, with only 50% being of European ancestry[34]. When assessing allele frequencies of the identified PGx signals across ancestries by using the gnomAD v4.1.0 resource[35], the largest allele frequency differences were observed for *APOB*, ranging from 5.2% in East Asians to 52.2% in the Amish (Supplementary Data 10). Significant differences were also found for *LDLR* (0.8–18.8%) and *SLCO1B1* (3.1–20.9%), while MAFs of other PGx variants varied by less than 10%. Based on this MAF spectrum, we conducted power analyses to determine the sample sizes required to detect PGx signals at genome-wide significance (5e−08) in different ancestral groups. For a strong signal, such as rs7412 at the *APOE* locus (beta = 0.232, *p*-value = 1.11e−28), a sample size of 4000 is needed to achieve 80% power in African/African Americans where the MAF is the highest (10.5%) compared to 28,500 in Amish people where the MAF is the lowest (1.2%). For a more modest signal, such as rs4149056 at the *SLCO1B1* locus (beta = −0.063, *p*-value = 1.10e−08), a sample size of 148,501 is needed for 80% power in African/African Americans where the MAF is the lowest (3.1%) compared to 27,501 in the Finnish where the MAF is the highest (20.9%; Supplementary Fig. 11).

## EHR-derived PGx GWAS recover known PGx loci

From the literature, we extracted genetic predictors reported for the assessed cardiometabolic medication-biomarker pairs. We adopted the criteria from Nelson et al., 2016[2] that provide a curated list up to July 2015 by querying the GWAS Catalog[36]. Briefly, genetic variants were required to pass the genome-wide significance threshold of 5e−8 and show evidence of replication. Reported GWAS stem either from RCTs, EHRs (GERA cohort[5,6]) or observational studies often meta-analyzed together. As we will elaborate in the longitudinal phenotype model later on (Fig. 3a), adjusting biomarker changes for baseline levels induces spurious associations for genetic variants that are also associated with baseline levels. However, 5 of the 7 studies reporting significant PGx variants have adjusted for baseline levels[4,6,23–25]. Thus, these reported loci could represent either baseline genetic or pharmacogenetic effects, or both. To be concordant with these studies, we report literature replication *p*-values for baseline-adjusted and unadjusted biomarker change.

Seven independent loci were reported for LDL-C response to statins of which three (*APOE*, *LPA* and *SORT1*) and two (*APOE* and *SORT1*) passed genome-wide significance in the (baseline adjusted) discovery (UKBB) and replication (AoU) cohort, respectively (Table 2). *SLCO1B1* locus was nominally significant in the UKBB (*p*-value = 1.84e−03) and genome-wide significant in the TC response GWAS for which sample size was larger (*p*-value = 1.89E−09, baseline adjusted). *ABCG2* associated with LDL-C reduction following rosuvastatin therapy in the JUPITER trial[20] was found to be insignificant in the UKBB and AoU (*p*-values of > 0.05) and did also not reach genome-wide significance in a later, larger GWAS meta-analysis of all statins combined[4]. The *LDLR* and *APOB* loci reached genome-wide significance in the baseline-adjusted GWAS in the GERA cohort, but were believed to be false positives as a consequence of baseline adjustment[6]. These SNPs were nominally significant in our baseline adjusted biomarker difference GWAS. However, the reported *LDLR*-associated SNP rs67337506 is in LD with the rs118068660 SNP ($r^2$ = 0.29) which we identified as a genuine PGx locus in the absolute LDL-C change GWAS (Table 1). Similarly, the reported *APOB*-associated SNP rs1713222 is in LD with the

rs10199768 SNP ($r^2$ = 0.14) which reached genome-wide significance in the absolute LDL-C change GWAS. Thus, although both loci were identified by a biased PGx model, SNPs in LD reached genome-wide significance in our unadjusted baseline GWAS. The HDL-C response GWAS to statins (baseline adjusted) identified *CETP* as a single genome-wide significant locus, which replicated at a genome-wide significance level in the UKBB[23]. Overall, EHR-derived PGx signals on lipids agree well with those reported in cohort studies, although, baseline adjustment can lead to spurious associations for variants that associate with the baseline levels of the biomarker. For instance, rs247616 (*CETP*) was strongly associated with HDL-C change when adjusting for baseline in the UKBB (*p*-value = 3.63E−10), but no longer in the unadjusted analysis (*p*-value > 0.3). Associations like *CETP* and *SORT1* that reached genome-wide significance only upon baseline adjustment were also found in our negative control analysis, where we conducted a baseline-adjusted longitudinal change GWAS in drug-naive participants (Supplementary Fig. 17), confirming our suspicion that baseline adjustment leads to biased results (see Section on *Modelling drug response and longitudinal change phenotypes*).

GWAS of HbA1c-response to metformin identified *ATM*[24], *SLC2A2*[7] and *PRPF31*[25], but only the *SLC2A2*-associated SNP (discovered in a baseline-unadjusted analysis) was recovered at nominal significance (*p*-value of 3.89E−02 and 4.07E−02 in the absolute and relative biomarker change model, respectively). *SLC2A2* was discovered in a sample size of 10,577 individuals[7], whereas sample sizes were 4424 and 3845 in the UKBB and AoU, respectively. Nonetheless, it should be noted that none of the metformin studies have reported the same locus twice and the *ATM* and *SLC2A2* loci were insignificant in the ACCORD clinical trial GWAS that was conducted later (*p*-value > 0.1)[25]. Although several loci have been found to influence blood pressure and heart rate response to anti-hypertensives at a suggestive *p*-value threshold, no genome-wide significant hits have been reported[37]. Among the 13 loci identified at a suggestive significance level in studies conducted in samples exceeding 300 participants[37], only the SNP rs4149601 in the *NEDD4L* gene region found to influence SBP response to diuretics could be replicated at a nominal significance level in the UKBB (*p*-value = 0.032; Supplementary Data 11). However, replication only occurred in the baseline adjusted analysis which is more likely to yield false positives (Fig. 3a).

## Rare variants have a modest impact

While common genetic variants have been assessed as predictors of drug response phenotypes in multiple studies, the impact of rare variation is less well known. Making use of sequencing data (WES and WGS in the UKBB and AoU, respectively), we conducted rare variant burden tests for all ten drug response phenotypes (Fig. 1). We included missense and putative loss-of-function (LoF) variants with MAF < 1% in optimal kernel association tests (SKATO)[38]. After correcting for multiple testing (*p*-value < 0.05/18,983 = 2.63e−06), we identified *GIMAP5* to impact absolute HbA1c response to metformin (*p*-value (SKATO) = 2.28e−06). Directionality could be inferred from an additive burden test which revealed that a higher burden is associated with a reduced biomarker reduction (additive burden test: beta = 0.66, *p*-value = 1.75e−06). *GIMAP5* is known to regulate lymphocyte function and survival, particularly in T-cells and it has previously been associated with autoimmune diseases such as type 1 diabetes, lupus, and inflammatory bowel disease[39]. When restricting the analysis to known pharmacogenes (66 very important (VIP) autosomal pharmacogenes defined by PharmGKB[40]), we found a significant association of rare variants in the *CYP1A2* gene to influence absolute HDL-C reduction following statin treatment (*p*-value = 2.37e−04). Neither *GIMAP5* (*p*-value = 0.53) nor *CYP1A2* (*p*-value = 0.35) replicated in the AoU indicating either false positive associations or a statistical power issue due to the lower sample size and the inclusion of different rare variants in the AoU.

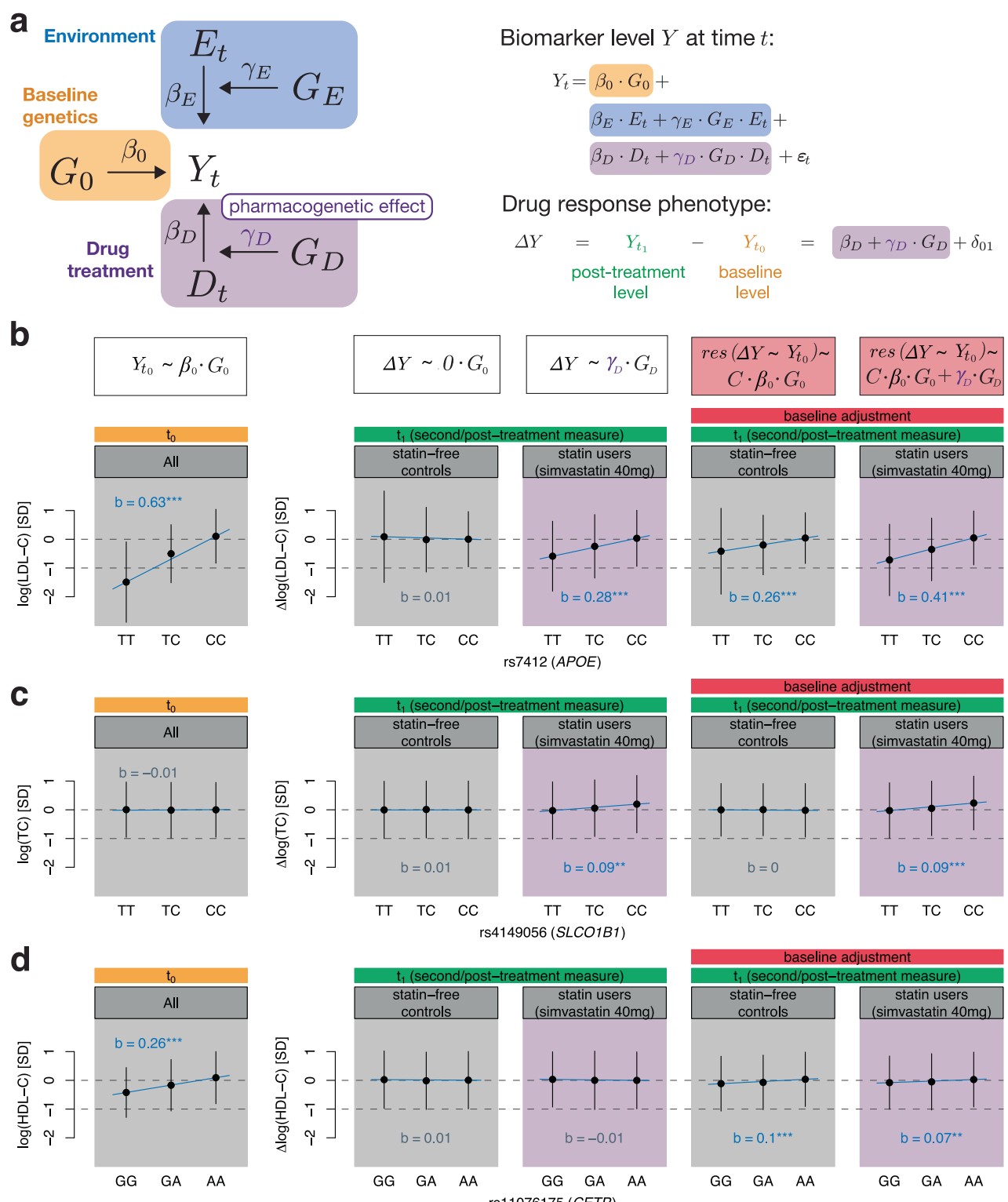

## Modelling drug response and longitudinal change phenotypes

In the following, we will propose a longitudinal phenotype model to analyze drug response and longitudinal change phenotypes in an unbiased manner. Biomarker levels $Y$ at time $t$ can be modelled as follows (Fig. 3a):

$$Y_t = \beta_0 \cdot G_0 + \beta_E \cdot E_t + \gamma_E \cdot G_E \cdot E_t + \beta_D \cdot D_t + \gamma_D \cdot G_D \cdot D_t + \epsilon_t$$

where $\beta_0$ is the baseline genetic effect, $G$ the genetics, $\beta_E$ the environmental effect, $E$ the environment, $\gamma_E$ the gene-environment interaction effect, $D$ the indicator of drug use, $\beta_D$ the drug effect and $\gamma_D$ the pharmacogenetic effect.

When modelling the drug response as the difference of post-treatment levels $Y_{t_1}$ and baseline levels $Y_{t_0}$, where the drug status is 1 and 0 at $t_1$ and $t_0$, respectively, the drug response phenotype simplifies

**Fig. 3 | Modelling longitudinal changes of biomarker levels with (or without) treatment effect. a** Biomarker levels $Y$ at time $t$ can be influenced by baseline genetics $G_0$ (orange), environment $E$ and gene-environment interactions ($G_E \cdot E$, blue), and drug status $D$ and pharmacogenetic interactions ($G_D \cdot D$, purple). Drug response phenotypes modelled as the difference of post-treatment ($t_1$) and baseline ($t_0$) levels allow the estimation of the pharmacogenetic effect $\gamma_D$ through genetic regression analyses (Supplementary Note 1). **b–d** Stratification at genetic variants that harbour pharmacogenetic $\gamma_D$ (**b**, **c**) and/or baseline $\beta_0$ (**b**, **d**) genetic effects. Adjusting drug response or longitudinal change phenotypes for baseline induces a bias that scales with $C \cdot \beta_0$ where $C$ equals $1 - (\beta_E^2 + \gamma_E^2) \cdot \mathrm{corr}(E_t, E_0)$ (Supplementary Note 1, red). Thus, variants with significant baseline effects spuriously associate with drug response phenotypes even if $\gamma_D$ is zero (**d**). Such measure of change, however, shows association in drug-naive individuals too. The baseline panel ($t_0$) groups statin-free controls and statin users (simvastatin 40mg corresponding to the largest starting statin type-dose group), and shows their sex and age-adjusted standardized baseline level stratified by genotype. The following four panels ($t_1$) show standardized longitudinal change (drug-naive individuals) and drug response phenotypes (statin users) adjusted for sex and age, once unadjusted (correct model) and adjusted (biased model) for baseline levels. Genotype regression coefficients (denoted with $b$) with baseline lipid levels, longitudinal change and drug response phenotypes were derived through regression of the standardized outcome measures on the genotype dosage adjusted for sex and age as well as baseline levels if indicated. The significance level of the slope ($b$) is indicated by colour and stars where grey indicates a $p$-value $> 0.05$, blue a $p$-value $\leq 0.05$, 2 stars a $p$-value $< 1e{-}3$ and 3 stars a $p$-value $< 5e{-}8$ (two-sided test statistics). Dots correspond to the mean and error bars to the standard deviation of covariate-adjusted baseline levels and drug response/longitudinal change phenotypes in each stratified group (numbers of individuals per stratum are shown in Supplementary Data 12). LDL cholesterol (LDL-C); total cholesterol (TC); HDL cholesterol (HDL-C).

to (Supplementary Note 1):

$$\Delta Y = Y_{t_1} - Y_{t_0} = \beta_D + \gamma_D \cdot G_D + \delta_{01}$$

Thus, the pharmacogenetic effect can be estimated from genetic regression analyses on the biomarker difference at post-treatment and baseline. Control individuals who do not take any related medications are not required for this estimation, however, analyzing longitudinal changes in these individuals (the drug status being zero at both time points) serves as a control to ensure that identified PGx signals are specific to drug treatment. Supplementary Fig. 12 presents the directed acyclic graph (DAG) that extends the graph in Fig. 3a for modeling the genetics of drug response phenotypes $\Delta Y$. This expression also holds when modelling the logarithm of the biomarker level. However, adjusting biomarker differences for baseline levels induces a bias when genetic variants are also associated with baseline levels ($\beta_0 \neq 0$). It can be shown that this bias can be approximated by $C \cdot \beta_0$ where $C$ equals $1 - (\beta_E^2 + \gamma_E^2) \cdot \mathrm{corr}(E_t, E_0)$ (Supplementary Note 1).

In Fig. 3b–d, we depict genetic variants that either have a significant pharmacogenetic effect $\gamma_D$, baseline effect $\beta_0$ or both, and showcase how baseline adjustment can introduce a bias in genetic effect estimation (Supplementary Data 12, 13). To this end, we compared genetic effect sizes of biomarker differences in medication-naive controls to those in statin users (simvastatin 40mg users who represent the largest starting statin type-dose group; Supplementary Data 5). The *APOE* missense variant rs7412 which is strongly associated with baseline levels ($\beta_0 = 0.634$, $p$-value $< 1e{-}300$) also exhibited a pharmacogenetic effect ($\gamma_D = 0.284$, $p$-value $= 3.52e{-}19$) while not being associated to longitudinal change in statin-free controls ($p$-value $= 0.59$; Fig. 3b; Supplementary Data 7). However, upon baseline adjustment, a significant genetic effect with longitudinal change was observed in drug naive individuals ($b = 0.256$, $p$-value $= 4.83e{-}102$) as well as a stronger association in statin users due to the bias that is proportional to $\beta_0$ ($b = 0.409$, $p$-value $= 1.07e{-}39$). Since this SNP is strongly associated with baseline levels, its effect on drug response is overestimated when adjusting for baseline and a spurious association is observed for longitudinal change without lipid-lowering treatment. The *SLCO1B1* missense variant rs4149056 was not associated with total cholesterol baseline levels ($p$-value $= 0.078$), and genetic effects remained similar between baseline adjusted and unadjusted results (adjusted: $\gamma_D = 0.092$, $p$-value $= 3.16e{-}08$; unadjusted: $\gamma_D = 0.092$, $p$-value $= 1.18e{-}07$ Fig. 3c), evidencing the sole implication of *SLCO1B1* in pharmacokinetics (no significant association was found for longitudinal change either; $p$-value $> 0.24$). In contrast, the SNP rs11076175 in the *CETP* locus is strongly associated with HDL-C baseline levels ($\beta_0 = 0.262$, $p$-value $= 4.26e{-}311$), but had no significant pharmacogenetic effect in the unbiased model ($\gamma_D = 0.006$, $p$-value $= 0.70$). Upon baseline adjustment, strong associations with both longitudinal change and drug response were observed ($p$-values of 4.85e

−33 and 7.95e−06, respectively; Fig. 3d). Together, these examples illustrate how the genetic component of baseline levels can result in the identification of false positive associations and/or overestimation of pharmacogenetic effects.

More generally, no genome-wide significant associations were found in (the correct) longitudinal biomarker progression GWAS in medication-naive individuals (Supplementary Figs. 13, 14). However, upon baseline adjustment, striking similarities could be observed between drug response and longitudinal change GWAS (Supplementary Figs. 15-17; Supplementary Data 14, 15). Additional genome-wide significant loci found in the baseline-adjusted PGx GWAS included the *SORT1/CELSR2/PSRC1* locus in the LDL-C response as well as *CETP* in the HDL-C response to statin GWAS both of which also reached genome-wide significance in the longitudinal change GWAS in drug-naive participants.

## Polygenic risk scores as predictors of drug response

We assessed whether high PRS of the underlying biomarker contribute to increased or decreased biomarker reductions in medication users of the UKBB. High LDL-C PRS resulted in an increased absolute, albeit lower relative LDL-C reduction following statin treatment ($b_{\mathrm{abs}} = -0.092$ mmol/L/SD PRS, $p$-value $= 5.84e{-}48$ and $b_{\mathrm{rel}} = 2.47\%$/SD PRS, $p$-value $= 4.22e{-}34$; Fig. 4a; Supplementary Data 16). Thus, individuals with a higher genetic predisposition to elevated LDL-C levels are more likely to experience a larger drop, however, relative to their starting level this change is smaller than in those with a lower genetic predisposition. These opposing effects of high PRS on absolute and relative drug efficacy were also reflected in the genetic correlations of drug response traits with baseline traits. While the absolute LDL-C genetic difference was negatively correlated to LDL-C baseline levels ($r_g = -1.14$, 95%CI = [−1.54, −0.74]), the point estimate of the genetic correlation with the relative LDL-C difference was positive, although not significant ($r_g = 0.146$, 95%CI = [−0.10, 0.39]; Supplementary Data 17). These analyses suggest that in the case of statin response and LDL-C as readout, the absolute LDL-C change is closer linked to the baseline LDL genetics. Association results between TC PRS and TC response to statins were highly significant ($p$-value $< 2.51e{-}19$) and directionally concordant with LDL-C results. Nominally significant results between biomarker PRS and drug response phenotypes were found for high HbA1c PRS decreasing relative change following metformin treatment ($b_{\mathrm{rel}} = 1.00\%$/SD PRS, $p$-value $= 4.43e{-}03$) and high SBP PRS increasing SBP reduction following ACEi and treatment to all antihypertensives combined (ACEi: $b_{\mathrm{abs}} = -0.56$ mmHg/SD PRS, $p$-value $= 0.014$, all: $b_{\mathrm{abs}} = -0.46$ mmHg/SD PRS, $p$-value $= 0.011$). No significant effects of PRS on drug response were observed for the remaining antihypertensives, nor for beta blockers on HR (complete results in Supplementary Data 16). As for genetic association analyses, care has to be taken not to adjust for baseline levels as this can reverse directionality due to PRS affecting both baseline and biomarker reduction: adjusting the LDL-C

**Table 2 | Genetic predictors of cardiometabolic drug response reported in the literature that were discovered or reproduced via genome-wide association study and passed a genome-wide significance threshold of 5e−8**

| Medication | Phenotype | Gene | SNP | p-value literature | N literature | Reference | p-value, baseline adj. (UKBB) | p-value, post-base (UKBB) | p-value, log(post/base) (UKBB) | p-value, baseline adj. (AoU) | p-value, post-base (AoU) | p-value, log(post/base) (AoU) |
|---|---|---|---|---|---|---|---|---|---|---|---|---|
| Rosuvastatin | LDL-C | ABCG2 | rs2199936 | 2.1E-12 | 3523 | 20 | 0.62[a,b] | 0.77[a,b] | 0.89[a,b] | 0.16[b] | 0.33[b] | 0.17[b] |
| Statins | LDL-C | APOE | rs445925 | 8.52E-29* | 17,522 | 4 | 6.37E-26 (8.13E-46[c]) | 0.59 (1.10E-02[c]) | 8.48E-19 (1.11E-28[c]) | 9.58E-08 (4.00E-24[c]) | 7.03E-02 (2.20E-03[c]) | 2.27E-06 (3.56E-16[c]) |
| | LDL-C | LPA | rs10455872 | 7.41E-44* | 31,056 | 4 | 7.37E-19 | 6.97E-04 | 4.87E-17 | 8.90E-04 | 0.12 | 1.52E-03 |
| | LDL-C | SORT1 | rs646776 | 1.05E-09* | 38,599 | 4 | 9.77E-12 | 1.57E-02 | 2.15E-05 | 6.47E-11 | 1.30E-02 | 4.36E-07 |
| | LDL-C | SLCO1B1 | rs2900478 | 1.22E-09* | 24,253 | 4 | 1.84E-03 (1.89E-09[d,e]) | 3.07E-02 (2.22E-05[d,e]) | 9.76E-04 (1.10E-08[d,e]) | 2.83E-02 (2.51E-02[d]) | 7.48E-02 (2.84E-02[d]) | 2.13E-02 (1.31E-02[d]) |
| | LDL-C | APOB | rs1713222 | 4.68E-08* | 34,874 | 6 | 2.96E-04 (0.41[f]) | 8.05E-04 (1.77E-08[f]) | 0.48 (6.01E-02[f]) | 6.00E-03 (1.61E-03[f]) | 0.97 (0.18[f]) | 5-85E-02 (7.40E-03[f]) |
| | LDL-C | LDLR | rs67337506 | 3.08E-08* | 34,874 | 6 | 2.37E-03 (5.18E-04[g]) | 3.24E-05 (1.23E-10[g]) | 0.83 (0.85[g]) | 0.60 (9.42E-04[g]) | 0.13 (0.90[g]) | 0.39 (8.93E-02[g]) |
| | HDL-C | CETP | rs247616 | 8.52E-13* | 27,720 | 23 | 3.63E-10 | 0.49 | 0.37 | 1.67E-06 | 0.48 | 0.83 |
| Metformin | HbA1c below 7% | ATM | rs11212617 | 2.9E-09* | 3920 | 24 | 0.48 | 0.69 | 0.64 | 0.48 | 0.84 | 0.91 |
| | HbA1c | SLC2A2 | rs8192675 | 6.6E-14 | 10,577 | 7 | 0.20 | 3.89E-02 | 4.07E-02 | 0.49 | 0.94 | 0.81 |
| | HbA1c | PRPF31 | rs254271 | 1.2E-08* | 8273 | 25 | 0.77 | 0.28 | 0.31 | 0.22 | 0.67 | 0.70 |

Corresponding significance levels were retrieved from the EHR-derived genetic analyses in the discovery (UK Biobank) and replication (All of Us) cohort. If the discovery analyses in the UK Biobank yielded a different lead SNP for the same locus, we report results for both SNPs. GWAS were performed using a linear additive model, with a two-sided test of association.

*baseline adj.* absolute biomarker difference adjusted for baseline, *post-base* absolute biomarker difference, *log(post/base)* logarithmic (relative) biomarker difference, *UKBB* UK Biobank, *AoU* All of Us.

*results for baseline adjusted analysis.

[a]results for LD-proxy rs45499402 ($r^2$ = 1).

[b]all statins combined.

[c]results for LD-proxy rs7412 ($r^2$ = 0.64).

[d]results for LD-proxy rs4149056 ($r^2$ = 0.87).

[e]results for total cholesterol.

[f]results for LD-proxy rs10199768 ($r^2$ = 0.14).

[g]results for LD-proxy rs118068660 ($r^2$ = 0.29).

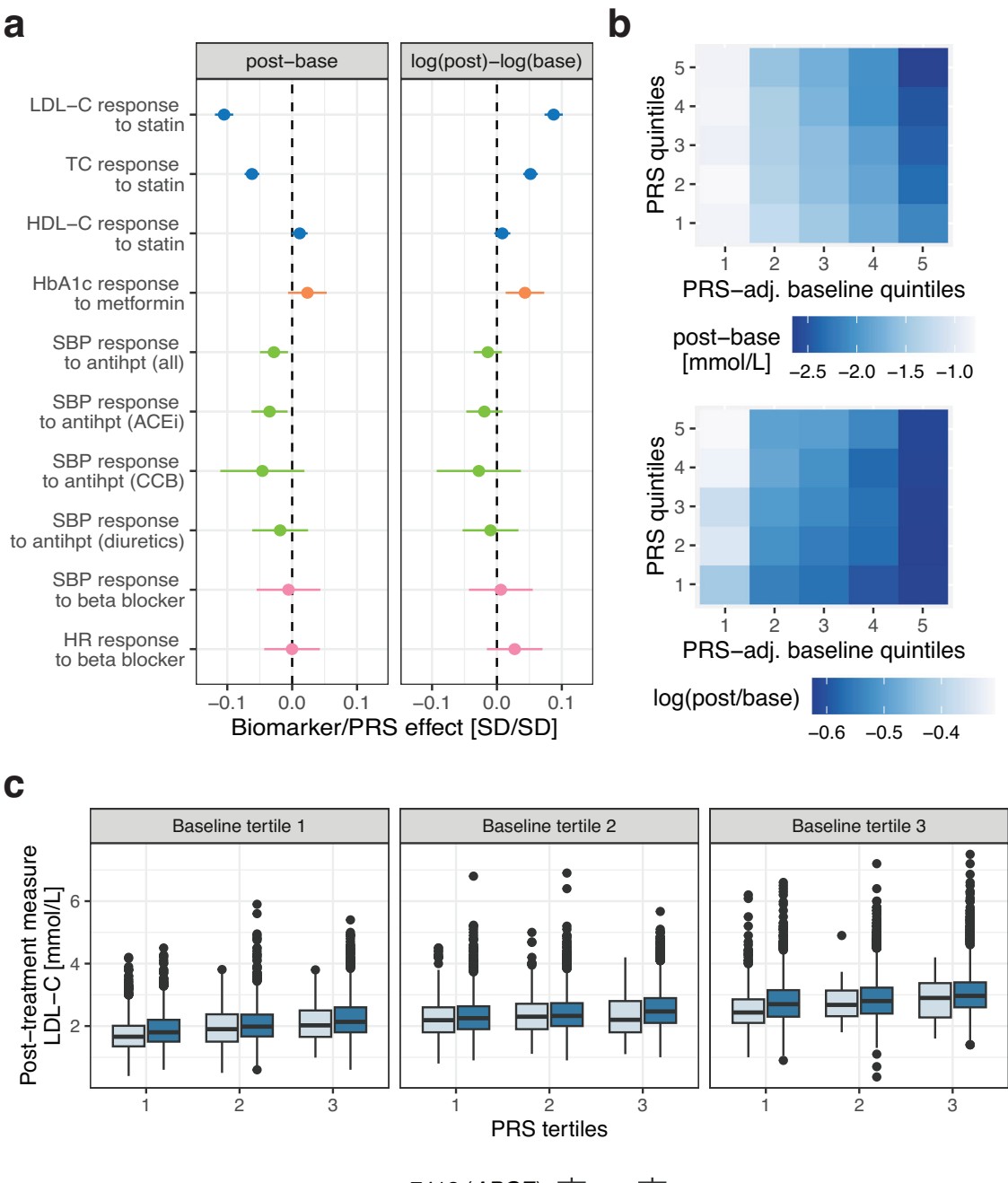

**Fig. 4 | Drug response phenotype associations with polygenic risk scores (PRS).**
**a** Drug response associations with PRS calculated for the absolute (post-base) and logarithmic relative (log(post)-log(base)) biomarker difference colour-coded by the drug (blue: statins, orange: metformin, green: antihypertensives, purple: beta blockers). Standardized effect sizes (biomarker/PRS effect) correspond to an SD change for 1SD increase in PRS. A negative sign means that increased PRS increases treatment efficacy (i.e., larger biomarker difference compared to low PRS). All associations are adjusted for sex, age and drug-specific covariates. The center of the error bars corresponds to the linear regression association estimate, and the error bars represent the 95% confidence intervals calculated in each drug response cohort (sample sizes in Supplementary Data 4). LDL cholesterol (LDL-C); total cholesterol (TC); HDL cholesterol (HDL-C); systolic blood pressure (SBP); heart rate

(HR); ACE-inhibitor (ACEi); calcium channel blocker (CCB). **b** Statin users stratified by 1) LDL-C baseline levels adjusted for LDL-C PRS and 2) LDL-C PRS quintiles with each tile showing the average LDL-C biomarker response (top: absolute, bottom: relative difference). Darker blue values correspond to stronger biomarker reductions. **c** Statin users stratified by 1) LDL-C baseline levels, 2) LDL-C PRS and 3) rs7412 genotype (individuals with the TT genotype are omitted as their sample size was too low). Boxes bound the 25th, 50th (median, centre), and the 75th quantile of LDL-C post-treatment measures. Whiskers range from minima (Q1 - 1.5*IQR) to maxima (Q3 + 1.5*IQR) with points above or below representing potential outliers. Numerical values and sample sizes in each stratum are shown in Supplementary Data 18.

reduction for LDL-C baseline levels suggests that high PRS decreases treatment efficacy ($b_{abs}$ = 0.14 mmol/L/SD PRS, $p$-value = 2.75e−141). As a control experiment, a high PRS led to a nominally significant increase in LDL-C in statin-free controls at the second measure ($b_{abs}$ = 0.006 mmol/L/SD PRS, $p$-value = 0.045), which upon baseline adjustment inflated to the same value as in statin users ($b_{abs}$ = 0.14 mmol/L/SD PRS, $p$-value < 1e−300).

Although PRS can serve as a predictor of drug response, starting baseline levels remain the best predictor of post-treatment levels (Fig. 4b, c). To disentangle the effect of genetics and environment on drug response, we adjusted baseline levels for PRS. Since baseline biomarker levels comprise both a genetic and environmental/lifestyle component (Fig. 3a), the resulting PRS-adjusted baseline levels should more closely reflect the environmental component only. As a consequence, individuals with high PRS and moderately high initial biomarker levels will have lower PRS-adjusted baseline levels than individuals with same initial biomarker levels and low PRS. PRS-adjusted baseline levels explained 43.0% and 10.7% of the variance of absolute and relative LDL-C difference, respectively. Increased LDL-C PRS increased LDL-C reduction which was most pronounced at high PRS-adjusted baseline levels (Fig. 4b). Conversely, high LDL-C PRS decreased relative LDL-C reduction which was more apparent at low PRS-adjusted baseline levels. By integrating the PRS, the explained variance increased from 43.0 to 44.3% and from 11.3 to 12.1% for absolute and relative LDL-C differences, respectively.

In Fig. 4c, we highlight how additional stratification by drug response genetic signals can improve prediction accuracy for post-treatment LDL-C levels following statin initiation (Supplementary Data 18). Additional stratification by the *APOE* genotype, the top signal in the LDL-C response GWAS, increased the explained variance of the relative reduction to 12.3% compared to 12.1% for adjusted baseline and LDL-C PRS predictors alone. Post-treatment levels remain higher for individuals with high PRS despite greater absolute LDL-C reductions.

## Discussion

In this study, we demonstrate the value of biobanks coupled with EHRs to study the genetics of cardiometabolic disease medications. We conducted discovery in the UKBB and replication analyses in the AoU, and assessed the impact of common and rare variations on drug efficacy. We show that signals from EHR-derived PGx GWAS are consistent with findings reported in the literature and present a theoretical framework to model drug response and longitudinal phenotypes.

Overall, we found only a few genetic variants to influence cardiometabolic drug response in line with other studies that often identified only a few or even no genome-wide significant signals[8,10,41]. A review on 76 drug efficacy GWAS reported that only 15% of drugs exhibit robust gene-treatment interactions[2], the extent of which largely depends on the drug's mode of action. In their review, the authors estimate that only 56% of these associations are clinically useful resulting in a probability of 8.1% that a drug has a clinically relevant genetic predictor[2]. Our study supports their conclusion that large pharmacogenetic effects are unlikely for the studied cardiometabolic drug traits, although more signals may emerge with larger sample sizes. While no single genetic variant will have strong predictive performance, well-powered PGx GWAS could enable the derivation of clinically useful PGx PRS. If we consider the LDL-C GWAS on unrelated UKBB participants of European ancestry (343,621 samples; http://www.nealelab.is/uk-biobank), 416 independent genome-wide significant hits are found. This number is estimated to decrease to 26 (6.3% of the hits) if the sample size were 28,800 which equals the sample size of the largest PGx GWAS in this study. With increased sample sizes, the number of PGx signals will also increase, however, low and often insignificant heritability estimates support that the genetic influence on the studied drug response phenotypes remains low (Supplementary Data 17). For LDL-C response to statins we estimated the genetic

heritability to be 3.4% (95% CI: [0%, 8.6%]) in line with an earlier study that reported a non-significant heritability estimate (11.7%, 95% CI: [0%, 29%],[5]). While inter-individual variability in drug response could also depend on rare variants, we only identified a single gene *GIMAP5* associated with HbA1c response to metformin.

Unlike RCT data, EHR data does not typically contain medication start along with baseline and post-treatment measures as direct read-outs and in this study, we propose analysis strategies to identify new drug users with corresponding biomarker measures. We tested two filtering options, a stringent and a lenient one, with the differences being the extension of the baseline and post-treatment time periods, the main medication acting as add-on therapy (e.g. metformin acting as add-on to sulfonylureas treatment), relaxed prescription regularity requirement and the allowance of medication dose changes. The most important difference between the stringent and lenient filtering settings was the resulting sample size which doubled for statin and tripled for antihypertensive cohorts when relaxing the filtering criteria (Supplementary Data 4). As a result of lower statistical power, the stringent filtering strategy only identified 4/10 PGx loci at a genome-wide significance level. We acknowledge that relaxed criteria, such as a longer time between the post-treatment measure and medication start, as well as less frequent prescriptions, were associated with smaller biomarker differences, suggesting that these variables are linked to drug adherence. Nonetheless, PGx signals identified in the lenient filtering scenario were confirmed as genuine, consistent with findings from the literature (e.g., *SLCO1B1*) and control analyses in statin-free individuals. Low drug adherence could potentially result in capturing the genetics of underlying health and behavioral aspects associated with adherence, although a previous study could not identify such signals even at a sample size of 116,439 for statins[42]. Furthermore, we tested whether there is a benefit in averaging over multiple baseline and post-treatment measures if available. Averaging over multiple measures should result in a more robust measure of the underlying biomarker level and reduce the chance of picking up regression-to-the-mean effects as we discuss in the following paragraph. The only difference we observed between these two strategies was that *SLCO1B1* reached genome-wide significance when averaging across multiple measures ($p$-value of 1.10e−08 compared to 1.49e−05). Taken together, we recommend averaging across multiple measures when available and adopting the stringent filtering criteria when using EHRs for drug response studies, as this approach more closely mimics RCT designs. However, if data availability limits the sample size, we recommend using more lenient criteria while incorporating parameters that best approximate drug adherence and account for concomitant medications as covariates, as this approach proved to significantly improve statistical power.

As similar study designs have used differing GWAS models to estimate pharmacogenetic effects, we developed a realistic model for biomarker differences and showed how baseline adjustment will induce biases for genetic variants associated with baseline levels. In recent years, there have been debates about baseline adjustment in analyses of change[6,43–45]. In the context of PGx GWAS, some researchers have advocated for no baseline adjustment[6], while others have argued otherwise[45]. There is a consensus that measurement errors can introduce a bias and spurious associations with longitudinal change upon baseline adjustment (regression-to-the-mean bias)[6,43–47], however, it is unclear whether baseline adjustment also affects drug response phenotypes since drug effects are likely larger than measurement errors. In the context of pharmacogenetic interactions with biomarker change, only the direct genetic effect on change is of interest, but genetic variants likely also affects baseline values which in turn impact the magnitude of change induced by the medication. Zhang et al. (2022) recommend adjusting for baseline as their model represents baseline as a mediator between genetic variants and quantitative change[45]. We argue that the mediator model is

inappropriate because it assumes that the change depends equally (i.e., same regression coefficient) on the genetic and the environmental component of the baseline value. However, their model does not test for this gene-environment "equivalence" and in all applications we examined, this is not the case: the change depends very little on the baseline genetics and almost all its association is with the environmental component of the baseline value. Hence adjusting for the baseline value rather than eliminating a bias, it introduces one. For this reason, we, instead, propose modeling drug response as the biomarker difference between two time points, as baseline genetics, which should remain constant during the post-treatment period, cancels out. We provide theoretical derivations as to the origin and magnitude of the bias arising from baseline adjustment and demonstrate in GWAS on drug-naive control individuals that our model, which supports not adjusting for baseline, does not generate inflated type I errors as suggested by the authors[45]. Furthermore, since control individuals are likely subjected to similar measurement errors and gene-environment interactions as medication users, the control experiments reassure that these effects are negligible compared to the pharmacogenetic effects. We show that for unbiased pharmacogenetic effect estimation, both absolute and logarithmic relative biomarker differences can be assessed. Whether absolute or relative reductions are most determining in lowering the risk of associated clinical diseases may be dependent on the studied medication-biomarker pair. If biomarker levels are linearly associated with the disease risk, absolute change is more relevant; however, if this relationship is exponential or quadratic, then relative change is of more importance. As to pharmacogenetic interactions, it is a priori unknown on which scale genetic variants act linearly, evidencing the usefulness of testing both absolute and relative biomarker changes. Regarding heritability estimates, there were no significant differences between the two phenotype definitions in this study (Supplementary Data 17). Thus, we recommend conducting discovery analyses on both scales, and in a next step, examine whether the observed associations on the respective scales are clinically relevant.

We compared the PGx effects reported in the literature to the ones obtained from the EHR study design to assess the value of using EHRs to derive drug response phenotypes. Previous studies have inconsistently conducted pharmacogenetic GWAS through baseline-adjusted and unadjusted models, and therefore, we conducted replication results for both models as well. For statins, seven out of the eight reported PGx loci reached genome-wide significance in the UKBB analyses when looking at all association models and SNPs in LD. However, not all of these PGx loci can be qualified as genuine PGx loci depending on which model was used in the literature. We replicate APOE, LPA, and SLCO1B1 as genuine PGx loci, while SORT1 and CETP only reached genome-wide significance in baseline-adjusted models (both in the drug response and longitudinal change GWAS in controls, Supplementary Data 14, 15) and likely do not harbour a drug-specific genetic component as reported earlier[6]. SNPs in the APOB and LDLR loci have been reported in the literature as spurious hits due to baseline adjustment[6]. However, for both loci, we identified SNPs in LD as genuine PGx signals that were significant in the absolute biomarker change model (Table 1). Although we did not identify PGx loci beyond what has been previously reported, the comparison with the literature confirmed that EHRs serve as a valuable data resource next to RCT and observational data for PGx study designs. Importantly, with the median cost estimated to be 409 USD per RCT participant[48], our study on 41,732 UKBB participants linked to their EHRs would cost 17 million, evidencing the cost effectiveness of this resource to study PGx at scale.

We found that high PRS of the underlying biomarker can lead to increased absolute, although lower relative biomarker reductions, further contributing to the growing body of literature that links PRS to treatment effectiveness[17,49–56]. A recent study showed that sulfonylureas therapy was more effective in participants with higher T2D PRS

with findings replicated in a separate cohort[54]. Other studies found that high schizophrenia PRS reduced antipsychotic efficacy[51] and similarly high LDL-C and SBP PRS were associated with uncontrolled hypercholesterolaemia and hypertension, respectively[55]. Additionally, high SBP PRS have been linked to lower rates of discontinuing hypertension medication[17,56]. Using RCT data, high coronary heart disease (CHD) genetic risk was found to be associated with more increased CHD risk over time, although the comparison between controls and treated participants revealed that relative risk reductions were higher among treated individuals with a high PRS, suggesting that this group benefited the most from lipid-lowering therapy[49,50,52,53]. Given the complexity of genetics affecting both baseline biomarker levels and disease risk as well as reductions thereof, disentangling whether genetic or environmental factors can be easier alleviated by medication requires careful considerations as adjusting for baseline levels induces genetic biases. While we found strong evidence for a higher genetic burden to increase low-density and total cholesterol reductions, this effect artificially reverses when adjusting for baseline levels. While RCT data with a control arm remains the gold standard for studying such complex interactions, large biobank data also allow the construction of (non-randomized) control groups. Both for variant-level and PRS association analyses, we demonstrate how drug response and disease progression genetics seemingly overlap when adjusting for baseline and how these baseline genetics signals disappear in the control group upon applying the correct longitudinal change model.

The reason as to why PRS contribute to drug-induced biomarker response requires further investigation and we propose two hypotheses: one related to biology and the other to either of the employed models. Biological pathways involved in the drug perturbation may coincide with genes highly contributing to PRS which means that the drug may work better for people with high baseline value for genetic reasons. Therefore, drugs designed to target key genes involved in the disease are likely to be more beneficial for individuals for whom these genetic networks are at the root of the disease. The second, more likely hypothesis concerns a model misspecification. In the current PRS association model, we assume baseline levels to be linearly associated with the biomarker difference and analogous to genetic variants, baseline levels encompass PRS as the genetic component. Thus, when adjusting the biomarker difference for both baseline and PRS, spurious associations are obtained with PRS and we show on the example of LDL response to statin, that the association is in the opposite direction than without baseline adjustment. If biomarker differences truly depend on the genetic and non-genetic components of baseline values, baseline PRS and the non-genetic baseline part are expected to correlate differently with baseline and biomarker change. Further investigations into deriving an environmental baseline component (either by approximating it by the residuals after PRS regression or explicitly identifying environmental correlates of baseline values), could provide more precise estimates on the relative contribution of genetic and environmental components on biomarker change. While we identify PRS as predictors of treatment efficacy, we also note that in our analyses PRS explained less than 2% of the variance in differential drug response.

Our study has several limitations. First, we rely on data from EHRs to derive before and after treatment biomarker levels, and thus cannot exclude the possibility that individuals were already on medication before the first recorded prescription. If these individuals were to make up a large proportion of the cohort, the analysis would no longer capture the genetics of drug response, but rather of disease progression whilst on medication. Second, despite a large fraction of individuals with medication records in the biobanks, final PGx cohort sample sizes are limited by the number of participants on a certain medication and further reduced due to incomplete or missing data. Of the ~ 65,000 participants with a statin prescription in the UKBB primary care records, 63% could not be considered for the LDL-C response analysis because of missing baseline and/or post-treatment measures. With an optimistic

outlook, sample sizes can be expected to double in the future when all UKBB participants are linked to their primary care records (this number currently stands at 45%) and could increase even further with longer follow-up times and an ultimate increase in cardiometabolic medication users. Third, and related to the previous limitation, most participants have follow-up measures after six months (Supplementary Fig. 5; Supplementary Data 4) which limits the study of immediate pharmacological effects, and measured drug responses may harbour a disease progression component. Frequency of follow-up measures constitutes a major limitation compared to RCT data where more frequent and broad biomarker measures are likely available. Forth, polypharmacy has only been taken into account within and not across medication groups, meaning that a statin user was excluded from the analysis if, for instance, fenofibrate (lipid-lowering medication) was initiated during the follow-up period while exclusion did not apply in cases where metformin (anti-diabetic medication) prescriptions were recorded during follow-up for that same individual. Even within, especially for antihypertensives where frequent changes in medication regimen occur, it can be difficult to determine appropriate filtering and covariate strategies to study individual drug classes as sample sizes are too low when restricting the analysis to individuals taking antihypertensives from a single class (i.e., stringent filtering strategy). Fifth, our analysis focused on continuous biomarkers and not on clinical events. LDL-C, SBP, HR and HBA1c merely serve as surrogate endpoints of CHD and T2D events, and the genetic interplay with drug efficacy may be different when assessing hard clinical endpoints. EHRs provide a rich resource for conducting analyses similar to those done with RCT data[49,50,52,53], and for studying genetic interactions with drug efficacy on clinical events in real-world settings. Sixth, analyses have been conducted in individuals of European ancestry. While allele frequencies of identified PGx signals were largely consistent across genetic ancestry groups (Supplementary Data 10), analyses in diverse ancestry groups are likely to identify PGx effects missed in individuals of European ancestry. For a PGx signal with modest effect size such as *SLCO1B1*, we estimated that a sample size of ~150,000 is required for 80% power in African/African Americans (MAF of 3.1% compared to 15% in Europeans), showcasing that even with cohorts the size of the current study increased numbers of PGx signals could be identified provided that the MAF is high. Finally, we rely on observational data to draw conclusions about drug efficacy. Although, we contrast the results with control analyses on longitudinal biomarker change, control and medication groups were not defined randomly and by definition have markedly different disease profiles. Better matching of controls through methods such as propensity scores that include genetic predisposition could address this issue in future studies.

To conclude, we show that EHRs enable new opportunities to study the genetics of drug response at scale and in a cost-effective manner. Although challenges remain with respect to the completeness and frequency of medication prescriptions and biomarker measures, these data allow us to shed light on the complex contribution of genetic and environmental components to drug efficacy. We find that the influence of common and rare genetic variants on drug response is relatively low, and larger sample sizes achieved by combining drug response GWAS from observational, EHR and RCT data as well as studies in more diverse ancestries will be needed to capture the full extent.

## Methods

### Study population

The UK Biobank is a prospective study of ~500,000 participants of whom 45% (*N* ≈ 230,000) are linked to the primary care data of the United Kingdom's National Health System[18]. The primary care resource contains longitudinal data of GP prescription records (datafield #42039) and GP clinical event records (datafield #42040) encoded through the British National Formulary (BNF), National Health Service

(NHS) dictionary of medicines and devices (DM+D), Read V2 and Clinical Terms Version 3 (CTV3) codes and are available up to 2016 or 2017, depending on the data provider (EMIS/Vision for Scotland and Wales, and TPP and Vision for England; detailed description of the linked primary care data is provided by the UKBB at https://biobank.ndph.ox.ac.uk/showcase/showcase/docs/primary_care_data.pdf). Analyses were conducted in individuals of white British ancestry (*N* = 190,754), with no excessive number of relatives (*N* = 76) and differing reported and genetically inferred gender (*N* = 133; we used the definition of white British ancestry, excess relatives, differing gender as defined by the UKBB Sample-QC file of datafield #531[18]), excluding participants who have withdrawn their consent up to April 2023 resulting in *N* = 190,545 participants who were considered in the drug response analyses.

### Study design and drug response phenotypes

We derived drug response phenotypes for the following cardiometabolic medication-phenotype pairs: statin-lipids (LDL-C, HDL-C, TC), metformin-HbA1c, antihypertensive-SBP (by antihypertensive class and all classes combined), beta blocker-SBP and beta blocker-HR. For each drug response phenotype, we considered stringent and lenient filtering scenarios which differed by regularity in prescription pattern, pre-treatment and post-treatment time windows as well as handling of treatment changes (e.g. dose change) and concomitant medication (e.g. add-on therapy, see below). In Supplementary Fig. 1 and Supplementary Data 3, we outline the different QC filters applied to each scenario. To further increase the number of available clinical measures, we added measures from the initial and repeated assessment visits with their respective time stamps to the pool of longitudinal data (LDL-C: #30780, HDL-C: #30760, TC: #30690, HbA1c: #30750, SBP: #4080, HR: #102). Read V2 and CTV3 codes encoding these variables in the primary care data are listed in Supplementary Data 1 (Supplementary Note 2 for HbA1c unit conversion). Baseline measures were taken three months (stringent filtering) or up to a year (lenient filtering) before treatment initiation and 7 days after, either as the closest measure to treatment start or an average of all available measures during the pre-treatment period. The post-treatment period was defined as 60 days for antihypertensives and beta-blockers, and 100 days for statins and metformin after medication start (statins and metformin have delayed effects on lipids[32] and HbA1c[33], respectively), up to 1 (stringent) and 2 (lenient) years after, and either the closest measure to treatment start or an average of all available measures during the post-treatment period were taken. Consequently, we derived drug response phenotypes for four scenarios: stringent filtering-single measure, stringent filtering-average measures, lenient filtering-single measure, and lenient filtering-average measures.

To determine medication regimens (medication start, treatment changes, prescription regularity), we first extracted all available prescriptions for each broader medication class (lipid-regulating, antidiabetic including insulin, and antihypertensives; BNF and Read V2 codes in Supplementary Data 2; when BNF codes were truncated to miss the drug ingredient, we extracted them by matching drug names and brand names in the drug description). We then selected individuals with entries of the medication of interest (primary medication) and omitted individuals taking medications other than the primary medication of the same class within a year of initiating the primary medication (in the stringent filtering setting, only monotherapies within the same medication class were allowed and no prior related medication prescriptions were permitted). This criterium was relaxed in the lenient filtering setting where the primary medication could act as add-on therapy in certain scenarios, with the related medication included as a covariate (Supplementary Data 3). More specifically, the following concomitant medication regimens were allowed in the lenient filtering setting:

1. Statins: Non-statin antilipemic medication prescriptions (e.g. fenofibrates) prior to statin prescriptions were permitted and use was included as covariate.
2. Metformin: Prior sulfonylureas prescriptions were permitted (covariate) as well as metformin acting as add-on to sulfonylureas (additional covariate, which was defined by recorded sulfonylureas prescriptions during baseline and post-treatment periods). Individuals with sulfonylureas prescriptions recorded only during the post-treatment period (and not baseline period) were filtered out.
3. First-line antihypertensives: Antihypertensives were allowed to act as add-on to beta blockers (covariate) and loop diuretics (additional covariate) if prescriptions were consistently recorded during baseline and post-treatment periods as illustrated in Supplementary Fig. 2.
4. Beta blockers: in the analysis with SBP we excluded individuals taking any other antihypertensives except for loop diuretics (covariate, same as described with first-line antihypertensives). In both the stringent and lenient analysis with HR, prescriptions of antihypertensives other than beta blockers were permitted since their effect on HR is much weaker than for beta blockers[57].

Note that in both stringent and lenient filtering settings, individuals taking primary medications in combination with a medication of the same class (e.g. statins in combination with ezetimibe) were filtered out (Supplementary Note 3). Prescriptions during the post-treatment period of medications of the same class as the primary medication (except for the allowed concomitant medication regimens in the lenient filtering setting described above) were defined as treatment changes, and accordingly, individuals were excluded from the respective drug cohort. A dose change, with dose information retrieved from the drug description using regular expressions[15], was defined as treatment change in the stringent, but not in the lenient filtering setting where the average over all prescriptions in the post-treatment period was used. Note that dosage information is not readily available, and we assume the dose information, available in the description of the medication, to correspond to the daily dosage. Since a given primary medication as defined herein can comprise multiple drugs (e.g. simvastatin and atorvastatin for statins, ramipril and lisinopril for ACEi), we only included drugs taken by at least 20 individuals (both in the stringent and lenient filtering setting). Finally, we defined a prescription regularity parameter by the presence of a prescription at least every two months for the duration of the post-treatment period with the completeness being 100% if this was the case (we required a completeness of 60% and 30% in the stringent and lenient filtering setting, respectively). In Supplementary Data 5, we show for each drug response cohort the number of individuals per drug type and dose which constituted covariates in all drug response analyses.

In Supplementary Data 4, we show the study characteristics of the individuals in each drug response phenotype cohort. Furthermore, bar plots in Supplementary Fig. 3 show the number of individuals after each QC step. The aforementioned QC steps can be summarized as follows: (i) available baseline and post-treatment measures, (ii) presence of a primary care record other than baseline/primary medication at least two years before the medication start to avoid falsely considering a change to a new health care provider as a first prescription, (iii) presence of a prescription part of the broader medication class after post-treatment measure, (iv) drug change between medication start and post-treatment measure, (v) regular prescriptions proxying drug adherence under the assumption that skipped prescriptions are the consequence of inconsistent medication intake and (vi) minimum baseline level (e.g. LDL-C $\geq$ 2 mmol/L). We only considered cohorts with more than 500 individuals for GWAS analyses.

## GWAS

In the genetic association analyses, we define the drug response phenotype as either the absolute ($Y_{t_1} - Y_{t_0}$) or the logarithmic relative ($\log(Y_{t_1}) - \log(Y_{t_0}) = \log(Y_{t_1}/Y_{t_0})$) difference between post-treatment $Y_{t_1}$ and baseline $Y_{t_0}$ levels. This difference was then adjusted for study-specific covariates including sex, age at the time of medication start, time between medication start and post-treatment measure, drug type and dose if applicable and the first 20 principal components (Supplementary Data 3). Importantly, the difference was not adjusted for baseline levels as this can induce a bias for genetic variants that are associated with baseline (Supplementary Note 1).

GWAS analyses were conducted using REGENIE (v3.2.6) which accounts for sample relatedness[58]. REGENIE first fits a whole-genome regression model (step 1) before testing each SNP in a leave-one-chromosome-out (LOCO) scheme (step 2). In step 1, genotyped SNPs were filtered as follows using PLINK2[59]: minor allele frequency (MAF) $\geq$ 0.01, Hardy-Weinberg equilibrium $p$-value $\geq$ 1e−15, genotyping rate $\geq$ 0.99, not present in high linkage disequilibrium (LD) regions[60], not involved in inter-chromosomal LD[58] and passing LD pruning at $r^2 < 0.9$ with a window size of 1,000 markers and a step size of 100 markers which resulted in 424,544 SNPs included in step 1. In step 2, variants imputed by the Haplotype Reference Consortium panel with a MAF $\geq$ 0.05 were tested (up to 5.5 million markers depending on phenotype sample size). Individuals with missing genetic data and/or not passing genetic QC were excluded from the analysis. Independent signals were defined as $r^2 < 0.001$ and clumping was performed using PLINK and the UK10K reference panel[61].

## Rare variant analysis

Rare variant analyses were conducted using REGENIE (v3.2.9). Phenotype definitions and covariates (Supplementary Data 3) were the same as in the GWAS analyses, except that biomarker differences were transformed by inverse quantile normalization to decrease the chance of false positives. Following step 1 whole genome regression (Method section: GWAS), we performed rare variant burden tests using optimal kernel association tests (SKATO) in step 2[38]. Masks were constructed from rare variants (MAF < 0.01) including missense and putative LoF variants, and REGENIE SKATO tests were computed with default parameters. Variant annotations and gene set definitions were derived following the original quality functionally equivalent (OQFE) protocol and provided on the UK Biobank DNAnexus research analysis platform[62]. Burden tests were then conducted on OQFE WES data (#23158)[62]. Genes classified as very important pharmacogenes (VIPs) were downloaded from the PharmGKB gene annotations (April 5, 2023 version)[40].

## Replication in the All of Us Biobank

The All of Us research program is a prospective cohort recruiting up to 1 million participants[19]. Replication analyses were conducted in the release v7 in which genotype data were available for ~310,000 and WGS data for ~250,000 individuals. In the AoU database, the Observational Medical Outcomes Partnership (OMOP) Common Data Model (CDM) is used for standardized vocabularies and harmonized data representations. Medication records were retrieved based on concept ID codes from the RxNorm vocabulary and phenotypes from the SNOMED vocabulary. Replication analyses were restricted to lipid response to statins and HbA1c response to metformin for which genome-wide significant signals were obtained either in the UKBB analyses or reported in the literature.

Similarly to the UKBB, we extracted medication records by starting from the broader medication class (lipid modifying agents (concept id 21601853) and drugs used in diabetes (concept id 21600712)) which were then classified into primary medications (statins (concept id 21601855) and metformin (concept id

1503297)), combination therapies (lipid modifying agents, combinations (concept id 21601898), blood glucose lowering drugs, combination (concept id 21600765) and sulfonylureas (concept id 21600749)) and related medication from the same class. Dose information was extracted from the drug concept entries using regular expression or imputed by the median dose of the drug in question when not available. Phenotypes were extracted based on the following ancestor concept IDs: LDL-C (3028437), HDL-C (3007070), TC (3027114) and HbA1c (3004410). Only measures with available units and values in the plausible range were retained (Supplementary Data 1). While lipid measures were recorded as mmol/L and primarily as mmol/mol for HbA1c in the UKBB (Supplementary Note 2), units were mg/dL and % for lipids and HbA1c, respectively, which we left unconverted. Following the extraction of longitudinal medication and biomarker measures, we followed the same QC steps as in the UKBB by applying the lenient filtering strategy with average baseline and post-treatment measures (Supplementary Fig. 1). Drug prescription regularity was found to be lower in the AoU, likely because drug prescriptions are only recorded from participating EHR sites. As a consequence, we lowered the drug regularity QC parameter and required a single prescription between medication start and post-treatment measures (QC9, Supplementary Fig. 1). Cohort characteristics and reason for removal are reported in Supplementary Data 9 and Supplementary Fig. 10, respectively.

GWAS: GWAS analyses were conducted using REGENIE (v3.2.4). For step 1, we used genotyped SNPs and filtered them as follows using PLINK2: autosomal SNPs, MAF ≥ 0.01, Hardy-Weinberg equilibrium $p$-value ≥ 1e−15, genotyping rate ≥ 0.99, not present in high linkage disequilibrium (LD) regions[60] and passing LD pruning at $r^2 < 0.9$ with a window size of 1000 markers and a step size of 100 markers which resulted in 238,888 SNPs. The first 20 PCs were computed on the same set of SNPs using the FastPCA algorithm implemented in PLINK2[63]. In step 2, we used WGS data from the Allele Count/Allele Frequency (ACAF) threshold callset to test associations between the genotypes of interest and drug response phenotypes.

Rare variant analysis: We conducted SKATO analyses on rare variants from the exon regions using REGENIE (v3.2.4) with step 1 being the same as in the GWAS. Variant annotations and gene set definitions were extracted from the Variant Annotation Table (VAT) provided by the AoU. Missense variants and putative LoF variants defined as stop-gain, frameshift, splice donor and splice acceptor with MAF < 0.01 were included in the burden tests.

## PRS and genetic correlations

We calculated PRS for UKBB participants with the PGS Catalog Calculator[64,65] using pre-calculated genetic effect sizes from the PGS Catalog[66]: LDL-C, PGS002150; HDL-C, PGS002172; TC, PGS002108; HbA1c, PGS002171; SBP, PGS002228; HR, PGS002193. These PRS genetic effects stem from the LDpred2-auto method (implemented in the R package bigsnpr) which assumes a point-normal mixture distribution for effect sizes, with only a proportion of causal variants contributing to the SNP heritability[66,67]. We then calculated the associations between PRS and the absolute and (logarithmic) relative biomarker difference adjusted for sex, age and drug-specific covariates (Supplementary Data 3).

We calculated genetic correlations between traits using the GenomicSEM R package (v0.0.5c)[68]. Trait GWAS summary statistics were obtained from the following consortia: LDL-C, HDL-C and TC from the Global Lipids Genetics Consortium[69] ($N$ up to 1,320,016; European ancestry), HbA1c from the UKBB (#30750, $N$ = 344,182), SBP from a meta-analysis of the UKBB and the International Consortium of Blood Pressure[70] ($N$ up to 757,601) and HR from the UKBB (#102, $N$ = 340,162) where the UKBB GWAS summary statistics came from Neale's lab (http://www.nealelab.is/uk-biobank).

## Longitudinal biomarker change GWAS in controls

We conducted biomarker change GWAS in control individuals that were part of the primary care data and that did not have any drug prescription indicated for the investigated disease/surrogate endpoint: controls in the lipid-change GWAS had no lipid-lowering medications, those in HbA1c-change GWAS had no antidiabetic medications, those in blood pressure-change GWAS had no antihypertensives, and those in heart rate-change GWAS had no beta-blockers (i.e., broad medication class, Supplementary Data 2). All participants in this set with two available measures spaced between 6 months and 3 years which corresponds to the allowed time interval between baseline and post-treatment measures were included. GWAS analyses were conducted analogous to the drug response GWAS, replacing baseline with first and post-treatment with second phenotype measure. We used the same covariates as in the corresponding drug response cohorts omitting drug-specific variables (Supplementary Data 3).

## Power analysis

We conducted statistical power analyses to determine the sample size needed to detect PGx signals at genome-wide significance in different ancestral groups. Minor allele frequencies were obtained from the gnomAD v4.1.0 resource[35]. We based the effect sizes on the effect sizes observed in the discovery analyses in the UKBB and calculated the required power as

$$\text{power}(\alpha, \lambda) = \Phi\left(\sqrt{\lambda} - c\right) + \Phi\left(-\sqrt{\lambda} - c\right)$$

where $\Phi()$ is the cumulative density function of the standard Gaussian distribution, and $c$ was set to $\Phi^{-1}(5e{-}08/2) = 5.45$ which is the test statistic corresponding to a two-sided discovery $\alpha$ of 5e−08.

The non-centrality parameter (NCP), $\lambda$, was calculated as

$$\lambda = \frac{N}{n_{obs}} \cdot \frac{\beta_{obs}^2}{se_{obs}^2} \cdot \frac{\text{MAF} \cdot (1 - \text{MAF})}{\text{MAF}_{obs} \cdot (1 - \text{MAF}_{obs})}$$

where the sample size $n_{obs}$, observed effect size $\beta_{obs}$, standard error $se_{obs}$ and minor allele frequency $\text{MAF}_{obs}$ stem from the discovery analyses in the UKBB.

## Data availability

Genetic and phenotypic data from the UK Biobank and the All of Us Research Program, including all linked electronic health records, are available to bona fide researchers upon application. Application details for the UK Biobank are available at http://www.ukbiobank.ac.uk/using-the-resource/. Access to the controlled tier of the All of Us Research Program requires institutional approval, completion of ethics training, and agreement to a Data Use and Registration Agreement (DURA), as outlined at https://www.researchallofus.org/register/. British National Formulary (BNF), National Health Service (NHS) dictionary of medicines and devices (DM+D), Read V2 and Clinical Terms Version 3 (CTV3) vocabularies encoding the UK Biobank primary care records, https://biobank.ndph.ox.ac.uk/showcase/refer.cgi?id=592. ATHENA - OHDSI vocabulary repository for RxNorm (drug) and SNOMED (phenotype) concept IDs, https://athena.ohdsi.org/. Polygenic risk score genetic effect sizes (UK Biobank), https://www.pgscatalog.org/publication/PGP000263/. Lipid GWAS summary statistics from the Global Lipids Genetics Consortium, https://csg.sph.umich.edu/willer/public/glgc-lipids2021/. Systolic blood pressure GWAS summary statistics from the UKBB and the International Consortium of Blood Pressure meta-analysis, https://www.ebi.ac.uk/gwas/publications/30224653. Neale's lab GWAS summary statistics (UK Biobank), http://www.nealelab.is/uk-biobank. UK10K individual-level data are available upon request, https://www.uk10k.org/data_access.html. Pharmacogenetic GWAS summary statistics are available on the GWAS Catalog

(https://www.ebi.ac.uk/gwas) under the accession IDs GCST90455589-GCST90455608. Source data are provided with this paper.

## Code availability

GWAS calculations were performed with REGENIE (v3.2.6) which is available at https://github.com/rgcgithub/regenie. PLINK2 is available at https://www.cog-genomics.org/plink/2.0/. PRS were calculated with the PGS Catalog Calculator (v2.0) available at https://github.com/PGScatalog/pgsc_calc. Genetic correlations were calculated with the GenomicSEM R package (v0.0.5c) available at https://github.com/GenomicSEM/GenomicSEM. All codes used in this analysis are available on GitHub at https://github.com/masadler/PGxEHR (https://doi.org/10.5281/zenodo.14026836[71]).

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

## Acknowledgements

This work was supported by the Swiss National Science Foundation (310030_189147) to ZK. RBA is supported by NIH HG010615, NIH GM153195, Chan Zuckerberg Biohub, and Burroughs Wellcome Fund. MCS would like to thank the Fulbright Program for funding her research stay at Stanford University where this research work has started under the co-supervision of RBA. This research has been conducted using the UK Biobank Resource under Application Number 16389. LD was calculated based on the UK10K data resource (EGAD00001000740, EGAD00001000741). Computations were performed on the Urblauna cluster of the University of Lausanne. We also would like to acknowledge the participants and investigators of the UK Biobank and All of Us study. We thank Greg McInnes for his help and support in analyzing medication prescription data in the UK Biobank and Ewan Pearson for his advice in defining drug response phenotypes from the UK Biobank primary care data.

## Author contributions

MCS and ZK conceived and designed the study. MCS performed statistical analyses in the UK Biobank. AA conducted replication analyses in the All of Us research program under the supervision of MCS and RBA. CC computed PRS on UK Biobank participants. CA contributed with the biological interpretation of the results. DMR provided guidance on analyzing rare variants from sequencing data. ZK supervised all statistical analyses. All the authors contributed by providing advice on the interpretation of results. MCS and ZK drafted the manuscript. All authors read, approved, and provided feedback on the final manuscript.

## Competing interests

MCS has been consulting for 5 Prime Sciences at the time of the submission; however, this study was performed separately with no relationship to 5 Prime Sciences. The results and opinions expressed in this paper do not represent those of 5 Prime Sciences. The other authors declare that they have no competing interests.
