## [Transparent Peer Review file · Nature Communications]

Leveraging large-scale biobank EHRs to enhance pharmacogenetics of cardiometabolic disease medications

Corresponding Author: Dr Marie Sadler

Version 0:

Reviewer comments:

Reviewer #1

(Remarks to the Author)

Summary

The authors have performed drug response efficacy PGX analyses using UKB as discovery and AOU as replication, in order to demonstrate the utility of large EHR-based biobanks for PGX. First, the authors perform GWAS analyses for variant discovery, for several different cardiometabolic drug responses, e.g. cholesterol response to statins, HbA1c response to metformin, and BP/HR response to antihypertensives. They also perform PRS analyses to test the hypothesis that the PRS of the corresponding biomarker trait is associated with the drug response phenotype. And finally, they consider different statistical models, to give recommendations for unbiased study design for PGX. This is novel work, as explained nicely in the Intro, that PGX has not yet fully attempted to exploit the large-scale biobank data.

I particularly liked the following aspects of the manuscript:

- The opening paragraph of the Intro, setting the scene for the challenges of PGX, and the nice perspective with the GWAS catalog statistic.
- The creativity in the figures provided
- The comprehensive inclusion of many thorough sensitivity analyses, which address many controversies in the literature on PGX study design and statistical models. Especially the “modelling drug response and longitudinal change phenotypes” section, which is very convincing indeed.

Comments to Authors

Please could the authors address the following points:

- 1) The authors have covered so much within this one paper: the genetic discovery of the PGX-GWAS with novel discovery and comparison to literature findings; the PRS analyses testing a completely separate hypothesis; the statistical model methodology; lots of sensitivity analyses; the study design features of novel EHR-biobank-based PGX. Whilst all content makes a valuable contribution to the field, I almost wonder whether it may be too much for one single manuscript, and whether it could benefit from being split into different papers. This may need guidance from the Editors.
- 2) There are (at least) two recent, major references, in the field of statin-response PGX that the authors have omitted from their literature review, and missing completely from their references: (i) Oni-Orisan et al, Genomic Medicine 2020 (PMID: 31969989); (ii) Zhang et al, Genomic Medicine 2022 (PMID: 35680959). Please can the authors ensure their whole literature review is completely up-to-date, across all drug-response traits considered within this paper.
- 3) In particular, the Oni-Orisan et al study performs statin-response PGX GWAS using the GERA study, which is also based on EHR data, meaning that the use of EHR data from UKB is not entirely novel for the first time, for LDL-statin response PGX GWAS.
- 4) Importantly, the Oni-Orisan et al paper also reports the LDLR locus for LDL statin response from their baseline-adjusted GWAS analysis, meaning that 1 of the only 2 novel loci the authors report here, is actually not a novel finding. This is important to correct.
- 5) Important data availability consideration prior to publication: Should the actual drug-response phenotypes derived here from UKB also be returned to UKB, and made publicly available to other researchers?
- 6) There is a clear missing gap in this article, where, having reported two novel loci, the authors provide no discussion or information on the biology or hypothesised mechanism or known function of these genes.

7) With the final conclusion being that “larger sample sizes will be needed”, there seems to be a big missed opportunity from this work. Having used AOU data for replication, why have the authors not considered meta-analyses fully combining GWAS data together from both UKB and AOU, in order to achieve this? Similarly, having compared their UKB PGX GWAS findings to previously reported results from the literature, when some of these GWAS data must presumably be publicly available, why have the authors not considered combining their biobank PGX GWAS data together with the already existing RCT-based GWAS data via a full, large-scale meta-analysis? What would the recommended study design be, for combining EHR biobank based PGX GWAS with more traditional RCT-based PGX GWAS?

8) Lines 158-160: Even though no GW-significant variants have been reported in the literature from BP drug response PGX, I still think it would be important for the authors to consider some of the “peri-significant” results that have been reported in the literature, to compare what their exact equivalent results are from biobank PGX analyses. Please add these lookups to your paper.

9) The “Study population” section of the Methods chapter is not at all sufficient and detailed enough for publication, e.g: how were individuals of “white British ancestry” selected? What threshold was used for “no excessive number of relatives”? Up to which date were withdrawn-consent individuals applied? In general, please explain the breakdown of N throughout the QC process steps in more detail. Furthermore, what does “#531” correspond to? What does “depending on the data supplier” mean?

10) (line 346) I have one major concern with the time intervals used. Whilst I appreciate the need for trying either stringent vs lenient filtering, and the need to try to maximize N, I am concerned as to why the post-treatment period has to start at least 6 months after medication started? What is the rationale behind this? Would there be UKB individuals with a smaller post-treatment time interval than this? I believe that RCTs in the PGX field have shown that a shorter post-treatment follow-up time makes drug response more accurate. Similarly, the end-point of the post-treatment time period for the stringent filtering, of up to 1.5 years, already seems fairly lenient, compared to follow-up times from RCT data used for PGX in the literature. Please could the authors discuss how their time-windows compare to equivalent time-windows of RCTs within the PGX literature, to give a clearer idea of how different the EHR-biobank based PGX study design actually is.

11) Furthermore, these optimal time-windows may be both drug- and trait-specific. Please could the authors discuss this.

12) Additionally, referring to line 469, could the criteria for controls have been stricter? And if so, would this have a beneficial impact?

13) Due to line 261, it is important, please, for the authors to discuss why there are so few PGX signals in general, so the authors can give recommendations for future PGX studies.

14) The two different phenotype versions “absolute” and “relative” are defined within the Methods, and presented within Fig 1. But the motivation behind these two different phenotypes needs more explanation, as it is essential for readers to understand the differences before the later results sections. Furthermore, please add more discussion in the paper about which of these two phenotypes is better.

15) The final limitations section of the Discussion is very important for this work, and requires further expansion. For the 1st limitation of already being on medication, please discuss how this would impact the data, bias and findings; for the 2nd limitation of restricted sample size, please discuss if N is hoped to improve in UKB in the future or not, for example perhaps with more follow-up visits data; for the 3rd limitation of polypharmacy, please provide more clarity on what their strategy actually was; for the 4th limitation of not using clinical endpoints, please discuss if, and how, this could be done in the future, in either RCTs or in EHR-biobanks; for the final limitation could the control and medication group be “matched” in any way, or could disease-PRS stratification help to improve this?

16) Line 323 of the Methods is the only place that clearly states that only “45% (N~230,000)” of UKB are linked to primary care data. This is indeed a major limitation, which needs emphasizing and discussing more.

17) The Introduction ends positively showing the benefits of this work. However, it is clear from the final sections of the Discussion, that there are still many challenges & limitations of biobank-based PGX research. So I think the authors need to ensure there is still a correct balance of strengths vs limitations throughout the whole manuscript. It is helpful for the field, if the challenges that still remain for biobank-based PGX are discussed in detail.

18) Similarly, the authors don't seem to acknowledge that there may still be some benefits of RCTs for PGX, if large enough RCT sample sizes could be available. Provided that future researchers perform PGX analyses of RCTs using appropriate statistical models, please could the authors discuss what benefits RCTs may still have over biobanks. For example, if there existed a biobank-based dataset and an RCT-based dataset of equal N sample sizes, would the cleaner phenotyping and study design of the RCT actually still be preferable? There seems to be a balance of EHR-biobanks with larger N vs RCTs with smaller N.

19) Furthermore, the Discussion ends by stating that “larger sample sizes will be needed”, so please could the authors discuss whether, considering the limitations and challenges already outlined, whether these larger sample sizes will ever actually be realizable?

20) Having discussed the bias from baseline adjustment, please could the authors discuss whether this bias would be as strong in RCTs, as shown here in the biobank PGX work.

21) After considering both the lenient and stringent filtering scenarios, please could the authors discuss what their study design and QC recommendations would be for biobanks for future PGX researchers. Clearly it is a trade-off vs sample-size. But more discussions on specific recommendations from authors would be valuable for readers. For example, the fact that the APOB locus was only significant in the stringent filtering analysis rather than the lenient filtering analysis, I feel that this suggests that the stricter phenotyping pipeline may actually be optimal, given a large enough N...even though the authors here have to use the lenient filtering analysis as their primary analysis for reporting, due to limitations on N.

22) In the PRS results text, results are only stated within the text for SBP response to ACEi treatment. Please ensure that the authors have conducted analyses for BP response to all other drugs that they ran GWAS for, and that all analyses are commented on within the Results text.

23) Please clarify whether the PRS analyses are considered for both the absolute and relative types of phenotypes, for all drug-type PGS, e.g. in the examples from lines 225 onwards, which were only nominally significant. Furthermore, I felt here like the reader would benefit from a brief reminder of the 2 different types of phenotype “absolute” and “relative”, as lines 216-

224 are quite complex. Also, the results focuses on a comparison of whether with vs without baseline adjustment is better. But, with regards to other comparisons being made, it is not always clear which of the absolute vs relative phenotypes is being used in the analyses, within the Results text. Or perhaps the authors have not always performed analyses for both, in which case they should. For example, lines 225-236 gives a result for b_abs for the absolute phenotype, but never for b_rel for the relative phenotype.

24) Even though I commend the authors on the thorough sensitivity analyses, the Results text becomes very busy, so caution to the authors that there are so many different comparisons being considered within the paper.

25) Lines 222-223, please could the authors discuss an explanation as to why the genetic correlation is significant for the absolute phenotype, but not for the relative phenotype.

26) Line 382: The covariate for “time between medication start and post-treatment measure” would be interesting to discuss in more detail, because this is a covariate that would not be needed in most RCT-based PGX.

27) Line 193 suggests that there may be several different types of statins and doses. I couldn't find anywhere a clear description of what all these different types of statins and ranges of different doses are within the analysed data. If I have missed it, please refer to such a Sup Table of information here; or if not, please provide detailed information on this.

28) Lines 324-5: please provide more detail for the BP drugs. Linking to lines 345-7, please clarify more clearly if analyses are restricted to monotherapy.

29) Lines 354-6: please clarify if this is the case for both the lenient and stringent filtering criteria?

30) Lines 362-4, the distinction between these 2 different scenarios is not written clearly enough.

31) Lines 375-6 about “proxying drug adherence”: please explain more clearly what you mean here?

32) Lines 239-242: please explain this in more detail, so the concepts here are clearer.

33) Lines 275-280: Please give more examples to illustrate this further.

34) Line 272: should the word be “can” or “will” in the sentence “...showed how baseline adjustment can induce biases...”

35) Please clarify in the Results section, that the PRS analyses (I believe) are within UKB only, not investigated within AOU data too. (Lines 416-8 explained which analyses the replication analyses were restricted to, but it does not say if PRS analyses were validated in AOU too, although line 454 suggests not, only stating that PRS were calculated for UKBB participants...but this needs clarity...)

36) Line 282 “in line with other studies”, please provide the references.

37) Line 75 is written strangely, suggesting that HDL-C is an exception, with biomarker change levels in the opposite direction, when actually this is as expected and not of concern or surprising, because increased HDL-C is a good thing.

38) Lines 134-5: this should state “or both”, for example, APOE is discussed in an example later, suggesting that this does have both baseline genetic and PGX effects.

39) Please state in the Abstract: (i) that analyses are of EUR ancestry only; (ii) the separate N sample sizes for UKB vs AOU.

40) The “Furthermore” sentence towards the end of the abstract, describing the different results for “absolute” vs “relative” biomarker reduction, is quite complex, when stand-alone in the Abstract, without these different phenotypes being clearly defined.

41) In the opening sentences of the Results, please re-state, as from the Methods, the N of UKB subjects with primary care data available. (It is strange otherwise, at the bottom of the first Results pg, that N is provided for AOU replication data, but not yet for UKB.)

42) In Table 1, for clarity, please also name “UKB” and “AOU” in the column headers, as well as the “discovery” and “replication” labels.

43) Figure 3: The LH-side diagram in part (a) may be quite complex for some readers of this broad readership journal; the examples in Figs 3b-d would still benefit from more detailed explanations, beyond what is already provided on pg.15. I am not surprised that the Fig 3 legend is so long, when this is quite a complex Figure, with so many nested multi-panel graphs, overall showing so much.

44) Figure 4(a): Please label the two side-by-side plots are “absolute” vs “relative” phenotypes, i.e. always labelling the figures with the same consistency to help the reader. Please also explain the three different colours.

45) Line 467 has a typo within “in individuals part of the primary care data”

(Remarks on code availability)

Reviewer #2

(Remarks to the Author)

In this large-scale biobanks coupled with electronic health records (EHRs) pharmacogenetic study, authors analyzed both common and rare variants, providing a more comprehensive view of genetic influences on drug response. Through the analyses, they were able to replicate previous findings and also identify new signals such as APOB and variants associated with lipid response to statins near LDLR and ZNF800 genes. Two large biobaks UKBB and AOU were used for discovery and replication analyses which is thorough and the analyses conducted (GWAS, EXWAS and PRS) were conducted to support the conclusions. The authors have used appropriate statistical methods and have been transparent about the limitations of their study. However, some points that might benefit from further clarification or discussion include:

1. One of the main drawback of using EHR data for PGx analyses is the potential impact of confounding factors such as medication adherence and lifestyle changes. Authors should discuss how non-adherence might bias the results and propose methods to mitigate this in future studies.

2. Use directed acyclic graphs (DAGs) to visually represent and explain the identified causal relationships.

3. Although the non-European analyses were on smaller sample sizes, authors should perform a power analysis to determine the sample size needed for adequately powered studies in other ancestral groups and discuss these limitations.
4. Additionally, comparing the allele frequencies and effect sizes of the identified variants across different populations using publicly available databases such as gnomAD may help understand these effects in other ancestral groups.
5. While the associations between PRS and drug responses are well-described, the paper could benefit from more discussion on the potential biological mechanisms underlying these relationships and to better understand why high PRS might lead to different responses.
6. Consider exploring potential interactions between different PRS or between PRS and other clinical factors that might influence drug response.
7. While the focus of this study was on immediate biomarker changes, it would be interesting to explore whether these PGX and PRS associated differences in drug response translate to differences in long-term clinical outcomes (e.g., cardiovascular events for statins, diabetes complications for metformin).

(Remarks on code availability)

There are no codes in the scripts folder. I also could not view the snakemake config file.

Reviewer #3

(Remarks to the Author)

The manuscript "Levering large-scale biobank EHRs to enhance pharmacogenetics of cardiometabolic disease medications" investigates the genetics mechanisms of several pharmacotherapies using trajectories. This work is well positioned and stands to make a nice contribution to this area of the literature and the study visuals/tables are well designed, however, some organizational and clarity questions remain.

1. The manuscript would be strengthened by a reworking of the introduction to include a more thorough review of the current state of the literature and how this study enhances what remains unknown. The final paragraph introduces the investigation into biomarker changes, which is not set up in intro nor how the results of this study will be clinically informative. (The lengthy "overview of analysis" in the results section could be greatly simplified, as much of this is already covered in the methods, to make room for a revised & expanded intro)
2. As the manuscript currently reads, genetic underpinnings and pharmacogenetics seem to be conflated and interchangeable - clarity around this language and what the study hopes to achieve would improve the manuscript.
3. The study samples and demographics are minimally described. It is unclear to me how concomitant meds were handled.
4. Authors veer in to interpretation and limitations at times during the results section (e.g., pg 12 "While this could be a power issue given the lower sample size...") rather than focusing on the findings.
5. How were PRS drug outcomes ("better or worse response") categorized/determined?
6. The discussion begins by stating this study presents a theoretical framework to model drug response & longitudinal phenotypes (and later that this study "elucidates the theory of modelling biomarker differences"). This is not how the rest of the manuscript is presented. Suggest moderating this phrasing or introducing what the framework is to a much greater extent.
7. Can authors comment on the choice to use All of Us at the replication dataset? It seems other biobanks would be better datasets given the limited EHR data.
8. Surprisingly, the limitations do not comment on the lack of diversity in this study.
9. The conclusion leaves me wanting in terms of next steps and how this data will be clinically relevant.
10. What was the rationale for the stringent/lenient time periods? Why take the average of all post-tx measures?
11. Was dosage accounted for in any modeling?
12. Who were the controls in the GWAS?
13. What method was used to calculate PRS?

(Remarks on code availability)

Version 1:

Reviewer comments:

Reviewer #1

(Remarks to the Author)

The authors have put a great deal of work into this very thorough revision. All of my comments from my many reviewer comments have all been addressed very clearly. I am really pleased that the authors found my comments and suggestions to be very helpful in improving the paper, and helping to spot some changes that needed to be made to some analyses. I have also read all responses to the comments of other reviewers, and I am happy that all these comments have also been addressed well. In particular, it is good to see that the authors have checked the code availability, as flagged by another

Reviewer, and have submitted their PGX phenotypes to UK Biobank.

(Remarks on code availability)

As above, it is good to see that the authors have checked the code availability, as flagged by another Reviewer, and have submitted their PGX phenotypes to UK Biobank.

Reviewer #2

(Remarks to the Author)

All points have been addressed.

(Remarks on code availability)

Reviewer #3

(Remarks to the Author)

I thank the authors for the thoughtful and thorough revisions to the manuscript. I have no remaining concerns.

(Remarks on code availability)

Colour code:

Answers to the reviewers are written in green.

Unchanged elements borrowed from the manuscript are written in dark blue (italic), and changes in the manuscript are written in brown (italic).

REVIEWER COMMENTS**Reviewer #1 (Remarks to the Author):**

Summary

The authors have performed drug response efficacy PGX analyses using UKB as discovery and AOU as replication, in order to demonstrate the utility of large EHR-based biobanks for PGX. First, the authors perform GWAS analyses for variant discovery, for several different cardiometabolic drug responses, e.g. cholesterol response to statins, HbA1c response to metformin, and BP/HR response to antihypertensives. They also perform PRS analyses to test the hypothesis that the PRS of the corresponding biomarker trait is associated with the drug response phenotype. And finally, they consider different statistical models, to give recommendations for unbiased study design for PGX. This is novel work, as explained nicely in the Intro, that PGX has not yet fully attempted to exploit the large-scale biobank data.

I particularly liked the following aspects of the manuscript:

- The opening paragraph of the Intro, setting the scene for the challenges of PGX, and the nice perspective with the GWAS catalog statistic.
- The creativity in the figures provided
- The comprehensive inclusion of many thorough sensitivity analyses, which address many controversies in the literature on PGX study design and statistical models. Especially the “modelling drug response and longitudinal change phenotypes” section, which is very convincing indeed.

We would like to thank the reviewer for the constructive feedback and thorough assessment of our work. We are particularly grateful for all the suggestions aimed towards clarification and expansion of the concepts that previously lacked details as well as for the requests to add further information on the available EHRs which we believe will be useful to fellow researchers. The comments also led us to identify a small mistake in our preliminary analyses of longitudinal data availability that impacted how we defined the post-treatment window in our submitted manuscript. In this revision, we have redefined the post-treatment windows in the stringent and lenient filtering strategies to align them with the data availability as well as to address the reviewer’s point that different drugs have different onsets of action. Please find our answers to the comments below.

Comments to Authors

Please could the authors address the following points:

1) The authors have covered so much within this one paper: the genetic discovery of the PGX-GWAS with novel discovery and comparison to literature findings; the PRS analyses testing a completely separate hypothesis; the statistical model methodology; lots of sensitivity analyses; the study design features of novel EHR-biobank-based PGX. Whilst all content makes a valuable contribution to the field, I almost wonder whether it may be too much for one single manuscript, and whether it could benefit from being split into different papers. This may need guidance from the Editors.

We thank the reviewer for this suggestion. It is true that the manuscript covers a lot of content, however, splitting it may result in incomplete papers as the different topics are intertwined. While EHR-biobank-based PGx is at the core of the paper, a deeper dive into the statistical model methodology justifies the chosen drug response modelling approach and makes the comparison with the literature more consistent. The PRS analyses could be presented as a work on its own, but we followed the structure of general GWAS papers, where it would be strange not to include PRS and heritability estimation. The analysis of the PRS is also particularly interesting as readers would think that the baseline PRS association with the change phenotype is only an artefact of baseline adjustment, while this is clearly not the case.

2) There are (at least) two recent, major references, in the field of statin-response PGx that the authors have omitted from their literature review, and missing completely from their references: (i) Oni-Orisan et al, Genomic Medicine 2020 (PMID: 31969989); (ii) Zhang et al, Genomic Medicine 2022 (PMID: 35680959). Please can the authors ensure their whole literature review is completely up-to-date, across all drug-response traits considered within this paper.

We thank the reviewer for pointing us to these papers and we have updated the introduction as follows:

Several PGx GWAS consortia have formed over the years to study the genetics of drug efficacy in larger sample sizes. For instance, the Genomic Investigation of Statin Therapy (GIST) consortium has identified variants in the LPA, APOE, SORT1/CELSR2/PSRC1 and SLCO1B1 regions as modulators of low-density lipoprotein cholesterol (LDL-C) response to statins by combining randomized controlled trials (RCTs) and observational studies (Postmus et al., 2014). Using electronic health records (EHRs), the Genetic Epidemiology Research on Adult Health and Aging (GERA) cohort has additionally identified the APOB and SMARCA4/LDLR loci as genetic determinants of statin response (Oni-Orisan et al., 2018, Oni-Orisan et al., 2020). Similarly, the Metformin Genetics (MetGen) consortium has identified [...]

As both papers focus on baseline adjustment with opposing conclusions we have updated the discussion to discuss this important topic more extensively and have included the following sentences:

As similar study designs have used differing GWAS models to estimate pharmacogenetic effects, we developed a realistic model for biomarker differences and showed how baseline adjustment will induce biases for genetic variants associated with baseline levels. In recent years, there have been debates about baseline adjustment in analyses of change (Glymour et al., 2005, McArdle and Whitcomb, 2009, Oni-Orisan et al., 2020, Zhang et al., 2022). In the context of PGx GWAS, some researchers have advocated for no baseline adjustment (Oni-Orisan et al., 2020), while others have argued otherwise (Zhang et al., 2022). There is a consensus that measurement errors can introduce a bias and spurious associations with longitudinal change upon baseline adjustment (regression-to-the-mean bias) (Blomqvist et al., 1987, Glymour et al., 2005, Tu et al., 2007, McArdle and Whitcomb, 2009, Oni-Orisan et al., 2020, Zhang et al., 2022), however, it is unclear whether baseline adjustment also affects drug response phenotypes since drug effects are likely larger than measurement errors. In the context of pharmacogenetic interactions with biomarker change, only the direct genetic effect on change is of interest, but genetic variants likely also affect baseline values which in turn impact the magnitude of change induced by the medication. Zhang et al., 2022 recommend adjusting for baseline as their model represents baseline as a mediator between genetic variants and quantitative change (Zhang et al., 2022). We argue that the mediator model is

inappropriate because it assumes that the change depends equally (i.e., same regression coefficient) on the genetic and the environmental component of the baseline value. However, their model does not test for this gene-environment “equivalence” and in all applications we examined, this is not the case: the change depends very little on the baseline genetics and almost all its association is with the environmental component of the baseline value. Hence adjusting for the baseline value rather than eliminating a bias, it introduces one. For this reason, we, instead, propose modeling drug response as the biomarker difference between two time points. We provide theoretical derivations as to the origin and magnitude of the bias arising from baseline adjustment and demonstrate in GWAS on drug-naive control individuals that our model, which supports not adjusting for baseline, does not generate inflated type I errors as suggested by the authors (Zhang et al., 2022). Furthermore, since control individuals are likely subjected to similar measurement errors and gene-environment interactions as medication users, the control experiments reassure that these effects are negligible compared to the pharmacogenetic effects. We show that for unbiased pharmacogenetic effect estimation, both absolute and logarithmic relative biomarker differences [...]

3) In particular, the Oni-Orisan et al study performs statin-response PGX GWAS using the GERA study, which is also based on EHR data, meaning that the use of EHR data from UKB is not entirely novel for the first time, for LDL-statin response PGX GWAS.

We have updated the introduction to mention that EHR data have been used previously to perform statin-response PGx GWAS by Oni-Orisan et al. The paragraph reads now as follows:

Biobanks coupled with EHRs that comprise medication data provide new opportunities to discover PGx associations [...]. Yet, PGx biobank studies so far have either focused on known pharmacogenes and their associations with adverse drug reactions, drug dosage and drug prescribing behaviour or analyzed the genetics of temporal medication use in isolation of disease phenotypes. Except for the GERA cohort which solely utilized EHRs for the study of LDL-C response to statins (Oni-Orisan et al., 2018, Oni-Orisan et al., 2020), the integration of longitudinal medication and phenotypic data to screen for genetic determinants of drug efficacy at a biobank scale remains largely unexplored.

4) Importantly, the Oni-Orisan et al paper also reports the LDLR locus for LDL statin response from their baseline-adjusted GWAS analysis, meaning that 1 of the only 2 novel loci the authors report here, is actually not a novel finding. This is important to correct.

We thank the reviewer for pointing out that the LDLR locus has been identified in an earlier study. We have removed all mentions as to the novelty of the LDLR locus (abstract, discussion) and have added the LDLR and APOB loci to the Table 2 where we do the comparison with the literature. We would also like to point out that in our revised analysis, both LDLR and APOB reached genome-wide significance. We have updated the literature comparison result section paragraph by adding the following sentences:

The LDLR and APOB loci reached genome-wide significance in the baseline-adjusted GWAS in the GERA cohort, but were believed to be false positives as a consequence of baseline adjustment (Oni-Orisan et al., 2020). These SNPs were nominally significant in our baseline adjusted biomarker difference GWAS. However, the reported LDLR-associated SNP rs67337506 is in LD with the rs118068660 SNP ($r^2 = 0.29$) which we identified as a genuine PGx locus in the absolute LDL-C change GWAS (Table 1). Similarly, the reported APOB-associated SNP rs1713222 is in LD with the rs10199768 SNP ($r^2 = 0.14$) which reached genome-wide significance in the absolute LDL-C change GWAS. Thus, although both loci

were identified by a biased PGx model, SNPs in LD reached genome-wide significance in our unadjusted baseline GWAS.

5) Important data availability consideration prior to publication: Should the actual drug-response phenotypes derived here from UKB also be returned to UKB, and made publicly available to other researchers?

We thank the reviewer for this suggestion. We have created tables for each of the ten drug response phenotypes (LLD-C, HDL-C and TC response to statins, HbA1c response to metformin, SBP response to antihypertensives and HR response to beta blockers) for our default filtering strategy (lenient filtering with average over multiple measures if available) which we have submitted to the UK Biobank. Based on our past experience, unfortunately, these data are released back to the community with substantial delays. Nonetheless, we hope that availability of these drug response phenotypes will foster new research in the space of pharmacogenomics and beyond.

6) There is a clear missing gap in this article, where, having reported two novel loci, the authors provide no discussion or information on the biology or hypothesised mechanism or known function of these genes.

We agree with the reviewer that information on the biology and hypothesized mechanisms were missing for the identified PGx loci and we have updated the PGx discovery result section (Result section: “*Drug response GWAS using EHRs from the UKBB*”) to include more details. We note that in our revised analyses, *ZNF800* no longer reached genome-wide significance.

In the LDL-C response to statin GWAS, APOB (rs10199768 G>T, beta = 0.056, p-value = 1.77e-08) and LDLR (rs118068660 C>T, beta = -0.119, p-value = 1.23e-10) were found to influence absolute biomarker change, while the SLC22A3/LPA (rs10455872 A>G, beta = -0.146, p-value = 4.87e-17) and APOE (rs7412 C>T, beta = 0.232, p-value = 1.11e-28) loci were found to influence relative (logarithmic) biomarker change (Table 1; Figure 2b; lenient filtering with average values if available, N = 18,753). All four genes encode proteins with a well-established role in serum lipids level regulation, with APOB, APOE, and LPA encoding for apolipoproteins that bind lipids such as cholesterol and triglycerides and organize them into various types of lipoprotein particles (e.g., LDL-C), enabling their transport in the blood and distribution throughout the body (Mehta et al., 2022). The two former apolipoproteins further act as ligands for the LDL receptor encoded by LDLR, allowing the binding and internalization of lipoprotein particles harboring these lipoproteins, ensuring delivery of lipids to the cell and regulation of serum LDL-C levels (Mehta et al., 2022). Importantly, all four genes have previously been involved in modulating response to statins (Oni-Orisan et al., 2020, Postmus et al., 2014, Chasman et al., 2012, Deshmukh et al., 2012). More specifically, the CARDS trial showed that the PGx association in the SLC22A3/LPA locus results from LDL-C levels also including LDL-C residing in Lp(a) particles (Deshmukh et al., 2012). The LPA variant rs10455872-G which is associated with a lower response to statins also associates with increased levels of Lp(a) which remain unchanged upon statin treatment. Thus, the relative higher proportion of Lp(a) particles in rs10455872-G carriers gives rise to this PGx association even if LDL-C residing in statin-responsive LDL particles drop to similar levels as in rs10455872-A carriers.

TC response to statins, for which we had a larger sample size (more TC than LDL-C measures are available in the primary care data, N = 28,880) confirmed the identified loci at APOB, LDLR, SLC22A3/LPA (the latter being identified in both the absolute and relative biomarker

*change GWAS), and APOE, while also identifying the SNP rs4149056 T>C in the SLCO1B1 locus (beta = -0.063, p-value = 1.10e-08). SLCO1B1 encodes for the OATP1B1 transporter, which mediates the intracellular uptake of a wide range of substrates, including statins (König et al., 2006, SEARCH Collaborative Group, 2008). Importantly, the variant we identified, also known as Val174Ala or SLCO1B1*5, has previously been associated with LDL-C statin response (Hopewell et al., 2013, Postmus et al., 2014) as well as myopathy (Patel et al., 2015), a rare but well-described side effect that has been attributed to increased statin blood concentrations due to the reduced uptake capacity of the encoded protein (SEARCH Collaborative Group, 2008).*

7) With the final conclusion being that “larger sample sizes will be needed”, there seems to be a big missed opportunity from this work. Having used AOU data for replication, why have the authors not considered meta-analyses fully combining GWAS data together from both UKB and AOU, in order to achieve this? Similarly, having compared their UKB PGX GWAS findings to previously reported results from the literature, when some of these GWAS data must presumably be publicly available, why have the authors not considered combining their biobank PGX GWAS data together with the already existing RCT-based GWAS data via a full, large-scale meta-analysis? What would the recommended study design be, for combining EHR biobank based PGX GWAS with more traditional RCT-based PGX GWAS?

We agree with the reviewer that this work could greatly contribute to the creation of a large meta-GWAS largely exceeding current sample sizes and overcoming the issue of limited statistical power. However, there are several reasons and issues which prevented us from generating such a meta-analysis within the framework of this study:

- 1) The primary goal of this study was to show how biobanks coupled to EHRs can contribute to PGx research with larger sample sizes than RCTs. Thus, we presented data filtering strategies and longitudinal PGx GWAS models (absolute and relative biomarker differences) to guide such PGx-EHR analyses while also flagging inconsistencies within the literature with respect to baseline adjustment. As pointed out by the reviewer, this single topic covered already a lot of content and adding a meta-analysis would probably result in a paper that is no longer digestible. The likely identification of novel hits would require a deeper analysis of plausible biological mechanisms which would add yet another dimension to the manuscript.
- 2) We have not considered a meta-analysis between the UKB and AOU GWAS as this would defeat the point of having an independent replication biobank. The AOU data was used to conduct replication analyses of both our main analyses in the UKBB and from the literature, and as such we focused on the extraction of lipid response to statins as well as HbA1c response to metformin. For systematic full meta-analyses, the AOU analyses would need to be extended to the ten drug-biomarker pairs which is not straightforward given the complexity of anti-hypertensive drug regimens as pointed out in the discussion paragraph of the limitations. Given the low sample sizes achieved for antihypertensives in the UKBB where EHR data is more complete, we expect sample sizes to be even lower, minorly contributing to an increased sample size. Already this replication effort revealed substantial heterogeneity between the two biobanks, the nature of which need to be explored in greater depth before just meta-analyzing them. A meta-analysis between UKB and AOU could be a work on its own and would also merit further analyses into the quality and EHR completeness between these two biobanks and how this affects observed biomarker reductions and resulting PGx associations. Also, such meta-analysis creates infinite loops, since the emerging genome-wide significant hits would then need to be replicated in yet

another cohort, but that could then be meta-analyzed, etc.

- 3) Unlike the reviewer's assumption, the full PGx GWAS that we consult in the literature comparison are not publicly available except for the GERA cohort which published the full GWAS summary statistics in the GWAS Catalog. The remaining studies only published the association results of their genome-wide significant independent hits. To generate a full, large-scale meta-analysis this constitutes a big hurdle, and it goes beyond the scope of this work to set up collaboration agreements with individual PGx cohorts to get access to their PGx GWAS summary statistics.
- 4) Beyond the unavailability of PGx GWAS summary statistics, a major issue is the heterogeneity of drug response GWAS models that have been used by the different cohorts. Some cohorts have considered the biomarker difference (either absolute or log-transformed) adjusted for baseline, others have not adjusted for baseline, and others have considered the percent change $(\text{post-base})/\text{base}$; this was the case for the GERA study which is publicly available). A full, large-scale meta-analysis likely requires the recreation of drug response phenotypes and GWAS summary statistics within each cohort according to a uniform model. We would very much like to see such a concerted effort and have even reached out to several PGx cohorts to probe their interest in a large-scale PGx meta-analysis. However, establishing such a consortium and getting back the PGx GWAS summary statistics from the individual cohorts takes time and goes beyond the scope of this study.

EHR-biobank based PGx GWAS and traditional RCT-based PGx GWAS can be combined in a simple meta-analysis, e.g., through an inverse-variance weighting scheme. Other consortia (for instance, the GIST consortium) have combined observational PGx GWAS (which is of similar nature than EHR-based GWAS) with RCT based GWAS in their lipid response to statin GWAS.

We have updated the conclusion sentence to provide more concrete avenues on how we see the future with respect to larger sample sizes:

[...] We find that the influence of common and rare genetic variants on drug response is relatively low, and larger sample sizes achieved by combining drug response GWAS from observational, EHR and RCT data as well as studies in more diverse ancestries will be needed to capture the full extent.

8) Lines 158-160: Even though no GW-significant variants have been reported in the literature from BP drug response PGX, I still think it would be important for the authors to consider some of the "peri-significant" results that have been reported in the literature, to compare what their exact equivalent results are from biobank PGX analyses. Please add these lookups to your paper.

We agree with the reviewer that the multitude of PGx analyses conducted on antihypertensive merit more attention in the replication analyses with the literature. We have thus added lookups from the analyses of SBP response to antihypertensives and HR response to betablockers as a Supplementary Material (Table S11). Given the numerous loci reported in studies as small as 50 individuals or less (Oliveira-Paula et al., 2019), we have applied a threshold of $N > 300$ to these reported loci and conducted systematic look-ups. As expected, these lookups resulted in no replication at all, as these signals resulted from highly underpowered studies. We updated the literature comparison section (Result section "*EHR-derived PGx GWAS recover known PGx loci*") to read as follows:

Although several loci have been found to influence blood pressure and heart rate response to anti-hypertensives at a suggestive p-value threshold, no genome-wide significant hits have been reported (Oliveira-Paula et al., 2019). Among the 13 loci identified at a suggestive significance level in studies conducted in samples exceeding 300 participants (Oliveira-Paula et al., 2019), only the SNP rs4149601 in the NEDD4L gene region found to SBP response to diuretics could be replicated at a nominal significance level in the UKBB (p-value = 0.032; Table S11). However, replication only occurred in the baseline adjusted analysis which is more likely to yield false positives (Figure 3a).

9) The “Study population” section of the Methods chapter is not at all sufficient and detailed enough for publication, e.g: how were individuals of “white British ancestry” selected? What threshold was used for “no excessive number of relatives”? Up to which date were withdrawn-consent individuals applied? In general, please explain the breakdown of N throughout the QC process steps in more detail. Furthermore, what does “#531” correspond to? What does “depending on the data supplier” mean?

We have added more details to the “Study population” section of the Methods chapter and clarified the breakdown of N throughout the QC process. The definition of “white British ancestry” (based on self-reporting and concordant with PC analysis) and “excessive relatives” (participants with more than 10 putative third-degree relatives in the kinship table) is provided by the UKBB in their QC file (resource #531) and results from the flagship paper (Bycroft et al., 2018) which we have now specified. We also added further information on the data suppliers that provide the linked primary care data.

The updated section reads now as follows:

The UK Biobank is a prospective study of ~ 500,000 participants of whom 45% (N ~ 230,000) are linked to the primary care data of the United Kingdom’s National Health System (Bycroft et al., 2018). The primary care resource contains longitudinal data of GP prescription records (datafield #42039) and GP clinical event records (datafield #42040) encoded through the British National Formulary (BNF), National Health Service (NHS) dictionary of medicines and devices (DM+D), Read V2 and Clinical Terms Version 3 (CTV3) codes and are available up to 2016 or 2017, depending on the data provider (EMIS/Vision for Scotland and Wales, and TPP and Vision for England; detailed description of the linked primary care data is provided by the UKBB at https://biobank.ndph.ox.ac.uk/showcase/showcase/docs/primary_care_data.pdf). Analyses were conducted on individuals of white British ancestry (N = 190,754), with no excessive number of relatives (N = 76) and differing reported and inferred gender (N = 133; we used the definition of white British ancestry, excess relatives, differing gender as defined by the UKBB Sample-QC file of datafield #531, Bycroft et al., 2018), excluding participants who have withdrawn their consent up to April 2023 resulting in N = 190,545 participants who were considered in the drug response analyses.

10) (line 346) I have one major concern with the time intervals used. Whilst I appreciate the need for trying either stringent vs lenient filtering, and the need to try to maximize N, I am concerned as to why the post-treatment period has to start at least 6 months after medication started? What is the rationale behind this? Would there be UKB individuals with a smaller post-treatment time interval than this? I believe that RCTs in the PGX field have shown that a shorter post-treatment follow-up time makes drug response more accurate. Similarly, the end-point of the post-treatment time period for the stringent filtering, of up to 1.5 years, already seems fairly lenient, compared to follow-up times from RCT data used for PGX in the literature. Please could the authors discuss how their time-windows compare to

equivalent time-windows of RCTs within the PGX literature, to give a clearer idea of how different the EHR-biobank based PGX study design actually is.

We will answer 10) and 11) together.

11) Furthermore, these optimal time-windows may be both drug- and trait-specific. Please could the authors discuss this.

We thank the reviewer for raising this point which has led us to identify a small mistake in our preliminary analyses of longitudinal data availability. Instead of assessing the time between the first prescription and first post-treatment biomarker measure, we assessed the time between the first prescription and *second* post-treatment biomarker measure. Consequently, we believed that the availability of post-treatment measures *before 6 months* is very sparse. We have added a supplementary figure (Figure S5) of the distribution of post-treatment measures which shows that there are measures before 6 months, and we have updated the post-treatment time windows as described in the *new* result section “*The impact of filtering in the PGx-EHR study design*” which is dedicated to EHR filtering strategies and their impact on the identified PGx loci. In the revised version, the post-treatment window starts before 6 months and is drug-specific. Furthermore, the post-treatment time window for the stringent filtering ends after 1 year to be more aligned with RCT data.

The new result section “*The impact of filtering in the PGx-EHR study design*” reads as follows:

In EHRs, medication start, baseline and post-treatment measures are not readily available, and we tested multiple strategies to extract drug response phenotypes and assess their impact on PGx associations. We introduced a prescription regularity parameter to proxy drug adherence under the assumption that skipped prescriptions are the result of inconsistent medication intake. This prescription regularity parameter (or prescription completeness) is defined by the presence of a prescription at least every two months for the duration of the post-treatment period and a completeness of 100% is obtained if this is the case. To account for varying time window thresholds, prescription regularities, changes in medication regimens, and single or multiple biomarker measures at baseline and post-treatment, we defined stringent and lenient filtering strategies, assessing the impact of using single versus multiple biomarker measures for each.

A main difference between the lenient and stringent filtering scenario was the extension of the baseline and post-treatment periods. In Figure S4, we show the distribution of the time between the closest baseline measure and prescription start, and in Figure S5, the distribution of the time between prescription start and the closest post-treatment measure. The median of this distribution was between 0 and 41 days for the baseline measure and 111 and 273 days for the post-treatment measure across medication-biomarker pairs (Table S4). As we anticipate the stringent filtering scenario to more closely reflect the design of an RCT such as the JUPITER trial (LDL-C response to rosuvastatin) where the post-treatment value was taken 1 year following medication start (Chasman et al., 2012), we chose a baseline period starting 100 days before prescription start and a post-treatment period starting 60 (antihypertensives and beta blockers) or 100 days (statins and metformin which have delayed effects on lipids (Tsiara et al., 2003) and HbA1c (Ulcickas et al., 2006)) up to 1 year following prescription start. In the lenient filtering, we extended the baseline period up to 1 year preceding and up to 2 years following prescription start. Note that the post-treatment period in the lenient filtering setting remains more stringent than in observational studies such as those included in the GIST study where the post-treatment period varied widely between cohorts and could last up

to 5 years (Postmus et al., 2014). When testing the impact of the time to first post-treatment measure on the biomarker difference, we observed that increased follow-up times resulted in reduced biomarker differences, even in the stringent filtering scenario where this parameter could not vary as widely (the follow-up time explained up to 1.1% of the variance, Table S6). Likewise, a lower prescription completeness value resulted in decreased biomarker differences in the stringent and the lenient filtering settings, where the required completeness were 60% and 30%, respectively (the prescription regularity explained up to 7.6% of the variance, Table S6). Despite the differences in the stringent and lenient filtering scenarios, average baseline and post-treatment biomarker levels were almost identical between the resulting drug cohorts (differences of less than 3%; Table S4).

Between the two filtering strategies, the sample size increased from 40% (metformin-HbA1c) up to 334% (CCB-SBP) when relaxing the filtering criteria (Table S4). For statins, this rise was largely due to the extended baseline and post-treatment period. For metformin and antihypertensives, we excluded individuals taking any related medication in the stringent filtering setting, whereas, in the lenient setting, sample size largely increased by allowing metformin and antihypertensives to act as add-on therapy to sulfonylureas and second-line antihypertensives, respectively, if consistently taken during pre- and post-treatment periods of the studied medication (Figures S2-3). As a consequence of lower statistical power, only 4 out of the 10 signals found in the lipid-statin GWAS were detected in the stringent filtering scenarios (Figures S8-9; Table S8). Furthermore, we tested the difference between assessing a single baseline and post-treatment measure (the closest to the prescription start) and averaging over all available measures present in the baseline and post-treatment periods. The impact was minimal, and the only difference was observed for the *SLCO1B1*-associated SNP rs4149056 which did not reach genome-wide significance with a single measure (p -value = $1.49e-5$).

Figure S4. Distribution of the time between baseline measure closest to prescription start and prescription start for the ten cardiometabolic medication-biomarker pairs. In this analysis, baseline measures were restricted to occur one year before and up to seven days after prescription start (-7 on the graph).

Figure S5. Distribution of the time between post-treatment measure closest to prescription start and prescription start for the ten cardiometabolic medication-biomarker pairs. In this analysis, post-treatment measures were restricted to occur 30 days and up to three years (1095 days) after prescription start.

12) Additionally, referring to line 469, could the criteria for controls have been stricter? And if so, would this have a beneficial impact?

We selected medication-naive controls according to the availability of two biomarker measures within an interval of the combined baseline and post-treatment periods which ranges up to three years. A shorter time period could be chosen, however, in that case the analysis would no longer match the drug analysis which would defeat the purpose of the control experiment.

13) Due to line 261, it is important, please, for the authors to discuss why there are so few PGX signals in general, so the authors can give recommendations for future PGX studies.

We have expanded the discussion section on why there are so few PGx signals in general and refer to the numbers estimated in Nelson et al. 2016, while also recapitulating on the power analyses that we have added to the manuscript. Furthermore, we put a greater emphasis on the heritability estimates which support a low genetic component.

Overall, we found only a few genetic variants to influence cardiometabolic drug response in line with other studies that often identified only a few or even no genome-wide significant signals (Dawed et al., 2023, Singh et al., 2019, Zhang et al., 2023). A review on 76 drug efficacy GWAS reported that only 15% of drugs exhibit robust gene-treatment interactions (Nelson et al., 2016), the extent of which largely depends on the drug's mode of action. In their review, the authors estimate that only 56% of these associations are clinically useful resulting in a probability of 8.1% that a drug has a clinically relevant genetic predictor (Nelson et al., 2016). Our study supports their conclusion that large pharmacogenetic effects are unlikely for the studied cardiometabolic drug traits, although more signals may emerge with larger sample sizes. While no single genetic variant will have strong predictive performance, well-powered PGx GWAS could enable the derivation of clinically useful PGx PRS. If we consider the LDL-C GWAS on unrelated UKBB participants of European ancestry (343,621 samples; <http://www.nealelab.is/uk-biobank>), 416 independent genome-wide significant hits are found. This number is estimated to decrease to 26 (6.3% of the hits) if the sample size were 28,800 which equals the sample size of the largest PGx GWAS in this study. With increased sample sizes, the number of PGx signals will also increase, however, low and often insignificant heritability estimates support that the genetic influence on the studied drug response phenotypes remains low (Table S17). For LDL-C response to statins we estimated the genetic heritability to be 3.4% (95% CI: [0%, 8.6%]) in line with an earlier study that reported a non-significant heritability estimate (11.7%, 95% CI: [0%, 29%], Oni-Orisan et al., 2019). While inter-individual variability in drug response could also depend on rare variants, we only identified a single gene GIMAP5 associated with HbA1c response to metformin.

14) The two different phenotype versions “absolute” and “relative” are defined within the Methods, and presented within Fig 1. But the motivation behind these two different phenotypes needs more explanation, as it is essential for readers to understand the differences before the later results sections. Furthermore, please add more discussion in the paper about which of these two phenotypes is better.

We thank the reviewer making this point and we have expanded on the motivation as to why we analyze “absolute” and “relative” biomarker differences in the first result section “Overview of the analysis”:

For each drug response phenotype, we derived an absolute and logarithmic relative biomarker difference as outcome traits as both approaches are commonly employed in drug response

studies. Lipid and blood pressure response have been studied on both the absolute (Chasman et al., 2012, Svensson et al., 2011, Hamrefors et al., 2012, Salvi et al., 2017, Singh et al., 2019) and relative scale (Chasman et al., 2012, Postmus et al., 2014, Postmus et al., 2016, Oni-Orisan et al., 2020, Svensson et al., 2011, Hamrefors et al., 2012), while HbA1c response has commonly been analyzed on the absolute scale (Godarts et al., 2011, Zhou et al., 2016, Rotroff et al., 2018).

Furthermore, we added explanations as to the difference between these two phenotype definitions in the Discussion:

We show that for unbiased pharmacogenetic effect estimation, both absolute and logarithmic relative biomarker differences can be assessed. Whether absolute or relative reductions are most determining in lowering the risk of associated clinical diseases may be dependent on the studied medication-biomarker pair. If biomarker levels are linearly associated with the disease risk, absolute change is more relevant; however, if this relationship is exponential or quadratic, then relative change is of more importance. As to pharmacogenetic interactions, it is a priori unknown on which scale genetic variants act linearly, evidencing the usefulness of testing both absolute and relative biomarker changes. Regarding heritability estimates, there were no significant differences between the two phenotype definitions in this study (Table S17). Thus, we recommend conducting discovery analyses on both scales, and in a next step, examine whether the observed associations on the respective scales are clinically relevant.

15) The final limitations section of the Discussion is very important for this work, and requires further expansion. For the 1st limitation of already being on medication, please discuss how this would impact the data, bias and findings; for the 2nd limitation of restricted sample size, please discuss if N is hoped to improve in UKB in the future or not, for example perhaps with more follow-up visits data; for the 3rd limitation of polypharmacy, please provide more clarity on what their strategy actually was; for the 4th limitation of not using clinical endpoints, please discuss if, and how, this could be done in the future, in either RCTs or in EHR-biobanks; for the final limitation could the control and medication group be “matched” in any way, or could disease-PRS stratification help to improve this?

We have updated the limitation section in the Discussion to address the points raised by the reviewer, and further included limitations with respect to ancestry diversity:

Our study has several limitations. First, we rely on data from EHRs to derive before and after treatment biomarker levels, and thus cannot exclude the possibility that individuals were already on medication before the first recorded prescription. If these individuals were to make up a large proportion of the cohort, the analysis would no longer capture the genetics of drug response, but rather of disease progression whilst on medication. Second, despite a large fraction of individuals with medication records in the biobanks, final PGx cohort sample sizes are limited by the number of participants on a certain medication and further reduced due to incomplete or missing data. Of the ~ 65,000 participants with a statin prescription in the UKBB, 63% could not be considered for the LDL-C response analysis because of missing baseline and/or post-treatment measures. With an optimistic outlook, sample sizes can be expected to double in the future when all UKBB participants are linked to their primary care records (this number currently stands at 45%) and could increase even further with longer follow-up times and an ultimate increase in cardiometabolic medication users. Third, and related to the previous limitation, most participants have follow-up measures after a year (Figure S5, Table S4) which limits the study of immediate pharmacological effects, and measured drug responses may harbour a disease progression component. Frequency of follow-up measures constitutes a major limitation compared to RCT data where more frequent and broad

biomarker measures are likely available. Forth, polypharmacy has only been taken into account within and not across medication groups, meaning that a statin user was excluded from the analysis if, for instance, fenofibrate (lipid-lowering medication) was initiated during the follow-up period while exclusion did not apply in cases where metformin (anti-diabetic medication) prescriptions were recorded during follow-up for that same individual. Even within, especially for antihypertensives where frequent changes in medication regimen occur, it can be difficult to determine appropriate filtering and covariate strategies to study individual drug classes as sample sizes are too low when restricting the analysis to individuals taking antihypertensives from a single class (i.e., stringent filtering strategy). Fifth, our analysis focused on continuous biomarkers and not on clinical events. LDL-C, SBP, HR and HBA1c merely serve as surrogate endpoints of CHD and T2D events, and the genetic interplay with drug efficacy may be different when assessing hard clinical endpoints. EHRs provide a rich resource for conducting analyses similar to those done with RCT data (Mega et al., 2015, Natarajan et al., 2017, Damask et al., 2020, Marston et al., 2020), and for studying genetic interactions with drug efficacy on clinical events in real-world settings. Sixth, analyses have been conducted in individuals of European ancestry. While allele frequencies of identified PGx signals were largely consistent across genetic ancestry groups (Table S10), analyses in diverse ancestry groups are likely to identify PGx effects missed in individuals of European ancestry. For a PGx signal with modest effect size such as SLCO1B1, we estimated that a sample size of ~150,000 is required for 80% power in African/African Americans (MAF of 3.1% compared to 15% in Europeans), showcasing that even with cohorts the size of the current study increased numbers of PGx signals could be identified provided that the MAF is high. Finally, we rely on observational data to draw conclusions about drug efficacy. Although, we contrast the results with control analyses on longitudinal biomarker change, control and medication groups were not defined randomly and by definition have markedly different disease profiles. Better matching of controls through methods such as propensity scores that include genetic predisposition could address this issue in future studies.

16) Line 323 of the Methods is the only place that clearly states that only “45% (N~230,000)” of UKB are linked to primary care data. This is indeed a major limitation, which needs emphasizing and discussing more.

We have now made this point clearer in the Discussion and mention how numbers are likely to change once all participants are linked to primary care records (see our answer to comment 15). Furthermore, we now mention this number earlier in the manuscript and have updated the first sentence of the result section “Overview of the analysis” to the following:

In the drug response discovery analyses, we extracted longitudinal prescription and response biomarker data from the UKBB primary care records which we combined with phenotypic data from the assessment visits (currently ~230,000 (45%) UKBB participants are linked to their EHRs).

17) The Introduction ends positively showing the benefits of this work. However, it is clear from the final sections of the Discussion, that there are still many challenges & limitations of biobank-based PGX research. So I think the authors need to ensure there is still a correct balance of strengths vs limitations throughout the whole manuscript. It is helpful for the field, if the challenges that still remain for biobank-based PGX are discussed in detail.

We thank the reviewer for pointing out the issue with the balance between strengths and limitations within the manuscript, and we agree that this work also serves to emphasize the many challenges that exist when using EHRs for PGx research. We made 1) changes in the introduction to introduce challenges early on, 2) added a new result section to analyze

issues linked to EHR data in more detail and 3) added a paragraph to the Discussion to discuss these results and link them to related studies, 4) updated the limitation paragraph in the Discussion and 5) updated the final conclusion to finish on strengths and limitations in a balanced manner.

- 1) We acknowledge that the introduction ends positively and was lacking the dimension of challenges. Thus, we have updated the final paragraph of the introduction to the following:

In summary, we provide guidance on how to design drug response studies with longitudinal medication prescriptions and biomarker measures stemming from real-world data, introduce a more reliable model for studying genetic associations with drug response and present a comprehensive resource on the genetic architecture of cardiometabolic drug response. Our study showcases the value as well as the challenges when analyzing EHR-coupled biobanks to study inter-individual variability in drug response and identify clinically relevant genetic predictors.

- 2) Furthermore, we have dedicated a new result section to the challenges of using EHR data and how filtering strategies can partially overcome these (Result section: *The impact of filtering in the PGx-EHR study design*). Please see our detailed answers to comment 10 and 11.

- 3) In the Discussion we added the following paragraph to discuss the challenges in more detail:

Unlike RCT data, EHR data does not typically contain medication start along with baseline and post-treatment measures as direct readouts and in this study, we propose analysis strategies to identify new drug users with corresponding biomarker measures. We tested two filtering options, a stringent and a lenient one, with the differences being the extension of the baseline and post-treatment time periods, the main medication acting as add-on therapy (e.g. metformin acting as add-on to sulfonylureas treatment), relaxed prescription regularity requirement and the allowance of medication dose changes. The most important difference between the stringent and lenient filtering settings was the resulting sample size which doubled for statin and tripled for antihypertensive cohorts when relaxing the filtering criteria (Table S4). As a result of lower statistical power, the stringent filtering strategy only identified 4/10 PGx loci at a genome-wide significance level. We acknowledge that relaxed criteria, such as a longer time between the post-treatment measure and medication start, as well as less frequent prescriptions, were associated with smaller biomarker differences, suggesting that these variables are linked to drug adherence. Nonetheless, PGx signals identified in the lenient filtering scenario were confirmed as genuine, consistent with findings from the literature (e.g., SLCO1B1) and control analyses in statin-free individuals. Low drug adherence could potentially result in capturing the genetics of underlying health and behavioral aspects associated with adherence, although a previous study could not identify such signals even at a sample size of 116,439 for statins (Cordioli et al., 2023).

- 4) We have updated the limitation section in the Discussion to discuss these challenges in more detail and provide an outlook as to data availability and further analyses that could be conducted with EHR data (see comment 15).

- 5) Finally, we modified the conclusion paragraph of the Discussion to mention not only opportunities, but also limitations when using EHRs for drug response analyses:

To conclude, we show that EHRs enable new opportunities to study the genetics of drug response at scale and in a cost-effective manner. Although challenges remain with respect to the completeness and frequency of medication prescriptions and biomarker measures, these data allow us to shed light on the complex contribution of genetic and environmental components to drug efficacy. We find that the influence of common and rare genetic variants on drug response is relatively low, [...]

18) Similarly, the authors don't seem to acknowledge that there may still be some benefits of RCTs for PGX, if large enough RCT sample sizes could be available. Provided that future researchers perform PGX analyses of RCTs using appropriate statistical models, please could the authors discuss what benefits RCTs may still have over biobanks. For example, if there existed a biobank-based dataset and an RCT-based dataset of equal N sample sizes, would the cleaner phenotyping and study design of the RCT actually still be preferable? There seems to be a balance of EHR-biobanks with larger N vs RCTs with smaller N.

We acknowledge that the manuscript lacks a comparison with RCT data and the mentioning of the benefits of RCT over EHR data. We have included a limitation in the Discussion to clearly mention the benefits of RCT data over EHR data (also see comment 15):

Third, and related to the previous limitation, most participants have follow-up measures after six months (Figure S5; Table S4) which limits the study of immediate pharmacological effects, and measured drug responses may harbour a disease progression component. Frequency of follow-up measures constitutes a major limitation compared to RCT data where more frequent and broad biomarker measures are likely available.

With regards to other benefits of RCT data, these are now emphasized throughout the Discussion:

- 1) The cleaner phenotyping and the contrast between RCTs and EHRs is discussed in more detail: *Unlike RCT data, EHR data does not typically contain medication start along with baseline and post-treatment measures as direct readouts and in this study [...].* Later, contamination of drug response phenotypes is discussed as the first limitation in the Discussion: *First, we rely on data from EHRs to derive before and after treatment biomarker levels, and thus cannot exclude the possibility that individuals were already on medication before the first recorded prescription. [...]*
- 2) Randomization of drug users and controls which constitutes another advantage of RCTs is mentioned as the last limitation: *Finally, we rely on observational data to draw conclusions about drug efficacy. [...], control and medication groups were not defined randomly and by definition [...]*

For a fair balance between advantages and disadvantages of RCT over EHR data, we now also mention the high cost of RCTs in the Discussion:

We compared the PGx effects reported in the literature to the ones obtained from the EHR study design to assess the value of using EHRs to derive drug response phenotypes. [...] Although we did not identify PGx loci beyond what has been previously reported, the comparison with the literature confirmed that EHRs serve as a valuable data resource next to RCT and observational data for PGx study designs. Importantly, with the median cost estimated to be 409 USD per RCT participant (Speich et al., 2018), our study on 41,732 UKBB participants linked to their EHRs would cost 17 million, evidencing the cost effectiveness of this resource to study PGx at scale.

19) Furthermore, the Discussion ends by stating that “larger sample sizes will be needed”, so please could the authors discuss whether, considering the limitations and challenges already outlined, whether these larger sample sizes will ever actually be realizable?

We have updated the conclusion sentence to provide more concrete avenues on how we see the future with respect to larger sample sizes (also see our answer to comment 7 of how we envision a future meta-analysis):

[...] We find that the influence of common and rare genetic variants on drug response is relatively low, and larger sample sizes achieved by combining drug response GWAS from observational, EHR and RCT data as well as studies in more diverse ancestries will be needed to capture the full extent.

20) Having discussed the bias from baseline adjustment, please could the authors discuss whether this bias would be as strong in RCTs, as shown here in the biobank PGX work.

The bias resulting from baseline adjustment depends on the regression coefficient strength of the biomarker change regressed on the baseline level. Thus, the bias is independent of the nature of the data (RCT, observational data, EHRs). Please see our detailed answer to comment 2 where we discuss the relationship between baseline levels and biomarker change in more detail.

21) After considering both the lenient and stringent filtering scenarios, please could the authors discuss what their study design and QC recommendations would be for biobanks for future PGX researchers. Clearly it is a trade-off vs sample-size. But more discussions on specific recommendations from authors would be valuable for readers. For example, the fact that the APOB locus was only significant in the stringent filtering analysis rather than the lenient filtering analysis, I feel that this suggests that the stricter phenotyping pipeline may actually be optimal, given a large enough N...even though the authors here have to use the lenient filtering analysis as their primary analysis for reporting, due to limitations on N.

We acknowledge that the study design and QC recommendations deserve more discussion and have added a new paragraph to the Discussion which covers this topic in more detail. In the revised analyses, APOB reached genome-wide significance with the lenient filtering strategy and we no longer observe a gene reaching genome-wide significance solely in the stringent filtering. The new paragraph discussing study design and QC recommendations in the Discussion reads as follows (please also see our answer to comment 17 which covers the study design part in more detail):

When using EHR data, medication start, baseline and post-treatment measures are not readily available[...].

Furthermore, we tested whether there is a benefit in averaging over multiple baseline and post-treatment measures if available. Averaging over multiple measures should result in a more robust measure of the underlying biomarker level and reduce the chance of picking up regression-to-the-mean effects as we discuss in the following paragraph. The only difference we observed between these two strategies was that SLCO1B1 reached genome-wide significance when averaging across multiple measures (p-value of 1.10e-08 compared to 1.49e-05). Taken together, we recommend averaging across multiple measures when available and adopting the stringent filtering criteria when using EHRs for drug response studies, as this approach more closely mimics RCT designs. However, if data availability limits the sample size, we recommend using more lenient criteria while incorporating parameters

that best approximate drug adherence and account for concomitant medications as covariates, as this approach proved to significantly improve statistical power.

22) In the PRS results text, results are only stated within the text for SBP response to ACEi treatment. Please ensure that the authors have conducted analyses for BP response to all other drugs that they ran GWAS for, and that all analyses are commented on within the Results text.

Analyses have been conducted for BP response to all other drugs and these results are now commented on within the results text. Please see our detailed answer to comment 23.

23) Please clarify whether the PRS analyses are considered for both the absolute and relative types of phenotypes, for all drug-type PGS, e.g. in the examples from lines 225 onwards, which were only nominally significant. Furthermore, I felt here like the reader would benefit from a brief reminder of the 2 different types of phenotype “absolute” and “relative”, as lines 216-224 are quite complex. Also, the results focuses on a comparison of whether with vs without baseline adjustment is better. But, with regards to other comparisons being made, it is not always clear which of the absolute vs relative phenotypes is being used in the analyses, within the Results text. Or perhaps the authors have not always performed analyses for both, in which case they should. For example, lines 225-236 gives a result for b_abs for the absolute phenotype, but never for b_rel for the relative phenotype.

We have made changes to the text of the result section “Polygenic risk scores as predictors of drug response” to improve clarity. All PRS analyses were conducted for both absolute and relative phenotypes, and for each drug-biomarker pair. The full numerical results are in Table S16 which we now mention more explicitly.

We assessed whether high PRS of the underlying biomarker contribute to increased or decreased biomarker reductions in medication users of the UKBB. High LDL-C PRS resulted in an increased absolute, albeit lower relative LDL-C reduction following statin treatment ($b_{abs} = -0.092$ mmol/L/SD PRS, p -value = $5.84e-48$ and $b_{rel} = 2.47\%$ /SD PRS, p -value = $4.22e-34$; Figure 4a; Table S16). Thus, individuals with a higher genetic predisposition to elevated LDL-C levels are more likely to experience a larger drop, however, relative to their starting level this change is smaller than in those with a lower genetic predisposition. These opposing effects of high PRS on absolute and relative drug efficacy were also reflected in the genetic correlations of drug response traits with baseline traits. While the absolute LDL-C genetic difference was negatively correlated to LDL-C baseline levels ($r_g = -1.14$, $95\%CI = [-1.54, -0.74]$), the point estimate of the genetic correlation with the relative LDL-C difference was positive, although not significant ($r_g = 0.146$, $95\%CI = [-0.10, 0.39]$; Table S17). These analyses also suggest that in case of statin response and LDL-C as readout, the absolute LDL-C change is closer linked to the baseline LDL genetics. Association results between TC PRS and TC response to statins were highly significant (p -value < $2.51e-19$) and directionally concordant with LDL-C results. Nominally significant results between biomarker PRS and drug response phenotypes were found for high HbA1c PRS decreasing relative change following metformin treatment ($b_{rel} = 1.00\%$ /SD PRS, p -value = $4.43e-03$) and high SBP PRS increasing SBP reduction following ACEi and treatment to all antihypertensives combined (ACEi: $b_{abs} = -0.56$ mmHg/SD PRS, p -value = 0.014 , all: $b_{abs} = -0.46$ mmHg/SD PRS, p -value = 0.011). No significant effects of PRS on drug response were observed for the remaining antihypertensives, nor for beta blockers on HR (complete results in Table S16).

24) Even though I commend the authors on the thorough sensitivity analyses, the Results text becomes very busy, so caution to the authors that there are so many different comparisons being considered within the paper.

We acknowledge that there are a considerable amount of sensitivity analyses and that this can result in confusion. We have made changes to the result section and thoroughly updated the discussion to redirect the focus on the most important results and conclusions. We would also like to thank the reviewer for pointing out paragraphs that were particularly complex and that benefitted from a clearer rephrasing such as the result section on PRS (Comment 22 and 23).

25) Lines 222-223, please could the authors discuss an explanation as to why the genetic correlation is significant for the absolute phenotype, but not for the relative phenotype.

We have added that it seems that baseline genetics of LDL-C are more closely linked to the genetics of the absolute biomarker difference upon treatment. Please see our answer to comment 23 for the updated text.

26) Line 382: The covariate for “time between medication start and post-treatment measure” would be interesting to discuss in more detail, because this is a covariate that would not be needed in most RCT-based PGX.

In the revised manuscript, we discuss the covariate “time between medication start and post-treatment measure” in more detail as follows:

Result section: “The impact of filtering in the PGx-EHR study design”

When testing the impact of the time to first post-treatment measure on the biomarker difference, we observed that increased follow-up times resulted in reduced biomarker differences, even in the stringent filtering scenario where this parameter could not vary as widely (the follow-up time explained up to 1.1% of the variance, Table S6).

Discussion:

As a result of lower statistical power, the stringent filtering strategy only identified 4/10 PGx loci at a genome-wide significance level. We acknowledge that relaxed criteria, such as a longer time between the post-treatment measure and medication start, as well as less frequent prescriptions, were associated with smaller biomarker differences, suggesting that these variables are linked to drug adherence. Nonetheless, PGx signals identified [...]

Please also see our detailed answers to comment 11 and 17 related to the results and discussion of challenges associated with EHR data.

27) Line 193 suggests that there may be several different types of statins and doses. I couldn't find anywhere a clear description of what all these different types of statins and ranges of different doses are within the analysed data. If I have missed it, please refer to such a Sup Table of information here; or if not, please provide detailed information on this.

For statins, antihypertensives and betablockers, the different types with the number of people are listed in Table S4 “Drug response cohort study characteristics in the UK Biobank”. However, it is true that the ranges of doses were missing in the Supplementary Tables. We have now added a Supplementary Table S5 with all drug-dose combinations and

number of users. This table is now referenced at the previous Line 193 (*To this end, we compare genetic effect sizes of biomarker differences in medication-naive controls to those in statin users (simvastatin 40mg users who represent the largest starting statin type-dose group; Table S5*), as well as in Methods section “Study design and drug response phenotypes” (see comment 30).

28) Lines 324-5: please provide more detail for the BP drugs. Linking to lines 345-7, please clarify more clearly if analyses are restricted to monotherapy.

We have provided more details about the different antihypertensives and beta blockers as well as the dose ranges (please see our answer to the previous comment 27).

Furthermore, we have clarified whether analyses are restricted to monotherapy (please see our answer to comment 30).

29) Lines 354-6: please clarify if this is the case for both the lenient and stringent filtering criteria?

Grouped with comment 30.

30) Lines 362-4, the distinction between these 2 different scenarios is not written clearly enough.

Both comments 29 and 30 refer to the method section where we describe how medication regimens were defined, Method section: *Study design and drug response phenotypes*. We have reworked the section to expand the concepts about related medication prior starting the primary medication (comment 29) and concomitant medications (comment 28 and 30) and provided more information about treatment changes and prescription regularity that was previously only mentioned in the Supplementary information and tables.

To determine medication regimens (medication start, treatment changes, prescription regularity), we first extracted all available prescriptions for each broader medication class (lipid-regulating, antidiabetic including insulin, and antihypertensives; BNF and Read V2 codes in Table S2; when BNF codes were truncated to miss the drug ingredient, we extracted them by matching drug names and brand names in the drug description). We then selected individuals with entries of the medication of interest (primary medication) and omitted individuals taking medications other than the primary medication of the same class within a year of initiating the primary medication (in the stringent filtering setting, only monotherapies within the same medication class were allowed and no prior related medication prescriptions were permitted). This criterium was relaxed in the lenient filtering setting where the primary medication could act as add-on therapy in certain scenarios, with the related medication included as a covariate (Table S3). More specifically, the following concomitant medication regimens were allowed in the lenient filtering setting:

- 1) Statins: Non-statin antilipemic medication prescriptions (e.g. fenofibrates) prior to statin prescriptions were permitted and included as covariate.*
- 2) Metformin: Prior sulfonylureas prescriptions were permitted (covariate) as well as metformin acting as add-on to sulfonylureas (additional covariate, which was defined by recorded sulfonylureas prescriptions during baseline and post-treatment periods). Individuals with sulfonylureas prescriptions recorded only during the post-treatment period (and not baseline period) were filtered out.*

- 3) *First-line antihypertensives: Antihypertensives were allowed to act as add-on to beta blockers (covariate) and loop diuretics (additional covariate) if prescriptions were consistently recorded during baseline and post-treatment periods as illustrated in Figure S2.*
- 4) *Beta blockers: in the analysis with SBP we excluded individuals taking any other antihypertensives except for loop diuretics (covariate, same as described with first-line antihypertensives). In both the stringent and lenient analysis with HR, prescriptions of antihypertensives other than beta blockers were permitted since their effect on HR is much weaker than for beta blockers (Materson et al., 1998).*

Note, that in both stringent and lenient filtering settings, individuals taking primary medications in combination with a medication of the same class (e.g. statins in combination with ezetimibe) were filtered out (Note S3). Prescriptions during the post-treatment period of medications of the same class as the primary medication (except for the allowed concomitant medication regimens in the lenient filtering setting described above) were defined as treatment changes, and accordingly, individuals were excluded from the respective drug cohort. A dose change, with dose information retrieved from the drug description using regular expressions (McInnes et al., 2020), was defined as treatment change in the stringent, but not in the lenient filtering setting where the average over all prescriptions in the post-treatment period was used. Note that dosage information is not readily available, and we assume the dose information, available in the description of the medication, to correspond to the daily dosage. Since a given primary medication as defined herein can comprise multiple drugs (e.g. simvastatin and atorvastatin for statins, ramipril and lisinopril for ACEi), we only included drugs taken by at least 20 individuals (both in the stringent and lenient filtering setting). Finally, we defined a prescription regularity parameter by the presence of a prescription at least every two months for the duration of the post-treatment period with the completeness being 100% if this was the case (we required a completeness of 60% and 30% in the stringent and lenient filtering setting, respectively). In Table S5, we show for each drug response cohort the number of individuals per drug type and dose which constituted covariates in all drug response analyses.

31) Lines 375-6 about “proxying drug adherence”: please explain more clearly what you mean here?

We have expanded this notion as follows:

The different QC steps were as follows: i) [...], v) regular prescriptions proxying drug adherence under the assumption that skipped prescriptions are the consequence of inconsistent medication intake, vi) [...]

32) Lines 239-242: please explain this in more detail, so the concepts here are clearer.

We have added further explanations as to why adjusting baseline levels for PRS can proxy the environmental/lifestyle component of a biomarker level:

Although PRS can serve as a predictor of drug response, starting baseline levels remain the best predictor of post-treatment levels (Figure 4b-c). To disentangle the effect of genetics and environment on drug response, we adjusted baseline levels for PRS. Since baseline biomarker levels comprise both a genetic and environmental/lifestyle component (Figure 3a), the resulting PRS-adjusted baseline levels should more closely reflect the environmental component only. As a consequence, individuals with high PRS and moderately high initial biomarker levels will have lower PRS-adjusted baseline levels than individuals with same initial biomarker levels and low PRS. PRS-adjusted baseline levels explained 42.2% and 11.3% of the variance of absolute and relative LDL-C difference, respectively. [...]

33) Lines 275-280: Please give more examples to illustrate this further.

We have expanded the discussion on absolute and relative biomarker change to further illustrate that linear and non-linear pharmacogenetic interactions require further testing depending on the context. Please see our detailed answer to comment 14.

34) Line 272: should the word be “can” or “will” in the sentence “...showed how baseline adjustment can induce biases...”

We changed the word « can » to « will »:

As similar study designs have used differing GWAS models to estimate pharmacogenetic effects, we developed a realistic model for biomarker differences and showed how baseline adjustment will induce biases for genetic variants associated with baseline levels.

35) Please clarify in the Results section, that the PRS analyses (I believe) are within UKB only, not investigated within AOU data too. (Lines 416-8 explained which analyses the replication analyses were restricted to, but it does not say if PRS analyses were validated in AOU too, although line 454 suggests not, only stating that PRS were calculated for UKBB participants...but this needs clarity...)

Yes, the PRS analyses are within UKB only and were not investigated within AOU data. We have updated the first sentence of the result section “Polygenic risk scores as predictors of drug response” as follows:

We assessed whether high PRS of the underlying biomarker contribute increased or decreased biomarker reductions in medication users of the UKBB. [...]

36) Line 282 “in line with other studies”, please provide the references.

We have updated the beginning of the paragraph that discusses PRS and included additional literature references. The beginning of this paragraph reads now as follows:

We found that high PRS of the underlying biomarker can lead to increased absolute, although lower relative biomarker reductions, further contributing to the growing body of literature that links PRS to treatment effectiveness (Mega et al., 2015, Natarajan et al., 2017, Zhang et al., 2019, Damask et al., 2020, Marston et al., 2020, Li et al., 2021, Tapela et al., 2022, Kiiskinen et al., 2023, Turkmen et al., 2024). A recent study showed that sulfonylureas therapy was more effective in participants with higher T2D PRS with findings replicated in a separate cohort (Li et al., 2021). Other studies found that high schizophrenia PRS reduced antipsychotic efficacy (Zhang et al., 2019) and similarly high LDL-C and SBP PRS were associated with uncontrolled hypercholesterolaemia and hypertension, respectively (Tapela et al., 2022). Additionally, high SBP PRS have been linked to lower rates of discontinuing hypertension medication (Kiiskinen et al., 2023, Turkmen et al., 2024). Using RCT data, high coronary heart disease (CHD) genetic risk was found to be associated with increased CHD risk, although the comparison between controls and treated participants revealed that relative risk reductions were higher among individuals with a high PRS, suggesting that this group benefited the most from lipid-lowering therapy (Mega et al., 2015, Natarajan et al., 2017, Damask et al., 2020, Marston et al., 2020). [...]

37) Line 75 is written strangely, suggesting that HDL-C is an exception, with biomarker

change levels in the opposite direction, when actually this is as expected and not of concern or surprising, because increased HDL-C is a good thing.

We changed the phrasing of this sentence to leave out the word “except” which could suggest that the direction of HDL-C change is unexpected. The sentence reads now as follows:

Following cardiometabolic drug treatment, lipid, HbA1c and blood pressure levels significantly dropped while HDL-C levels moderately increased (Δ HDL-C = 0.014 mmol/L, two-sided paired t-test p-value = 1.49e-24; Figure 2a; Table S4).

38) Lines 134-5: this should state “or both”, for example, APOE is discussed in an example later, suggesting that this does have both baseline genetic and PGX effects.

We thank the reviewer for pointing this out and added “both” to the sentence:

Thus, these reported loci could represent either baseline genetic or pharmacogenetic effects, or both.

39) Please state in the Abstract: (i) that analyses are of EUR ancestry only; (ii) the separate N sample sizes for UKB vs AOU.

We have updated the abstract to reflect the results of our revised analyses and have stated that analyses are of EUR ancestry only and state the separate sample sizes for UKB and AOU.

Electronic health records (EHRs) coupled with large-scale biobanks offer great promises to unravel the genetic underpinnings of treatment efficacy. However, medication-induced biomarker trajectories stemming from such records remain poorly studied. Here, we extract clinical and medication prescription data from EHRs and conduct GWAS and rare variant burden tests in the UK Biobank (discovery) and the All of Us program (replication) on ten cardiometabolic drug response outcomes including lipid response to statins, HbA1c response to metformin and blood pressure response to antihypertensives (N = 932-28,880). Our discovery analyses in participants of European ancestry recover previously reported pharmacogenetic signals at genome-wide significance level (APOE, LPA and SLCO1B1) and a novel rare variant association in GIMAP5 with HbA1c response to metformin. Importantly, these associations are treatment-specific and not associated with biomarker progression in medication-naïve individuals. We also found polygenic risk scores to predict drug response, though they explained less than 2% of the variance. In summary, we present an EHR-based framework to study the genetics of drug response and systematically investigated the common and rare pharmacogenetic contribution to cardiometabolic drug response phenotypes in 41,732 UK Biobank and 14,277 All of Us participants.

40) The “Furthermore” sentence towards the end of the abstract, describing the different results for “absolute” vs “relative” biomarker reduction, is quite complex, when stand-alone in the Abstract, without these different phenotypes being clearly defined.

We agree with the reviewer that this sentence was quite complex and we have removed the different PRS results for absolute and relative biomarker reduction. Please see the updated abstract in comment 39.

41) In the opening sentences of the Results, please re-state, as from the Methods, the N of UKB subjects with primary care data available. (It is strange otherwise, at the bottom of the first Results pg, that N is provided for AOU replication data, but not yet for UKB.)

We have now stated this number in the first sentence of the first Result section “Overview of the analysis” (please also see our answer to comment 16).

In the drug response discovery analyses, we extracted longitudinal prescription and response biomarker data from the UKBB primary care records which we combined with phenotypic data from the assessment visits (currently ~230,000 (45%) UKBB participants are linked to their EHRs).

42) In Table 1, for clarity, please also name “UKB” and “AOU” in the column headers, as well as the “discovery” and “replication” labels.

We have changed the column headers of Table 1 to add “UKBB (discovery)” and “AoU (replication)”.

43) Figure 3: The LH-side diagram in part (a) may be quite complex for some readers of this broad readership journal; the examples in Figs 3b-d would still benefit from more detailed explanations, beyond what is already provided on pg.15. I am not surprised that the Fig 3 legend is so long, when this is quite a complex Figure, with so many nested multi-panel graphs, overall showing so much.

We have updated the LH-side diagram of panel (a) in Figure 3 to identify the three components of biomarker levels more clearly through colored shading (baseline genetics, environment, drug treatment). The panel now looks as follows:

Figure 3. Modelling longitudinal changes of biomarker levels with (or without) treatment effect. **a** Biomarker levels Y at time t can be influenced by baseline genetics G_0 (orange), environment E and gene-environment interactions ($G_E * E$, blue), and drug status D and pharmacogenetic interactions ($G_D * D$, purple). Drug response phenotypes modelled as the difference of post-treatment (t_1) and baseline (t_0) levels allow the estimation of the pharmacogenetic effect γ_D through genetic regression analyses (Note S1).

Furthermore, we have provided more explanations to the result section which covers the figure:

In Figure 3b-d, we depict genetic variants that either have a significant pharmacogenetic effect γ_D , baseline effect β_0 or both, and showcase how baseline adjustment can introduce a bias in genetic effect estimation. To this end, we compare genetic effect sizes of biomarker differences in medication-naïve controls to those in statin users (simvastatin 40mg users who represent the largest starting statin type-dose group, Table S5). The SNP rs7412 in the APOE region which is strongly associated to baseline levels ($\beta_0 = 0.634$, p -value $< 1e-300$) also

exhibits a pharmacogenetic effect ($\gamma_D = 0.284$, $p\text{-value} = 3.52e-19$) while not being associated to longitudinal change in statin-free controls ($p\text{-value} = 0.59$; Figure 3b; Table S7). However, upon baseline adjustment, a significant genetic effect with longitudinal change is observed in drug naive individuals ($b = 0.256$, $p\text{-value} = 4.83e-102$) as well as a stronger association in statin users due to the β_0^2 bias ($b = 0.409$, $p\text{-value} = 1.07e-39$). Since this SNP is strongly associated with baseline levels, its effect on drug response is overestimated when adjusting for baseline and a spurious association is observed for longitudinal change without lipid-lowering treatment. The *SLCO1B1* missense variant rs4149056 was not associated with total cholesterol baseline levels ($p\text{-value} = 0.078$), and genetic effects remained similar between baseline adjusted and unadjusted results (adjusted: $\gamma_D = 0.092$, $p\text{-value} = 3.16e-08$; unadjusted: $\gamma_D = 0.092$, $p\text{-value} = 1.18e-07$; Figure 3c), evidencing the sole implication of *SLCO1B1* in pharmacokinetics (no significant association was found for longitudinal change either; $p\text{-value} > 0.24$). In contrast, the SNP rs11076175 in the *CETP* locus is strongly associated with HDL-C baseline levels ($\beta_0 = 0.262$, $p\text{-value} = 4.26e-311$), but had no significant pharmacogenetic effect in the unbiased model ($\gamma_D = 0.006$, $p\text{-value} = 0.70$). Upon baseline adjustment, strong associations with both longitudinal change and drug response were observed ($p\text{-values}$ of $4.85e-33$ and $7.96e-06$, respectively; Figure 3d). Together, these examples illustrate how the genetic component of baseline levels can result in the identification of false positive associations and/or overestimation of pharmacogenetic effects.

44) Figure 4(a): Please label the two side-by-side plots are “absolute” vs “relative” phenotypes, i.e. always labelling the figures with the same consistency to help the reader. Please also explain the three different colours.

We thank the reviewer for making this point. We believe that the label change to “absolute” and “relative” is a subjective and think that “post-base” and “log(post)-log(base)” is more explicit. For consistency, we left the titles of the side-by-side plots to read “post-base” for the absolute biomarker difference and “log(post)-log(base)” for the relative biomarker difference since in Figure 1 and 2 absolute and relative biomarker differences are defined by these same labels. However, the legend provides explicit explanations for the “*absolute (post-base) and logarithmic relative (log(post)-log(base)) biomarker difference*” and we added explanations for the colors:

Figure 4. [...] a) Drug response associations with PRS calculated for the absolute (post-base) and logarithmic relative (log(post)-log(base)) biomarker difference colour-coded by the drug (blue: statins, orange: metformin, green: antihypertensives, pink: beta blockers).

45) Line 467 has a typo within “in individuals part of the primary care data”

Unfortunately, we could not spot the typo in this first sentence of the Method section “Longitudinal biomarker change GWAS”. In the GWAS on medication-naïve controls, we use solely individuals of the UK Biobank who are linked to the primary care data to make sure we have comprehensive information about medication prescriptions and thus their medication regimen. To avoid confusion, we have updated the Method section *Longitudinal biomarker change GWAS in controls* to detail how controls in the longitudinal biomarker change GWAS were selected:

We conducted biomarker change GWAS in control individuals that were part of the primary care data and that did not have any drug prescription indicated for the investigated disease/surrogate endpoint : controls in the lipid-change GWAS had no lipid-lowering medications, those in HbA1c-change GWAS had no antidiabetic medications, those in blood pressure-change GWAS had no antihypertensives, and those in heart rate-change GWAS

had no beta-blockers (i.e., broad medication class, Table S2). [...]

Reviewer #2 (Remarks to the Author):

In this large-scale biobanks coupled with electronic health records (EHRs) pharmacogenetic study, authors analyzed both common and rare variants, providing a more comprehensive view of genetic influences on drug response. Through the analyses, they were able to replicate previous findings and also identify new signals such as APOB and variants associated with lipid response to statins near LDLR and ZNF800 genes. Two large biobanks UKBB and AOU were used for discovery and replication analyses which is thorough and the analyses conducted (GWAS, EXWAS and PRS) were conducted to support the conclusions. The authors have used appropriate statistical methods and have been transparent about the limitations of their study. However, some points that might benefit from further clarification or discussion include:

We thank the reviewer for the positive assessment of our work and the constructive feedback. We are grateful for the reviewer's comments aimed at strengthening our work, such as the suggestions of adding power analyses, analyzing the findings in diverse ancestral groups, and expanding the discussions on the employed statistical model, drug adherence and PRS. Please find our answers to the comments below.

1. One of the main drawback of using EHR data for PGx analyses is the potential impact of confounding factors such as medication adherence and lifestyle changes. Authors should discuss how non-adherence might bias the results and propose methods to mitigate this in future studies.

We thank the reviewer for raising this point. We have added a new result section "*The impact of filtering in the PGx-EHR study design*" where we describe the impact of filtering strategies on the derived drug response phenotypes and analyze the component of medication adherence in more detail. At the introductory paragraph of this new Result section we explain under which assumptions we define drug adherence in this study and then present the impact on drug response as follows:

In EHRs, medication start, baseline and post-treatment measures are not readily available, and we tested multiple strategies to extract drug response phenotypes and assess their impact on PGx associations. We introduced a prescription regularity parameter to proxy drug adherence under the assumption that skipped prescriptions are the result of inconsistent medication intake. This prescription regularity parameter (or prescription completeness) is defined by the presence of a prescription at least every two months for the duration of the post-treatment period and a completeness of 100% is obtained if this is the case. To account for varying time window thresholds, prescription regularities, changes in medication regimens, and single or multiple biomarker measures at baseline and post-treatment, we defined stringent and lenient filtering strategies, assessing the impact of using single versus multiple biomarker measures for each.

A main difference between the lenient and stringent filtering scenario was the extension of the baseline and post-treatment periods. [...] When testing the impact of the time to first post-treatment measure on the biomarker difference, we observed that increased follow-up times resulted in reduced biomarker differences, even in the stringent filtering scenario where this parameter could not vary as widely (the follow-up time explained up to 1.1% of the variance, Table S6). Likewise, a lower prescription completeness value resulted in decreased biomarker differences in the stringent and the lenient filtering settings, where the required completeness were 60% and 30%, respectively (the prescription regularity explained up to 7.6% of the variance, Table S6). Despite the differences in the stringent and lenient filtering scenarios,

average baseline and post-treatment biomarker levels were almost identical between the resulting drug cohorts (differences of less than 3%; Table S4).

In the Discussion, we elaborate on how drug adherence impacts the study design and discuss its implications in the context of a similar study that has been conducted in the field (Cordioli et al., 2023, PMID: 39443518). Their study supports the conclusion that the risk of identifying false positives is very low or even non-existent. In contrast, non-adherence may introduce a downward bias since the observed biomarker difference become noisier. The corresponding Discussion paragraph reads as follows:

Unlike RCT data, EHR data does not typically contain medication start along with baseline and post-treatment measures as direct readouts and in this study, we propose analysis strategies to identify new drug users with corresponding biomarker measures. [...] We acknowledge that relaxed criteria, such as a longer time between the post-treatment measure and medication start, as well as less frequent prescriptions, were associated with smaller biomarker differences, suggesting that these variables are linked to drug adherence. Nonetheless, PGx signals identified in the lenient filtering scenario were confirmed as genuine, consistent with findings from the literature (e.g., SLCO1B1) and control analyses in statin-free individuals. Low drug adherence could potentially result in capturing the genetics of underlying health and behavioral aspects associated with adherence, although a previous study could not identify such signals even at a sample size of 116,439 for statins (Cordioli et al., 2023).

While indeed lifestyle changes coupled with medication can impact the assessed biomarker changes, the genetic basis of such precise behavioural traits is expected to be negligible, thus, could not noticeably bias our association results.

2. Use directed acyclic graphs (DAGs) to visually represent and explain the identified causal relationships.

We thank the reviewer for this suggestion. In addition to Figure 3a which shows a general graph for modelling biomarker levels at a given time and drug status, we have added a DAG specific to the drug response scenario (two time points, the second time point with the drug status equaling 1). We believe that this graph will help the reader in visualizing the identified causal relationships. This graph is introduced in the Result section *Modelling drug response and longitudinal change phenotypes* and reads as follows:

When modelling the drug response as the difference of post-treatment levels Y_{t1} and baseline levels Y_{t0} , where the drug status is 1 and 0 at t_1 and t_0 , respectively, the drug response phenotype simplifies to (Note S1):

$$\Delta Y = Y_{t1} - Y_{t0} = \beta_D + \gamma_D \cdot G_D + \delta_{01}$$

Thus, the pharmacogenetic effect can be estimated from genetic regression analyses on the biomarker difference at post-treatment and baseline. Control individuals who do not take any related medications are not required for this estimation, however, analyzing longitudinal changes in these individuals (the drug status being zero at both time points) serves as a control to ensure that identified PGx signals are specific to drug treatment. Figure S12 presents the directed acyclic graph (DAG) that extends the graph in Figure 3a for modeling the genetics of drug response phenotypes ΔY . This expression also holds when modelling the logarithm of the biomarker level. [...]

Figure S12. Directed acyclic graph (DAG) to model the genetics of drug response based on biomarker levels before and after drug treatment. This figure accompanies Figure 3 in the main text. Biomarker levels Y at time t can be influenced by baseline genetics G_0 , environment E and gene-environment interactions ($G_E \cdot E$), and drug status D and pharmacogenetic interactions ($G_D \cdot D$; Figure 3). Drug response phenotypes ΔY modelled as the difference of post-treatment ($t = t; +1$) and baseline ($t = 0; -1$) levels allow the estimation of the pharmacogenetic effect γ_D (purple). The DAG illustrates how baseline genetics cancels out when taking the difference of biomarker levels, whereas the pharmacogenetic effect γ_D only acts through the post-treatment pathway.

3. Although the non-European analyses were on smaller sample sizes, authors should perform a power analysis to determine the sample size needed for adequately powered studies in other ancestral groups and discuss these limitations.

This comment will be answered together with comment 4.

4. Additionally, comparing the allele frequencies and effect sizes of the identified variants across different populations using publicly available databases such as gnomAD may help understand these effects in other ancestral groups.

We thank the reviewer for raising the important subject of differing PGx allele frequencies in other ancestral groups and the implications of allele frequencies and sample size on statistical power. We have added a Table that compares the allele frequencies of all identified PGx variants across ancestral groups by consulting the gnomAD v4.1 resource. Based on these frequencies, we have conducted power analyses to determine the sample size needed for adequately powered studies.

We expanded the result section “Replication analysis in the All of Us research program” to include these results:

Participants in the AoU biobank represent a more diverse range of genetic ancestries compared to those in the UKBB, with only 50% being of European ancestry (The All of Us Research Program Genomics Investigators, 2024). When assessing allele frequencies of the identified PGx signals across ancestries by using the gnomAD v4.1.0 resource (Chen et al. 2024), the largest allele frequency differences were observed for APOB, ranging from 5.2% in East Asians to 52.2% in the Amish (Table S10). Significant differences were also found for

LDLR (0.8%-18.8%) and SLCO1B1 (3.1%-20.9%), while MAFs of other PGx variants varied by less than 10%. Based on this MAF spectrum, we conducted power analyses to determine the sample sizes required to detect PGx signals at genome-wide significance ($5e-08$) in different ancestral groups. For a strong signal, such as rs7412 at the APOE locus ($\beta = 0.232$, $p\text{-value} = 1.11e-28$), a sample size of 4,000 is needed to achieve 80% power in African/African Americans where the MAF is the highest (10.5%) compared to 28,500 in Amish people where the MAF is the lowest (1.2%). For a more modest signal, such as rs4149056 at the SLCO1B1 locus ($\beta = -0.063$, $p\text{-value} = 1.10e-08$), a sample size of 148,501 is needed for 80% power in African/African Americans where the MAF is the lowest (3.1%) compared to 27,501 in the Finnish where the MAF is the highest (20.9%; Figure S11).

The methodology of the power analysis is added to a new Method section: *Power analysis* as follows:

We conducted statistical power analyses to determine the sample size needed to detect PGx signals at genome-wide significance in different ancestral groups. Minor allele frequencies were obtained from the gnomAD v4.1.0 resource (Chen et al., 2024). We based the effect sizes on the effect sizes observed in the discovery analyses in the UKBB and calculated the required power as:

$$\text{power}(\alpha, NCP) = \Phi(NCP - c) + \Phi(-NCP - c)$$

where c was set to 5.45 which is the test statistic corresponding to a one-sided discovery α of $5e-08$.

The non-centrality parameter (NCP) was calculated as:

$$NCP = \sqrt{\frac{N}{n_{\text{obs}}}} \cdot \frac{\beta_{\text{obs}}}{\text{se}_{\text{obs}}} \cdot \sqrt{\frac{\text{MAF} \cdot (1 - \text{MAF})}{\text{MAF}_{\text{obs}} \cdot (1 - \text{MAF}_{\text{obs}})}}$$

where the sample size n_{obs} , observed effect size β_{obs} , standard error se_{obs} and minor allele frequency MAF_{obs} stem from the discovery analyses in the UKBB.

Figure S11. Power analysis across ancestral groups. Power analyses to determine the sample size needed to detect PGx signals at genome-wide significance in different ancestral groups where minor allele frequencies (MAFs) were obtained from the gnomAD v4.1.0 resource. The effect sizes for the APOE and SLCO1B1 pharmacogenetic signals are based on the effect sizes obtained in the discovery analyses in participants of European ancestry (Non-Finnish; EUR (NFE)) in the UK Biobank (UKBB). The dashed horizontal line indicates 80% power.

b = effect size; se = standard error; N = sample size; AFR = African/African American; AMI = Amish; AMR = Admixed American; ASJ = Ashkenazi Jewish; EAS = East Asian; FIN = Finnish; EUR (NFE) = Non-Finnish European; SAS = South Asian

Furthermore, we have added a limitation to the Discussion section to acknowledge that the main analyses were conducted in Europeans which may limit the identification of PGx variants specific to other ancestries.

Sixth, analyses have been conducted in individuals of European ancestry. While allele frequencies of identified PGx signals were largely consistent across genetic ancestry groups (Table S10), analyses in diverse ancestry groups are likely to identify PGx effects missed in individuals of European ancestry. For a PGx signal with modest effect size such as SLCO1B1, we estimated that a sample size of ~150,000 is required for 80% power in African/African Americans (MAF of 3.1% compared to 15% in Europeans), showcasing that even with cohorts the size of the current study increased numbers of PGx signals could be identified provided that the MAF is high.

Finally, we have updated the concluding paragraph in the Discussion to acknowledge the importance of diverse ancestries to complement an increase in sample size.

We find that the influence of common and rare genetic variants on drug response is relatively low, and larger sample sizes achieved by combining drug response GWAS from observational, EHR and RCT data as well as studies in more diverse ancestries will be needed to capture the full extent.

5. While the associations between PRS and drug responses are well-described, the paper could benefit from more discussion on the potential biological mechanisms underlying these relationships and to better understand why high PRS might lead to different responses.

We thank the reviewer for making this suggestion and we have extended the discussion on PRS and potential biological mechanisms underlying these associations. The new Discussion paragraph reads as follows:

The reason as to why PRS contribute to drug-induced biomarker response requires further investigation and we propose two hypotheses: one related to biology and the other to either of the employed models. Biological pathways involved in the drug perturbation may coincide with genes highly contributing to PRS which means that the drug may work better for people with high baseline value for genetic reasons. Therefore, drugs designed to target key genes involved in the disease are likely to be more beneficial for individuals for whom these genetic networks are at the root of the disease. The second, more likely hypothesis concerns a model misspecification. In the current PRS association model, we assume baseline levels to be linearly associated with the biomarker difference and analogous to genetic variants, baseline levels encompass PRS as the genetic component. Thus, when adjusting the biomarker difference for both baseline and PRS, spurious associations are obtained with PRS and we show on the example of LDL response to statin, that the association is in the opposite direction to the one without baseline adjustment. If biomarker differences truly depend on the genetic and non-genetic components of baseline values, baseline PRS and the non-genetic baseline parts are expected to correlate differently with baseline and biomarker change. Further investigations into deriving an environmental baseline component (either by approximating it by the residuals after PRS regression or explicitly identifying environmental correlates of baseline values), could provide more precise estimates on the relative contribution of genetic and environmental components on biomarker change. While we identify PRS as predictors of treatment efficacy, we also note that in our analyses PRS explained less than 2% of the variance in differential drug response.

6. Consider exploring potential interactions between different PRS or between PRS and other clinical factors that might influence drug response.

We agree with the reviewer that interactions between different PRS and other clinical factors are of interest, however, this falls outside the scope of this study, especially since the observed effects are very small. Interaction analyses would only be meaningful and clinically relevant if the observed magnitudes were higher.

7. While the focus of this study was on immediate biomarker changes, it would be interesting to explore whether these PGX and PRS associated differences in drug response translate to differences in long-term clinical outcomes (e.g., cardiovascular events for statins, diabetes complications for metformin).

We agree with the reviewer that investigating whether our conclusions on continuous biomarkers apply to long-term clinical outcomes is of great interest, especially, given RCT data on cardiovascular events showing increased statin efficacy in individuals with high PRS. If the causal effect of the biomarker on the disease can be accurately estimated, our PRS-to-biomarker effects could be extrapolated to a PRS-to-disease outcome. However, this investigation is outside the scope of our study, and accordingly we have updated this limitation in the Discussion as a direction for future work.

Fifth, our analysis focused on continuous biomarkers and not on clinical events. LDL-C, SBP, HR and HBA1c merely serve as surrogate endpoints of CHD and T2D events, and the genetic interplay with drug efficacy may be different when assessing hard clinical endpoints. EHRs provide a rich resource for conducting analyses similar to those done with RCT data (Mega et al., 2015, Natarajan et al., 2017, Damask et al., 2020, Marston et al., 2020), and for studying genetic interactions with drug efficacy on clinical events in real-world settings.

Reviewer #2 (Remarks on code availability):

There are no codes in the scripts folder. I also could not view the snakemake config file.

We apologize that the reviewer was not able to access the codes in the scripts folder. We have added these shortly after our submission to *Nature Communications*. We have updated the code on Github with the changes made during this revision and archived a stable release on Zenodo.

This is now stated in the *Code availability* as follows:

All codes used in this analysis are available on GitHub at <https://github.com/masadler/PGxEHR> a (<https://doi.org/10.5281/zenodo.14026836>).

Reviewer #3 (Remarks to the Author):

The manuscript "Levering large-scale biobank EHRs to enhance pharmacogenetics of cardiometabolic disease medications" investigates the genetics mechanisms of several pharmacotherapies using trajectories. This work is well positioned and stands to make a nice contribution to this area of the literature and the study visuals/tables are well designed, however, some organizational and clarity questions remain.

We thank the reviewer for the thorough assessment of our work and for the comments and suggestions aimed at improving the clarity of the manuscript, emphasizing the clinical relevance of this work, and expanding the discussion and comparison with literature. Please find the detailed answers to the comments below.

1. The manuscript would be strengthened by a reworking of the introduction to include a more thorough review of the current state of the literature and how this study enhances what remains unknown. The final paragraph introduces the investigation into biomarker changes, which is not set up in intro nor how the results of this study will be clinically informative. (The lengthy "overview of analysis" in the results section could be greatly simplified, as much of this is already covered in the methods, to make room for a revised & expanded intro)

We thank the reviewer for pointing out that important studies were missing from the introduction and that we failed to clearly describe how our work contributes to the field of pharmacogenetics. Our updated Introduction 1) introduces the investigation into biomarker changes early on, 2) adds important publications that we omitted previously, and 3) defines the scope of this study more clearly in the final paragraph. Regarding the final point, we note that the focus of this study is a thorough investigation into the benefits and challenges when using EHRs for cardiometabolic PGx research. While the outcome of this study is likely not of direct clinical relevance, we anticipate that our derived insights on using EHRs for PGx research inform translational studies in the future.

We would also like to point out that the "Overview of analysis" result section is a very frequent structure of the Nature Communications journal which allows readers to understand the results without consulting the method section beforehand which in Nature Communications figures at the end of the paper.

- 1) Investigation into biomarker changes is introduced in the first sentence of the second paragraph to introduce related work from the field:

Several PGx GWAS consortia have formed over the years to study the genetics of drug efficacy in larger sample sizes by investigating the change in biomarker levels following medication start. For instance, the Genomic Investigation of Statin Therapy (GIST) [...]

- 2) We added the studies conducted on the GERA cohort where lipid response to statins was investigated using EHRs and we emphasize that PGx studies that use EHRs at scale to study biomarker differences are lacking.

Using electronic health records (EHRs), the Genetic Epidemiology Research on Adult Health and Aging (GERA) cohort has additionally identified the APOB and SMARCA4/LDLR loci as genetic determinants of statin response (Oni-Orisan et al., 2018, Oni-Orisan et al., 2020). Similarly, the Metformin Genetics (MetGen) consortium [...]

Biobanks coupled with EHRs that comprise medication data provide new opportunities to discover PGx associations [...]. Yet, PGx biobank studies so far have either focused on known

pharmacogenes and their associations with adverse drug reactions, drug dosage and drug prescribing behaviour or analyzed the genetics of temporal medication use in isolation of disease phenotypes. Except for the GERA cohort which solely utilized EHRs for the study of LDL-C response to statins (Oni-Orisan et al., 2018, Oni-Orisan et al., 2020), the integration of longitudinal medication and phenotypic data to screen for genetic determinants of drug efficacy at a biobank scale remains largely unexplored.

3) We define the scope of this study more clearly in the final introduction paragraph

[...] In summary, we provide guidance on how to design drug response studies with longitudinal medication prescriptions and biomarker measures stemming from real-world data, introduce a more reliable model for studying genetic associations with drug response and present a comprehensive resource on the genetic architecture of cardiometabolic drug response. Our study showcases the value as well as the challenges when analyzing EHR-coupled biobanks to study inter-individual variability in drug response and identify clinically relevant genetic predictors.

2. As the manuscript currently reads, genetic underpinnings and pharmacogenetics seem to be conflated and interchangeable - clarity around this language and what the study hopes to achieve would improve the manuscript.

We have made changes to the text to avoid any confusion between genetic underpinnings and pharmacogenetics. Genetic underpinnings of baseline and pharmacogenetics are indeed conflated when we use the wrong genetic association model, i.e., when we add baseline levels as a covariate.

In addition to state the goals more clearly in the final introductory paragraph (see the point 3) in our response to comment 1 of defining the scope of the study more clearly), we have reworked the Result section *Modelling drug response and longitudinal change phenotypes* to clarify the distinction between the genetics of baseline, and how baseline adjustment can lead to spurious genetic associations with drug response when genetic variants are also associated with baseline. The updated result section reads now as follows:

In Figure 3b-d, we depict genetic variants that either have a significant pharmacogenetic effect γ_D , baseline effect β_0 or both, and showcase how baseline adjustment can introduce a bias in genetic effect estimation. To this end, we compare genetic effect sizes of biomarker differences in medication-naïve controls to those in statin users (simvastatin 40mg users who represent the largest starting statin type-dose group, Table S5). The SNP rs7412 in the APOE region which is strongly associated to baseline levels ($\beta_0 = 0.634$, p-value < $1e-300$) also exhibits a pharmacogenetic effect ($\gamma_D = 0.284$, p-value = $3.52e-19$) while not being associated to longitudinal change in statin-free controls (p-value = 0.59; Figure 3b; Table S7). However, upon baseline adjustment, a significant genetic effect with longitudinal change is observed in drug naïve individuals ($b = 0.256$, p-value = $4.83e-102$) as well as a stronger association in statin users due to the β_0^2 bias ($b = 0.409$, p-value = $1.07e-39$). Since this SNP is strongly associated with baseline levels, its effect on drug response is overestimated when adjusting for baseline and a spurious association is observed for longitudinal change without lipid-lowering treatment. The SLCO1B1 missense variant rs4149056 was not associated with total cholesterol baseline levels (p-value = 0.078), and genetic effects remained similar between baseline adjusted and unadjusted results (adjusted: $\gamma_D = 0.092$, p-value = $3.16e-08$; unadjusted: $\gamma_D = 0.092$, p-value = $1.18e-07$; Figure 3c), evidencing the sole implication of SLCO1B1 in pharmacokinetics (no significant association was found for longitudinal change either; p-value > 0.24). In contrast, the SNP rs11076175 in the CETP locus is strongly

associated with HDL-C baseline levels ($\beta_0 = 0.262$, $p\text{-value} = 4.26e\text{-}311$), but had no significant pharmacogenetic effect in the unbiased model ($\gamma_D = 0.006$, $p\text{-value} = 0.70$). Upon baseline adjustment, strong associations with both longitudinal change and drug response were observed ($p\text{-values}$ of $4.85e\text{-}33$ and $7.96e\text{-}06$, respectively; Figure 3d). Together, these examples illustrate how the genetic component of baseline levels can result in the identification of false positive associations and/or overestimation of pharmacogenetic effects.

3. The study samples and demographics are minimally described. It is unclear to me how concomitant meds were handled.

We agree with the reviewer that this section was previously lacking important information on the followed methodology. We have updated the Method section: *Study design and drug response phenotypes* and to expanded how concomitant medications were handled.

To determine medication regimens (medication start, treatment changes, prescription regularity), we first extracted all available prescriptions for each broader medication class (lipid-regulating, antidiabetic including insulin, and antihypertensives; BNF and Read V2 codes in Table S2; when BNF codes were truncated to miss the drug ingredient, we extracted them by matching drug names and brand names in the drug description). We then selected individuals with entries of the medication of interest (primary medication) and omitted individuals taking medications other than the primary medication of the same class within a year of initiating the primary medication (in the stringent filtering setting, only monotherapies within the same medication class were allowed and no prior related medication prescriptions were permitted). This criterium was relaxed in the lenient filtering setting where the primary medication could act as add-on therapy in certain scenarios, with the related medication included as a covariate (Table S3). More specifically, the following concomitant medication regimens were allowed in the lenient filtering setting:

- 1) *Statins: Non-statin antilipemic medication prescriptions (e.g. fenofibrates) prior to statin prescriptions were permitted and included as covariate.*
- 2) *Metformin: Prior sulfonylureas prescriptions were permitted (covariate) as well as metformin acting as add-on to sulfonylureas (additional covariate, which was defined by recorded sulfonylureas prescriptions during baseline and post-treatment periods). Individuals with sulfonylureas prescriptions recorded only during the post-treatment period (and not baseline period) were filtered out.*
- 3) *First-line antihypertensives: Antihypertensives were allowed to act as add-on to beta blockers (covariate) and loop diuretics (additional covariate) if prescriptions were consistently recorded during baseline and post-treatment periods as illustrated in Figure S2.*
- 4) *Beta blockers in the analysis with SBP we excluded individuals taking any other antihypertensives except for loop diuretics (covariate, same as described with first-line antihypertensives). In both the stringent and lenient analysis with HR, prescriptions of antihypertensives other than beta blockers were permitted since their effect on HR is much weaker than for beta blockers (Materson et al., 1998).*

Note, that in both stringent and lenient filtering settings, individuals taking primary medications in combination with a medication of the same class (e.g. statins in combination with ezetimibe) were filtered out (Note S3). Prescriptions during the post-treatment period of medications of the same class as the primary medication (except for the allowed concomitant medication regimens in the lenient filtering setting described above) were defined as treatment changes, and accordingly, individuals were excluded from the respective drug cohort. A dose change, with dose information retrieved from the drug description using regular expressions (McInnes

et al., 2020), was defined as treatment change in the stringent, but not in the lenient filtering setting where the average over all prescriptions in the post-treatment period was used. Note that dosage information is not readily available, and we assume the dose information, available in the description of the medication, to correspond to the daily dosage. Since a given primary medication as defined herein can comprise multiple drugs (e.g. simvastatin and atorvastatin for statins, ramipril and lisinopril for ACEi), we only included drugs taken by at least 20 individuals (both in the stringent and lenient filtering setting). Finally, we defined a prescription regularity parameter by the presence of a prescription at least every two months for the duration of the post-treatment period with the completeness being 100% if this was the case (we required a completeness of 60% and 30% in the stringent and lenient filtering setting, respectively). In Table S5, we show for each drug response cohort the number of individuals per drug type and dose which constituted covariates in all drug response analyses.

4. Authors veer in to interpretation and limitations at times during the results section (e.g., pg 12 "While this could be a power issue given the lower sample size...") rather than focusing on the findings.

We have updated the result section to remove vague interpretation statements such as the one pointed out by the reviewer. We have reworded the phrasing of this sentence as follows:

GWAS of HbA1c-response to metformin identified ATM (Godarts et al., 2011), SLC2A2 (Zhou et al., 2016) and PRPF31 (Rotroff et al., 2018), but only the SLC2A2-associated SNP (discovered in a baseline-unadjusted analysis) was recovered at nominal significance (p-value of 3.89E-02 and 4.07E-02 in the absolute and relative biomarker change model, respectively). SLC2A2 was discovered in a sample size of 10,577 individuals (Zhou et al., 2016), whereas sample sizes were 4,424 and 3,845 in the UKBB and AoU, respectively. Nonetheless, it should be noted that none of the metformin studies have reported the same locus twice and the ATM and SLC2A2 loci were insignificant in the ACCORD clinical trial GWAS that was conducted later (p-value > 0.1) (Rotroff et al., 2018).

5. How were PRS drug outcomes ("better or worse response") categorized/determined?

We thank the reviewer for pointing out the wording of the introductory sentence of the PRS result section "Polygenic risk scores as predictors of drug response". We believe that the chosen phrasing may have suggested a dichotomized categorization of drug outcomes as the original phrasing read: *We assessed whether high PRS of the underlying biomarker contribute to better or worse drug response outcomes.* To avoid confusion, we have changed the sentence to the following:

We assessed whether high PRS of the underlying biomarker contribute to increased or decreased biomarker reductions in medication users of the UKBB. [...]

6. The discussion begins by stating this study presents a theoretical framework to model drug response & longitudinal phenotypes (and later that this study "elucidates the theory of modelling biomarker differences"). This is not how the rest of the manuscript is presented. Suggest moderating this phrasing or introducing what the framework is to a much greater extent.

We would like to clarify that we do not start the discussion with the theoretical framework, but first explain all the main results and then finish the paragraph by mentioning that response definition requested a theoretical justification. If we added more explanation to the last part, it would steal focus from the more important aspects. However, we have added a new

Discussion paragraph where we discussed and compared our proposed pharmacogenetic model to models previously used for drug response and longitudinal change. This paragraph reads as follows:

As similar study designs have used differing GWAS models to estimate pharmacogenetic effects, we developed a realistic model for biomarker differences and showed how baseline adjustment will induce biases for genetic variants associated with baseline levels. In recent years, there have been debates about baseline adjustment in analyses of change (Glymour et al., 2005, McArdle and Whitcomb, 2009, Oni-Orisan et al., 2020, Zhang et al., 2022). In the context of PGx GWAS, some researchers have advocated for no baseline adjustment (Oni-Orisan et al., 2020), while others have argued otherwise (Zhang et al., 2022). There is a consensus that measurement errors can introduce a bias and spurious associations with longitudinal change upon baseline adjustment (regression-to-the-mean bias) (Blomqvist et al., 1987, Glymour et al., 2005, Tu et al., 2007, McArdle and Whitcomb, 2009, Oni-Orisan et al., 2020, Zhang et al., 2022), however, it is unclear whether baseline adjustment also affects drug response phenotypes since drug effects are likely larger than measurement errors. In the context of pharmacogenetic interactions with biomarker change, only the direct genetic effect on change is of interest, but genetic variants likely also affect baseline values which in turn impact the magnitude of change induced by the medication. Zhang et al., 2022 recommend adjusting for baseline as their model represents baseline as a mediator between genetic variants and quantitative change (Zhang et al., 2022). We argue that the mediator model is inappropriate because it assumes that the change depends equally (i.e. same regression coefficient) on the genetic and the environmental component of the baseline value. However, their model does not test for this gene-environment “equivalence” and in all applications we examined, this is not the case: the change depends very little on the baseline genetics and almost all its association is with the environmental component of the baseline value. Hence adjusting for the baseline value rather than eliminating a bias, it introduces one. For this reason, we, instead, propose modeling drug response as the biomarker difference between two time points. We provide theoretical derivations as to the origin and magnitude of the bias arising from baseline adjustment and demonstrate in GWAS on drug-naïve control individuals that our model, which supports not adjusting for baseline, does not generate inflated type I errors as suggested by the authors (Zhang et al., 2022). Furthermore, since control individuals are likely subjected to similar measurement errors and gene-environment interactions as medication users, the control experiments reassure that these effects are negligible compared to the pharmacogenetic effects. We show that for unbiased pharmacogenetic effect estimation, both absolute and logarithmic relative biomarker differences [...]

7. Can authors comment on the choice to use All of Us at the replication dataset? It seems other biobanks would be better datasets given the limited EHR data.

We agree with the reviewer that there are other biobanks with genetic data coupled to longitudinal medication prescriptions such as FinnGen, the Estonian Biobank, and the GERA cohort. However, it is not straightforward to obtain access to these biobanks. On the other hand, the UK Biobank and now the All of Us are the biobanks that have protocols in place to grant access to any approved researcher. Thus, the choice of the All of Us was the result of this biobank comprising genetic and longitudinal medication data while also being publicly available. Also, as many other researchers are faced with the same choice of biobanks, we believe that there is great value in presenting these data even if the conclusion is that medication prescriptions are unfortunately not as complete in the All of Us database compared to the UK Biobank.

8. Surprisingly, the limitations do not comment on the lack of diversity in this study.

We acknowledge that the lack of diversity in this study was missing and have added this limitation to the Discussion:

[...]. Sixth, analyses have been conducted in individuals of European ancestry. While allele frequencies of identified PGx signals were largely consistent across genetic ancestry groups (Table S10), analyses in diverse ancestry groups are likely to identify PGx effects missed in individuals of European ancestry. For a PGx signal with modest effect size such as SLCO1B1, we estimated that a sample size of ~150,000 is required for 80% power in African/African Americans (MAF of 3.1% compared to 15% in Europeans), showcasing that even with cohorts the size of the current study increased numbers of PGx signals could be identified provided that the MAF is high. Finally, [...]

Furthermore, allele frequencies of the identified PGx signals across ancestral groups have been added and we expanded the result section “Replication analysis in the All of Us research program” to describe these differences. We added a power analysis to analyze the impact of allele frequency differences on the needed sample sizes in discovery analyses.

Participants in the AoU biobank represent a more diverse range of genetic ancestries compared to those in the UKBB, with only 50% being of European ancestry (The All of Us Research Program Genomics Investigators, 2024). When assessing allele frequencies of the identified PGx signals across ancestries by using the gnomAD v4.1.0 resource (Chen et al. 2024), the largest allele frequency differences were observed for APOB, ranging from 5.2% in East Asians to 52.2% in the Amish (Table S10). Significant differences were also found for LDLR (0.8%-18.8%) and SLCO1B1 (3.1%-20.9%), while MAFs of other PGx variants varied by less than 10%. Based on this MAF spectrum, we conducted power analyses to determine the sample sizes required to detect PGx signals at genome-wide significance ($5e-08$) in different ancestral groups. For a strong signal, such as rs7412 at the APOE locus ($\beta = 0.232$, $p\text{-value} = 1.11e-28$), a sample size of 4,000 is needed to achieve 80% power in African/African Americans where the MAF is the highest (10.5%) compared to 28,500 in Amish people where the MAF is the lowest (1.2%). For a more modest signal, such as rs4149056 at the SLCO1B1 locus ($\beta = -0.063$, $p\text{-value} = 1.10e-08$), a sample size of 148,501 is needed for 80% power in African/African Americans where the MAF is the lowest (3.1%) compared to 27,501 in the Finnish where the MAF is the highest (20.9%; Figure S11).

9. The conclusion leaves me wanting in terms of next steps and how this data will be clinically relevant.

We have updated the conclusion sentence to provide more concrete avenues on how we see the next steps with respect to larger sample sizes. Furthermore, we emphasize again that the focus of this work is on the use of EHRs and the associated benefits and challenges.

To conclude, we show that EHRs enable new opportunities to study the genetics of drug response at scale and in a cost-effective manner. Although challenges remain with respect to the completeness and frequency of medication prescriptions and biomarker measures, these data allow us to shed light on the complex contribution of genetic and environmental components to drug efficacy. We find that the influence of common and rare genetic variants on drug response is relatively low, and larger sample sizes achieved by combining drug response GWAS from observational, EHR and RCT data as well as studies in more diverse ancestries will be needed to capture the full extent.

10. What was the rationale for the stringent/lenient time periods? Why take the average of all post-tx measures?

We have added the rationale for the stringent and lenient time periods to the new Result section *The impact of filtering in the PGx-EHR study design*. This reads as follows:

A main difference between the lenient and stringent filtering scenario was the extension of the baseline and post-treatment periods. In Figure S4, we show the distribution of the time between the closest baseline measure and prescription start, and in Figure S5, the distribution of the time between prescription start and the closest post-treatment measure. The median of this distribution was between 0 and 41 days for the baseline measure and 111 and 273 days for the post-treatment measure across medication-biomarker pairs (Table S4). As we anticipate the stringent filtering scenario to more closely reflect the design of an RCT such as the JUPITER trial (LDL-C response to rosuvastatin) where the post-treatment value was taken 1 year following medication start (Chasman et al., 2012), we chose a baseline period starting 100 days before prescription start and a post-treatment period starting 60 (antihypertensives and beta blockers) or 100 days (statins and metformin which have delayed effects on lipids (Tsiara et al., 2003) and HbA1c (Ulcickas et al., 2006) up to 1 year following prescription start. In the lenient filtering, we extended the baseline period up to 1 year preceding and up to 2 years following prescription start. Note that the post-treatment period in the lenient filtering setting remains more stringent than in observational studies such as those included in the GIST study where the post-treatment period varied widely between cohorts and could last up to 5 years (Postmus et al., 2014). When testing the impact of the time to first post-treatment measure on the biomarker difference, we observed that increased follow-up times resulted in reduced biomarker differences, even in the stringent filtering scenario where this parameter could not vary as widely (the follow-up time explained up to 1.1% of the variance, Table S6).

Furthermore, we added a discussion point as to why we use the average of biomarker levels.

Furthermore, we tested whether there is a benefit in averaging over multiple baseline and post-treatment measures if available. Averaging over multiple measures should result in a more robust measure of the underlying biomarker level and reduce the chance picking up regression-to-the-mean effects as we discuss in the following paragraph. The only difference we observed between these two strategies was that SLCO1B1 reached genome-wide significance when averaging across multiple measures (p -value of $1.10e-08$ compared to $1.49e-05$).

11. Was dosage accounted for in any modeling?

In all the drug response GWAS, we accounted for medication dose. We have updated the method section *Study design and drug response phenotypes* to state this point more explicitly and we have added a Supplementary Table 5 where we show the number of individuals per drug and dose. The updated paragraph reads as follows:

Note that dosage information is not readily available, and we assume the dose information, available in the description of the medication, to correspond to the daily dosage. Since a given primary medication as defined herein can comprise multiple drugs (e.g. simvastatin and atorvastatin for statins, ramipril and lisinopril for ACEi), we only included drugs taken by at least 20 individuals (both in the stringent and lenient filtering setting). Finally, we defined a prescription regularity parameter by the presence of a prescription at least every two months

for the duration of the post-treatment period with the completeness being 100% if this was the case (we required a completeness of 60% and 30% in the stringent and lenient filtering setting, respectively). In Table S5, we show for each drug response cohort the number of individuals per drug type and dose which constituted covariates in all drug response analyses.

12. Who were the controls in the GWAS?

In the GWAS on controls, we included individuals with EHR data who lacked prescriptions affecting the biomarker under study: controls in the lipid-change GWAS had no lipid-lowering medications, those in HbA1c-change GWAS had no antidiabetic medications, those in blood pressure-change GWAS had no antihypertensives, and those in heart rate-change GWAS had no beta-blockers. The longitudinal GWAS in medication-naïve individuals was then compared to the drug response GWAS. While medication-naïve controls are not required to estimate pharmacogenetic effects, this comparison helps confirm that the genetic effects identified are drug-specific and not due to baseline genetics.

We have renamed the Method section to *Longitudinal biomarker change GWAS in controls* and describe the definition of control individuals more clearly:

We conducted biomarker change GWAS in control individuals that were part of the primary care data and that did not have any drug prescription indicated for the investigated disease/surrogate endpoint: controls in the lipid-change GWAS had no lipid-lowering medications, those in HbA1c-change GWAS had no antidiabetic medications, those in blood pressure-change GWAS had no antihypertensives, and those in heart rate-change GWAS had no beta-blockers (i.e., broad medication class, Table S2). All participants in this set with two available measures spaced between 6 months and 3 years which corresponds to the allowed time interval between baseline and post-treatment measures were included. GWAS analyses were conducted analogous to the drug response GWAS, replacing baseline with first and post-treatment with second phenotype measure. We used the same covariates as in the corresponding drug response cohorts omitting drug-specific variables (Table S3).

Furthermore, we clarified the purpose of the control individuals more explicitly in the Result section *Modelling drug response and longitudinal change phenotypes*:

Thus, the pharmacogenetic effect can be estimated from genetic regression analyses on the biomarker difference at post-treatment and baseline. Control individuals who do not take any related medications are not required for this estimation, however, analyzing longitudinal changes in these individuals (the drug status being zero at both time points) serves as a control to ensure that identified PGx signals are specific to drug treatment. Figure S12 presents the directed acyclic graph (DAG) that extends the graph in Figure 3a for modeling the genetics of drug response phenotypes ΔY . This expression also holds when modelling the logarithm of the biomarker level. [...]

13. What method was used to calculate PRS?

We updated the Method section “PRS and genetic correlations” to provide more details about the method that was used to calculate PRS:

We calculated PRS for UKBB participants with the PGS Catalog Calculator (PGS Catalog Calculator v2.0; Lambert et al., 2021) using pre-calculated genetic effect sizes from the PGS Catalog (Privé et al., 2022): LDL-C, PGS002150; HDL-C, PGS002172; TC, PGS002108;

HbA1c, PGS002171; SBP, PGS002228; HR, PGS002193. These PRS genetic effects stem from the LDpred2-auto method (implemented in the R package bigsnpr) which assumes a point-normal mixture distribution for effect sizes, with only a proportion of causal variants contributing to the SNP heritability (Privé et al., 2020, Privé et al., 2022).